

# An improved mechanistic model for ammonia volatilization in Earth system models: Flow of Agricultural Nitrogen, version 2 (FANv2)

Julius Vira[1], Peter Hess[1], Jeff Melkonian[2], and William R. Wieder[3,4]

[1]Department of Biological and Environmental Engineering, Cornell University, Ithaca, NY, USA
[2]Section of Soil and Crop Sciences, Cornell University, Ithaca, NY, USA
[3]Climate and Global Dynamics Laboratory, National Center for Atmospheric Research, Boulder, CO 80307, USA
[4]Institute of Arctic and Alpine Research, University of Colorado, Boulder, CO 80309, USA

**Correspondence:** Julius Vira (julius.vira@gmail.com)

**Abstract.** Volatilization of ammonia ($NH_3$) from fertilizers and livestock wastes forms a significant pathway of nitrogen losses in agricultural ecosystems, and constitutes the largest source of atmospheric emissions of $NH_3$. This paper describes a major update to the process model FAN (Flow of Agricultural Nitrogen), which evaluates the $NH_3$ emissions interactively within an Earth system model; in this work, the Community Earth System Model (CESM) is used. The updated version (FANv2)

includes a more detailed treatment of both physical and agricultural processes, which allows the model to differentiate between the volatilization losses from animal housings, manure storage, grazed pastures, and from application of manure and different types of mineral fertilizers. FANv2 is connected to the interactive crop model within the land component of CESM, which determines the amount and timings of fertilizer applications for major types of crops. The model is first evaluated at local scale against experimental data for various types of fertilizers and manure, and subsequently run globally to evaluate present-day

$NH_3$ emissions. Comparison of regional emissions shows that FANv2 agrees with previous inventories for North America and Europe, and is within the range of previous inventories for China. However, due to higher $NH_3$ emissions in Africa, India and Latin America, the global emissions simulated by FANv2 (47 Tg N) are 30–40 % higher than in the existing inventories.

## 1 Introduction

Volatilization of ammonia ($NH_3$) from livestock wastes and synthetic fertilizers forms a globally significant pathway of nu-

trient losses in agricultural ecosystems (Bouwman et al., 1997; Beusen et al., 2008). Once emitted to atmosphere, ammonia contributes to formation of secondary aerosols with implications for public health and climate (Heald et al., 2012; Paulot et al., 2016). Deposition of ammonia and other reactive nitrogen species onto natural ecosystems has widely documented adverse effects on biodiversity (Duprè et al., 2010; Payne et al., 2017), but also potentially significant effects on the ecosystem productivity (Zaehle and Dalmonech, 2011). Thus, the atmospheric emission, transport and deposition of ammonia forms a societally

and ecologically important part of the global nitrogen cycle.

Based on global emission inventories, atmospheric chemistry models have been used extensively to evaluate the global and regional deposition of reduced nitrogen (Dentener et al., 2006; Vet et al., 2014). More recently, Earth system models have begun to include representations of the terrestrial N cycle and its interaction with atmospheric reactive N transport and





deposition. As an example, in the Community Earth System Model (CESM), the N deposition from the atmospheric model can be coupled interactively to the land surface model to evaluate its impact on terrestrial N cycle. However, the agricultural and soil N emissions used in atmospheric models are typically prescribed by a static inventory. Consequently, even if the land and atmospheric models are run interactively, the emitted N is not subtracted from the soil nitrogen pools, nor do the emissions

respond to changes in the simulated meteorological forcing. A fully coupled land-atmosphere N cycle and its interaction with climate has therefore yet to be simulated.

As a step towards more consistent simulation of the land-atmosphere N cycle, this study focuses on evaluating the ammonia volatilization interactively within the Earth system model. Reproducing the geographic and temporal variability in $NH_3$ emissions requires consideration of not only agricultural practices, but also environmental factors such as temperature and pre-

cipitation (Bouwman et al., 2002). The environmental factors are especially important when using the model under projections of future climate, since although the volatilization losses are believed to be sensitive to changes in surface temperature and precipitation (Sutton et al., 2013), the response of global $NH_3$ emissions to climate drivers has so far not been quantified in detail.

The process model FAN (Flow of Agricultural Nitrogen) described by Riddick et al. (2016) was developed in part to assess

the climate sensitivity of ammonia volatilization. In contrast to specialized models developed to evaluate ammonia emissions arising in application of manure slurry (Genermont and Cellier, 1997; Hamaoui-Laguel et al., 2011), synthetic fertilizers (Rachhpal-Singh and Nye, 1986; Bash et al., 2013; Xu et al., 2019), or from urine patches on pastures (Sherlock and Goh, 1985; Móring et al., 2016; Giltrap et al., 2017), FAN aims to evaluate $NH_3$ emissions globally and throughout the agricultural sector.

The present paper describes a major update to the first version of FAN (Riddick et al., 2016, hereafter FANv1) with improvements in representation of both soil processes and agricultural practices. The new version (FANv2) includes a more detailed treatment of diffusion, leaching, and adsorption of ammonium in soil, and a new numerical scheme links the simulated local processes to the spatial scales resolved by an Earth system model. The additional mechanistic detail in FANv2 allows the model to avoid some of the simplifications made in FANv1. In particular, FANv1 treated all fertilizers as urea, and included only a

generic type of manure, while FANv2 reproduces the higher volatilization losses of urea compared to other synthetic fertilizers and includes separate sub-models for $NH_3$ volatilization from pastures and from mechanically spread manure.

Besides the additional process-level detail, FANv2 has an improved representation of agricultural practices. Similar to FANv1, the model is integrated to the Community Land Model (CLM), which forms the land surface component of CESM, but unlike FANv1, FANv2 makes use of the interactive crop model included in CLM (Lawrence et al., 2019; Lombardozzi et al.,

2019) to determine the timing and amounts of fertilization appropriate to each crop. FANv2 also incorporates a parameterization (Gyldenkærne et al., 2005) for evaluating volatilization losses from manure in animal housings or during storage.

In this study FANv2 is run globally within the CLM for the six-year period 2010–2015 to simulate the present-day $NH_3$ emissions, which are then compared with existing global and regional inventories. The ammonia emissions from FANv2 are also evaluated against local measurements of $NH_3$ emissions from various types of synthetic fertilizers and manure, under vari-

ous environmental conditions. The model formulation, the local-scale evaluation, and the global simulation setup are described



in Section 2. Section 3 presents the results of the model evaluation as well as the simulated global emissions. Discussion and conclusions are presented in Sections 4 and 5.

## 2    Methods

FANv2 simulates the flows of nitrogen stemming from manure and synthetic fertilizer application, including the volatilization
of ammonia from soils, animal housings and manure storage. The model is formulated in four steps. Section 2.2 describes the physical processes simulated by FANv2. Section 2.3 introduces an upscaling scheme for linking these patch-scale processes to grid-scale emission fluxes in the CLM, and Section 2.4 describes how the generic approach outlined in the preceding sections is applied to specific agricultural processes. Finally, Section 2.5 describes the representation of global agriculture and animal husbandry in the model.

### 2.1    The Community Land Model

The FANv2 process model was implemented as an extension to the CLM version 5 (CLM5), which forms the terrestrial component of the CESM version 2. The CLM simulates the key input variables required by FANv2, including soil temperature and moisture, precipitation infiltration, and the resistances describing the exchange between the soil surface and the atmospheric boundary layer. Furthermore, the interactive crop model (Levis et al., 2012, 2018; Lombardozzi et al., 2019) in CLM5 deter-
mines the amount and timing of fertilizer application in FANv2. Since the present study focuses on the emissions of $NH_3$, the coupling between FANv2 and CLM is unidirectional: the soil properties and fertilization simulated by the CLM were used to drive FANv2, but the simulated N losses did not affect the remaining terrestrial nitrogen cycle simulated by CLM.

The CLM uses a hierarchical structure to represent sub-grid scale variations in the soil and vegetation. Each CLM grid cell may contain several soil columns with different moisture and carbon and N pools, and each column contains one or more plant
functional types. For natural vegetation, the soil columns are shared among plant functional types. Each crop type, in contrast, corresponds to a single soil column. Since the primary input variables in FANv2 are related to N cycling and hydrology in the CLM, FANv2 is introduced into the CLM sub-grid structure on the soil column level.

### 2.2    Soil processes in FANv2

Similar to FANv1, the main N species solved for in FANv2 is the total ammoniacal nitrogen (TAN), which consists of gaseous,
dissolved, and adsorbed $NH_3$ and ammonium ($NH_4^+$). Both FANv1 and FANv2 include additional N species representing organic precursors to TAN; this includes urea and two organic N fractions for manure. However, compared to FANv1, FANv2 includes more detailed formulations of the transport of TAN in soil.

Similarly to the models of Sherlock and Goh (1985), Móring et al. (2016), and Giltrap et al. (2017), FANv2 solves the nitrogen budget for a thin soil layer immediately below the surface. In FANv2, the layer covers the topmost $\Delta z = 2$ cm of
the soil profile, which coincides with the topmost soil layer in CLM5. The soil below the topmost layer is treated as a sink with a TAN concentration much lower than the topmost layer, and all N transported below the 2 cm layer is assumed to be





permanently unavailable for volatilization. FANv2 does currently not interact with the base nitrogen cycle in CLM, and the effects of microbial immobilization and plant uptake of fertilizer and manure N are therefore not simulated. These constraints may be relaxed in future versions as FANv2 is integrated more closely into the CLM nitrogen cycle.

The budget of TAN or other simulated N species within the soil layer can be written as

$$\frac{dN}{dt} = f(N,t) = P - R - D - Q - M \tag{1}$$

where $N$ (g m$^{-2}$) is the mass (per surface area) of the particular N species within the layer, and the terms on right denote the production or inputs of the nitrogen species ($P$), reactive losses $R$ due to chemical and biological processes, the net diffusive flux $D$ (including the volatilization loss) in the aqueous and gas phases, and the leaching flux $Q$ in aqueous phase. The term $M$ denotes losses due to bioturbation (disturbances caused by living organisms) and other mechanical disturbances. This "mechanical" loss $M$ is evaluated similarly to Riddick et al. (2016) as a first order process with a constant time scale of one year, which makes it mainly significant for the organic N species whose decay time constants in FANv2 are comparable to that of $M$.

The simulated N transformations are the nitrification of ammonium, hydrolysis of urea, and mineralization of organic N, which are all simulated with first order kinetics with rate expressions given in the Appendix A. The nitrification rate depends on temperature and moisture following a modified version of the formulation of Stange and Neue (2009) as described in Riddick et al. (2016). The decomposition of urea is also simulated as in FANv1; a fixed time scale of 2.4 days is used for synthetic fertilizers, whereas urea in manure is introduced directly into the TAN pool.

The N in other organic compounds within manure is split into available, resistant, and unavailable fractions. The N in the resistant and available fractions mineralizes at temperature and moisture dependent rates, while the unavailable fraction does not contribute to the TAN pools in FAN. The mineralization rates used in FANv2 include the temperature dependency used in FANv1, but FANv2 adds a moisture-dependent multiplicative factor to avoid unrealistically fast mineralization in warm but dry conditions. The moisture-dependent factor (Eq. A19) is the same as used in CLM for decomposition of soil organic matter (Lawrence et al., 2018b).

The prognostic equation (1) for TAN can be expanded into

$$
\begin{aligned}
\frac{dN_{TAN}}{dt} &= I_{TAN} - F_{atm} \\
&+ k_U N_U + k_A N_A + k_R N_R - k_N N_{TAN} - k_m N_{TAN} \\
&- F_{\downarrow}^{TAN} - Q_r^{TAN} - Q_p^{TAN}
\end{aligned} \tag{2}
$$

where $I_{TAN}$ denotes the rate TAN is applied to the soil. $N_U$, $N_A$ and $N_R$ refer to TAN precursors in forms of urea and available and resistant organic N, and $k_U$, $k_A$ and $k_R$ are the decomposition rates of each precursor. The coefficients $k_N$ and $k_m$ denote the rates of nitrification and removal due to mechanical disturbances. The diffusive flux $D$ is split into the atmospheric flux $F_{atm}$ and the aqueous and gaseous downward diffusion out of the thin soil layer, $F_{\downarrow}^{TAN} = F_{aq\downarrow} + F_{gas\downarrow}$. The leaching flux $Q$ is split into surface runoff $Q_r$ and subsurface leaching $Q_p$.





The prognostic equations for urea and organic N fractions are similar to Eq. (2) with straightforward modifications given in Appendix A. The diffusion and leaching fluxes are not evaluated for the available and resistant organic N. For urea, the gaseous fluxes are not evaluated, but in contrast to FANv1, FANv2 allows urea to to be transported by leaching and diffusion in the aqueous phase. The chemical production terms corresponding to TAN formation are omitted for urea and other organic N, and conversely, the nitrification rate $k_N$ is replaced by the corresponding decomposition rate.

The fluxes of TAN within soil depend fundamentally on the partitioning between the gaseous, dissolved, and adsorbed forms of TAN. By combining the Henry's law for ammonia and the chemical equilibrium between the dissolved ammonia and the ammonium ion (e.g. Sutton et al., 1994), the gaseous concentration (gN m$^{-3}$ air) can be expressed using the partitioning coefficient $K_{\mathrm{NH3}}$ as

$$[\mathrm{NH_3\ (g)}] = \frac{[\mathrm{NH_3\ (aq)}] + [\mathrm{NH_4^+\ (aq)}]}{K_H + [\mathrm{H^+}]/K_{\mathrm{NH4}}} = K_{\mathrm{NH3}} \times [\mathrm{TAN\ (aq)}] \tag{3}$$

where $K_H = [\mathrm{NH_3(aq)}]/[\mathrm{NH_3(g)}]$ is the dimensionless Henry's law (solubility) constant for ammonia,

$$K_{\mathrm{NH4}} = \frac{[\mathrm{H^+}][\mathrm{NH3(aq)}]}{[\mathrm{NH4^+}]} \tag{4}$$

is the dissociation constant of $\mathrm{NH_4^+}$, and the square brackets denote concentrations of ammonia, ammonium and the hydrogen ion $\mathrm{H^+}$. The sum of $\mathrm{NH_4^+}$ (aq) and $\mathrm{NH_3}$ (aq) is denoted by TAN (aq). The aqueous solutions are are assumed to be dilute, so that effects of ionic strength are neglected.

Soils may adsorb some of the TAN due to cation exchange. While neglected in FANv1, FANv2 simulates the adsorption according to a linear isotherm (e.g. Bear and Verruijt, 2012),

$$[\mathrm{TAN\ (s)}] = K_d \times [\mathrm{TAN\ (aq)}], \tag{5}$$

where $K_d$ (m$^3$ m$^{-3}$) is the partitioning coefficient and [TAN (s)] denotes concentration of sorbed ammonium with respect to volume of soil solids.

Adsorption of $\mathrm{NH_4^+}$ varies between different soils (Buss et al., 2004; Sommer, 2013). However, simulating this in FANv2 would require a more detailed characterization of soil chemistry than is currently available in CLM or other global models. Thus, FANv2 assumes a constant $K_d = 1.0$ chosen based on the comparison with observed volatilization losses (Section 2.6). Assuming a soil particle density of 2.6 g cm$^{-3}$, $K_d = 1.0$ is equal to $\sim 0.4$ ml g$^{-1}$, which is within the overall range presented in Buss et al. (2004).

The aqueous and gaseous concentrations are defined here with respect to the water or air filled pore volume, and are therefore related to the TAN pool $N_{TAN}$ and the adsorbed N as

$$N_{\mathrm{TAN}} = \Delta z \left( \theta[\mathrm{TAN\ (aq)}] + \varepsilon[\mathrm{NH_3\ (g)}] + (1 - \theta_s)[\mathrm{TAN\ (s)}] \right), \tag{6}$$

where $\theta$ is the volumetric soil water content (m$^3$ water m$^{-3}$ soil), $\varepsilon$ is the fraction of air-filled soil volume (m$^3$ air m$^{-3}$ soil). The air fraction is evaluated using the soil water content $\theta_s$ at saturation as $\varepsilon = \theta_s - \theta$. The chemical equilibria (Eqs. 3 and 5) are assumed instant, and consequently, only the total TAN pool $N_{\mathrm{TAN}}$ needs to be evaluated prognostically.



**Figure 1.** A resistance scheme representing transport processes between atmosphere, soil immediately below surface, and the deeper soil. The aerodynamic and quasi-laminar layer resistances are denoted by $R_a$ and $R_b$. Resistances controlling the diffusive transport upwards ($\uparrow$) and downwards ($\downarrow$) are denoted by $R_{aq}$ and $R_{gas}$ for aqueous and gaseous phases; runoff and leaching fluxes are denoted by $Q_r$ and $Q_p$. Phase equilibria are denoted with $\rightleftharpoons$.





The transport of TAN in FANv2 is described by the resistance diagram in Fig. 1, where the loss due to mechanical perturbation is omitted for clarity. The conceptual approach is similar to the resistance formulations for evaluating dry deposition of gases (e.g. Wesely, 1989) or the bi-directional surface exchange of $NH_3$ (e.g. Cooter et al., 2010), however, FANv2 includes explicit treatment of both aqueous and gaseous fluxes and concentrations within the soil layer. This is achieved with the parallel

soil resistances ($R_{aq}$ and $R_{gas}$ in Fig. 1), which are discrete analog of the two-phase diffusion analyzed in detail by Tang and Riley (2014).

The exchange of $NH_3$ between the soil surface and the atmospheric boundary layer is controlled by the aerodynamic and quasi-laminar resistances $R_a$ and $R_b$. Below the soil surface, TAN is transported diffusively in the gas and aqueous phases, or advectively in soil water. In FANv2, the dissolved TAN and urea can be leached either by surface runoff, representing lateral

transport along the soil-air interface, or by percolating soil water, representing vertical transport within the soil column.

Following the resistance analogy, the surface flux of $NH_3$ can be expressed using the $NH_3$ concentration [$NH_3$ (g,sfc)] at the soil-atmosphere interface,

$$F_{atm} = \frac{[NH_3 \ (g, sfc)] - [NH_3 \ (g, atm)]}{R_a + R_b} \tag{7}$$

where [$NH_3$ (g, atm)] denotes the concentration at the atmospheric reference height consistent with $R_a$. The surface concentra-

tion [$NH_3$ (g,sfc)] is a diagnostic variable determined by atmospheric concentration [$NH_3$ (g, atm)] and the TAN concentration in soil.

The diffusive fluxes in soil are defined similarly to the atmospheric flux with resistances evaluated from the molecular diffusivities in soil:

$$F_*^{\uparrow} \quad = \quad R_{*,\uparrow}^{-1} \left( [TAN \ (*, sfc)] - [TAN \ (*, soil)] \right) \tag{8}$$

where $*$ denotes either the aqueous or gaseous phase, and the soil resistances are given by

$$R_*^{\uparrow} \quad = \quad \frac{\Delta z}{2\xi_*(\theta) D_*}. \tag{9}$$

The diffusion distance is taken as $\Delta z/2$ and the molecular diffusivities $D_*$ are multiplied with the tortuosity factors $\xi_*$ of Millington and Quirk (1961) to adjust for the soil porosity and water content. The aqueous-phase molecular diffusivity of ammonium (Eq. A8) is used for both ammonium and urea. The soil resistances for the downwards diffusion out of the topmost

layer (marked with $\downarrow$ in Fig. 1) are evaluated similarly to Eq. (9), but the diffusion distance is set to 3 cm, which corresponds to the distance to the midpoint of the second soil layer in CLM5.

The aqueous phase fluxes $Q_r$ (surface runoff) and $Q_p$ (subsurface leaching) are not diffusive (gradient-driven), but may nevertheless be included in the computations as

$$Q_{\mathrm{p}} \quad = \quad q_{\mathrm{p}} \times [TAN \ (aq, soil)] \tag{10}$$

$$Q_{\mathrm{r}} \quad = \quad q_{\mathrm{r}} \times [TAN \ (aq, srf)], \tag{11}$$

where $q_r$ (m s$^{-1}$) is the surface runoff flux and $q_p$ percolation flux of water at the bottom of the soil layer. An important difference between the modeled $Q_{\mathrm{r}}$ and $Q_{\mathrm{p}}$ is that the leaching flux $Q_p$ is evaluated from the mean concentration in the layer,





while the runoff flux is evaluated from the concentration at the soil surface. Thus, $Q_r$ is moderated by the resistances $R_{gas,\uparrow}$ and $R_{aq,\uparrow}$ between the soil layer and the soil surface. The runoff water flux $q_{\text{roff}}$ is evaluated by CLM, evaluating $q_p$ will be discussed in Section 2.3. The atmospheric flux $F_{atm}$ is determined by first solving the surface concentration [NH$_3$ (g,srf)] as a function of the atmospheric and soil concentrations. Conservation of mass requires that the aqueous and gaseous fluxes from

soil to the surface are equal to the sum of the volatilization and runoff fluxes $F_{atm}$ and $Q_r$,

$$F_{\text{aq}}^{\uparrow} + F_{\text{gas}}^{\uparrow} = F_{atm} + Q_r. \tag{12}$$

Using Eqs. (3) and (5) for both the surface and soil concentrations, it is possible to solve for the aqueous and gaseous concentrations at the soil-atmosphere interface and subsequently for the fluxes $F_{atm}$ and $Q_r$. The expressions are given in Appendix A.

In summary, FANv2 largely inherits its parameterizations for chemical and biological processes from FANv1 but adds a more detailed description of the processes which transport TAN within the soil. FANv1 included leaching due to runoff ($Q_R$), but not due to the vertical movement of soil water ($Q_p$). Furthermore, while diffusion of TAN in soils was included in FANv1, only downwards aqueous phase diffusion deeper into soil was considered, and adsorption of ammonium was neglected. Introducing these effects in FANv2 substantially changes the model's response to temperature and soil moisture.

The two-phase diffusion in FANv2, depicted in Fig. 1, allows TAN to be transported in either aqueous or gaseous phase within the soil layer. The relative importance of the two pathways depends on the equilibrium determined by $K_{\text{NH3}}$ and the resistances $R_{aq}$ and $R_{gas}$, which in turn depend on the water content trough the tortuosity $\xi$. This impacts how the volatilization flux $F_{atm}$ responds to changes in $K_{\text{NH3}}$, which is seen by considering the limiting cases of Eq. (A5) for low and high soil water content $\theta$. The following analysis assumes that [NH$_3$ (g,atm)] $\ll$ [NH$_3$ (g,srf)] and that the runoff flux $Q_r$ is negligible.

For low soil water content $\theta$, the aqueous phase diffusion can be neglected. By evaluating Eq. (A5) at the limit $R_{gas} \ll R_{aq}$ and substituting into Eq. (7), the atmospheric flux $F_{atm}$ is found to be proportional to

$$F_{atm} \sim \frac{N_{TAN} K_{\text{NH3}}}{(R_a + R_b + R_{gas,\uparrow})(K_d(1-\theta_s) + \theta + \varepsilon K_{\text{NH3}})} \propto \frac{K_{\text{NH3}}}{K_{\text{NH3}} + \alpha}, \tag{13}$$

where $\alpha = (K_d(1-\theta_s) + \theta)/\varepsilon$. Conversely, when the soil is near saturation, so that $R_{gas} \gg R_{aq}$ and the air-filled pore volume $\varepsilon \sim 0$,

$$F_{atm} \sim \frac{N_{TAN} K_{\text{NH3}}}{(K_d - K_d\theta + \theta)(R_a + R_b + K_{\text{NH3}} R_{aq,\uparrow})} \propto \frac{K_{\text{NH3}}}{K_{\text{NH3}} + \beta} \tag{14}$$

where $\beta = (R_a + R_b)/R_{aq,\uparrow}$.

Eqs. (13) and (14) show that the flux $F_{atm}$ is nonlinear with respect to phase equilibrium determined by $K_{\text{NH3}}$, which is in contrast to FANv1, where $F_{atm} \propto K_{\text{NH3}}/\theta$. While both FANv1 and FANv2 predict the ammonia flux to increase with temperature (Fig. 2), the joint response to soil moisture and temperature differs between the versions: in FANv1, the flux

decreases monotonously towards higher $\theta$, while in FANv2, the flux has a pH and temperature dependent minimum at $\sim$10–50 % of saturation. In FANv2 the atmospheric flux ($F_{atm}$) at pH = 8.5 is 2–10 times higher than at pH = 7, however, the temperature sensitivity is higher at the lower pH. The higher pH = 8.5 corresponds to the typical conditions following a urea





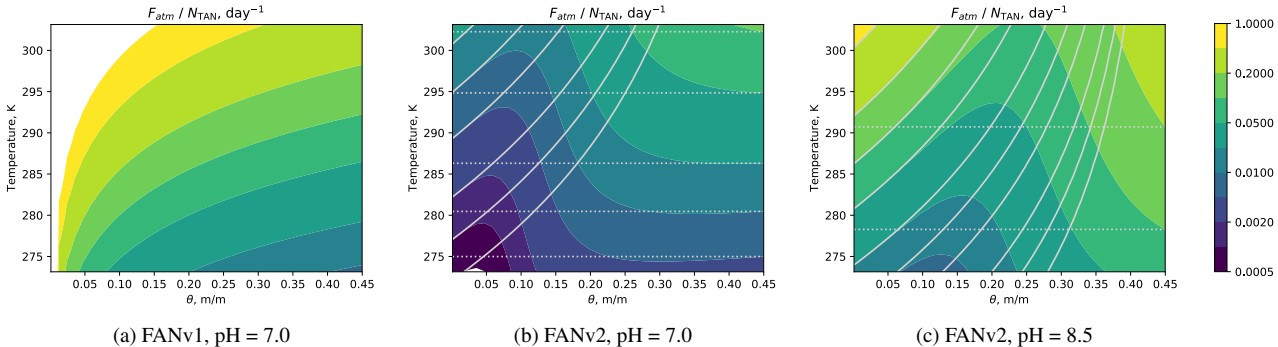

**Figure 2.** The instantaneous volatilization flux normalized with the TAN pool, $F_{atm}/N_{\text{TAN}}$ (day$^{-1}$) as a function of temperature and volumetric soil moisture $\theta$ in FANv1 (panel a) in and FANv2 at pH = 7.0 (panel b) and pH = 8.5 (panel c). In all figures, $\theta_s = 0.45$ and $R_a + R_b = 200.0$ s m$^{-1}$. The contour lines correspond to the approximations at low (solid) and high (dotted lines) water content $\theta$ (Eqs. 13 and 14).

application, as discussed in Section 2.4.4. FANv1 applies a 60 % reduction to the emissions to account for the NH$_3$ captured by plant canopy; this reduction is included in the flux shown in Fig. 2a.

Contrary to FANv1, FANv2 evaluates NH$_3$ volatilization from bare soil and excludes the effects of vegetation. Several studies have shown that presence of vegetation can significantly reduce the volatilization losses (Black et al., 1989; Whitehead and Raistrick, 1992; Sommer et al., 1997), and thus, FANv2 is likely to overestimate the NH$_3$ emission under some conditions. However, for manure, the issue is not straightforward, since depending on application method, presence of vegetation may increase volatilization by intercepting the manure spread before it reaches the ground (Sommer et al., 1997). The canopy effect might be important for fertilizers applied later during the growing season, but as noted in Section 2.5.2, this practice is not simulated by CLM. For pastures, however, the simulations might be improved by including the effect of a canopy. Ideally, this would take into account interactions between grazing and plant growth.

Although the atmospheric NH$_3$ concentration is included in Eq. (7), only gross fluxes are evaluated using FANv2 in this study, and [NH$_3$ (g,atm)] is therefore set to zero in all simulations. This is consistent with the coupling to the atmospheric model, where the dry deposition of ammonia is evaluated separately from emission. Although not evaluated here, the net NH$_3$ exchange could be obtained by subtracting the dry deposition flux from the gross emission flux.

## 2.3 Upscaling from patch to grid scale

The model described in Section 2.2 can be used to evaluate the nitrogen fluxes from a horizontally homogeneous soil patch, given the knowledge of the forcing variables such as soil temperature, moisture, pH, and the moisture fluxes $q_r$ and $q_p$. However, some of the required parameters, such as pH and soil moisture, are sufficiently affected by the addition of manure or synthetic fertilizer to influence the volatilization fluxes. The perturbations in pH and moisture evolve as time since the N addition passes, and their magnitudes depend on the type of manure or fertilizer. As a part of a global model, FANv2 needs to handle a





heterogeneous distribution of soil patches in varying states with regard to nitrogen additions. The typical dimension of the soil patches might vary from less than 1 m (urine patches) to several km (fertilized fields); in either case, the patches are small compared to $\sim$ 100 km horizontal resolution of current Earth system models.

This heterogeneity of patches is handled by assuming that the state of a nitrogen patch at a given time can be characterized by its age $a$, which we define as the time elapsed since the last N (fertilizer or manure) addition. We split each N (TAN or urea) pool into age classes and prescribe the perturbations in pH and moisture separately for each class. Thus, although the perturbations are prescribed, this approach allows using physically meaningful parameters to describe the differences between different types of N additions.

To formulate the approach mathematically, we distinguish between patch-scale nitrogen densities $N$ (gN m$^{-2}$ patch area) governed by Eq. (2), and the grid-scale nitrogen densities $n$ (gN m$^{-2}$ gridcell area). The patches of given type are divided into age classes $i$ so that the total nitrogen pool is obtained by the summation

$$n(t) = \sum_i w_i(t) N_i(t) = \sum_i n_i(t) \tag{15}$$

where $w_i$ is the fraction of gridcell area covered by patches in the $i$th age class and $n_i = w_i N_i$. The variables $N_i$ and $n_i$ can be considered as a discrete representation of a two-dimensional function describing the distribution of nitrogen across patches of different ages at a given time. This interpretation leads to a scheme for updating the $n_i$ at each model time step as follows:

1. For each $i$, update $n_i(t)$ according to Eq. (1) as

$$n_i(t') = n_i(t) + w_i(t)\overline{f\left(N_i(t),t\right)}\Delta t \tag{16}$$

where $\Delta t$ denotes the time step and the tendency $\overline{f\left(N_i(t),t\right)}$ is evaluated as a mean over the $i$th age class.

2. Transfer nitrogen from the younger to older age classes according to

$$n_i(t+\Delta t) = \begin{cases} n_i(t') - \Delta t \frac{n_i(t')}{\Delta a_i}, i = 1 \\ n_i(t') - \Delta t \left( \frac{n_i(t')}{\Delta a_i} - \frac{n_{i-1}(t')}{\Delta a_{i-1}} \right), i > 1 \end{cases} \tag{17}$$

where the ages $a_{i-1}$ and $a_i$ define the $i$th age class and $\Delta a_i = a_i - a_{i-1}$.

In FANv2, the tendency $f$ in Eq. (1) is linear with respect to $N$. Substituting $N_i = n_i/w_i$ simplifies Eq. (16) to

$$n_i(t') = n_i(t) + \overline{f\left(n_i(t),t\right)}\Delta t, \tag{18}$$

and the area fractions $w_i$ are therefore not needed. A generalization to nonlinear models, along with a more detailed rationale of Eqs. (16) and (17), is given in Section 1 of the supplement.

The fertilizer or manure N is initially introduced to the youngest age class, and subsequently transferred through the sequence of age classes as described by Eq. (17), until reaching the final class $i*$. By Eq. (17), nitrogen is removed from the final age class at a rate equal to $1/\Delta a_{i*}$, which can be made arbitrarily small by the choice of $\Delta a_{i*}$. In FANv2, the final bins have







**Figure 3.** Age-segregated nitrogen pools in FANv2 for manure TAN on pastures (G1-G3), manure TAN in slurry (S0-S3), urea N (U1-U2), TAN produced by urea hydrolysis (F1-F3) and from other fertilizers (F4). GA and GR and SA and SR represent available and resistant organic N on pastures and in slurry. The age extent $\Delta a$ in days or hours is indicated for each age class.





$\Delta a_{i*} = 360$ days, which sets the maximum age of the N patches considered. Although not implemented in the current version, the nitrogen aged beyond $\Delta a_{i*}$ could be transferred into the soil N pools in the CLM.

The variation of soil pH and water content with patch age is embedded into the evaluation of $f$. In effect, adopting the generic model, described in Section 2.2, for different sources of ammoniacal nitrogen becomes an exercise in defining the properties

of a set of nitrogen pools as a function of age and the manure or synthetic fertilizer type. FANv2 considers two types of both manure and synthetic fertilizers, each described by a TAN pool with 1 to 4 age classes, resulting in the model structure shown schematically in Fig. 3. Additional nitrogen pools are needed for organic nitrogen in the case of manure, and for unhydrolyzed urea in the case of urea fertilizer. The details of the N pools and age classes are given in the next section.

## 2.4 Applications to specific agricultural processes

The parameterization of the soil processes and the setup of the age classes depend on the agricultural practice simulated. We simulate the volatilization losses for four different processes: manure spreading, animals grazing in pastures, and synthetic fertilization modeled either as urea or a generic ammonium fertilizer.

### 2.4.1 Manure

FANv2 considers ammonia emissions separately for grazed pastures and for application of stored manure. The emissions from

manure application are simulated by the slurry sub-model (Section 2.4.3), while a simpler scheme focusing on urine patches is used for pastures (Section 2.4.2). The global attribution of manure N between the pastures and managed manure is discussed in Section 2.5.

Regardless of the form, livestock manure contains nitrogen in the form of urea and more complex organic compounds. A typical fraction of urea nitrogen in dairy cattle manure is 60 % (Sommer and Hutchings, 2001); in FANv2, this fraction is used

for all manure. The remaining manure N is split between organic N fractions with different mineralization rates as described in Section 2.2.

Decomposition of urea and other short-lived organic N forms is not evaluated explicitly within manure, as the urea contained within stored manure typically hydrolyses during the storage, and relatively short half-lives of less than 12 hours have been observed for urea within urine patches in pastures (Sherlock and Goh, 1984). Similar to FANv1, FANv2 therefore assumes that

all urea N in manure enters the soil as TAN.

Using slurry to represent used manure management and spreading practices globally is a large simplification. However, the abundance of literature on ammonia volatilization from manure slurries supports the adoption of slurry as a "prototype" of global manure management practices in FANv2.

### 2.4.2 Grazed pastures

On pastures, the manure N enters soil separately in urine and feces. In urine patches, the rapid hydrolysis of urea results in a local increase of soil pH, which exposes the newly formed ammoniacal N to rapid volatilization. Simultaneously, the





volatilization loss is reduced by the infiltration and percolation of urine deeper into the soil. In contrast, the faecal N remains on the soil surface, but the N is mineralized at a much slower rate, which normally results in much lower losses as $NH_3$ (Ryden et al., 1987).

The manure N excreted on pastures is represented by three age classes for TAN – G1, G2 and G3 – and the two organic

N pools, GA and GR (Fig. 3). The latter correspond to the available and resistant organic N fractions (see Section 2.2). The TAN pools G1 and G2 represent fresh urine patches with elevated pH and water content, and the pool G3 represents feces and old urine patches, which are simulated without changes to the ambient soil pH or moisture. The ammoniacal nitrogen is continuously transferred from the younger to older TAN age classes according to Eq. (17)

The age class G1 represents the conditions during the first 24 hours after deposition of urine. The evolution of pH in urine

patches is prescribed based on the measurements of Vallis et al. (1982), Sherlock and Goh (1984) and Laubach et al. (2012); for G1, a peak pH of 8.5 is used. In addition to the elevated pH, the urine patches initially have a higher moisture content than the surrounding soil, which affects the diffusive fluxes (Eq. 8). The water content is assumed to relax back to the background soil level during the 24 h age span of G1.

Urine is assumed to instantly infiltrate the soil, and the initial ($a = 0$) volumetric water content of urine patches ($m^3 m^{-3}$) is

evaluated as

$$\theta_0 = \min(\theta_s, d_0/\Delta z + \theta_b), \tag{19}$$

where $\theta_s$ is the volumetric water content at saturation, $d_0$ (m) is the ratio of urine volume to the area of a patch, and $\theta_b$ is the volumetric water content of unaffected soil. The parameter $d_0$ is likely to depend on the type of livestock; the value of 6 mm is adopted following Móring et al. (2016). If the soil layer becomes saturated, the excess urine is assumed to percolate directly

to the underlying soil, and the corresponding fraction of TAN is not added to the TAN pool within FANv2.

Depending on ambient conditions, the relaxation from $\theta_0$ to $\theta_b$ may consist of evaporation or vertical or lateral transport of moisture. The possible lateral spreading of urine patches is ignored in FANv2. The N leaching flux $Q_p$ is evaluated by diagnosing the flux of soil water $q_p$ at the layer bottom from the water budget of the layer,

$$\overline{q_{top}} - \overline{q_p} = \Delta z \left( \overline{\frac{\partial \theta_b}{\partial t}} + \frac{\Delta \theta}{\Delta a} \right), \tag{20}$$

where $\overline{q_{top}}$ is the net water flux (infiltration − evaporation) at the surface, the overbars denote averages over the age range $\Delta a$, and $\Delta \theta = \theta(a_{i+1}) - \theta(a_i)$. The tendency $\partial \theta_b / \partial t$ is common to all patches and evaluated within the hydrological scheme of CLM. FANv2 assumes that the evaporation rate of the urine patch can be approximated by that of the surrounding soil, so that $q_{top}$ is also taken from CLM.

Eq. (20), derived in Section 1.1 of the Supplement, states that the flux $q_p$ can be obtained from the water budget of the

unaffected soil by adding the term $\Delta \theta / \Delta a$, which expresses the rate at which the perturbed soil moisture relaxes towards $\theta_b$. Since the relaxation is assumed to occur entirely within the 24 h age span of G1, $\Delta \theta = \theta_b - \theta_0$ for G1. The soil moisture for evaluating the soil resistances for pool G1 is set to the average of $\theta_0$ and $\theta_b$, corresponding to the midpoint of the age span.





The age class G2 spans the subsequent 10 days following G1. In typical conditions, this time is sufficient for the $NH_3$ flux from the surface to decrease to close to its background level (Sherlock and Goh, 1984; Laubach et al., 2012). As noted by Sherlock and Goh (1985), the soil pH remains elevated during this stage, and accordingly, a pH 8.0 is used for G2. The soil water content in G2 is kept equal to $\theta_b$, and thus $\Delta\theta = 0$ for G2.

5     The final TAN age class G3 represents the nitrogen remaining in urine patches after 11 days, but more importantly, G3 receives the mineralized TAN from the organic N pools GA and GR, which differ in their decomposition rate (Section 2.2). The pH value for G3 is assumed equal to the unaffected soil and taken from the Harmonized World Soil Database (HWSD; FAO and IIASA, 2009). This value is normally lower than the values prescribed for G1 and G2, and thus, the volatilization rate for the mineralized TAN is much lower than for urine. Similar to G2, $\Delta\theta = 0$ for G3.

### 2.4.3 Slurry

Manure slurries consist of animal feces, urine, washing water, bedding, spilt feeds and drinking water, and possibly rainwater (Sommer and Hutchings, 2001). The amount of suspended solids in slurry is measured by the dry matter (DM) content (g DM $g^{-1}$ slurry), which can vary due to different management practices from < 5 % up to about 20 %. Manure with a higher DM content can normally be handled as a solid (Lorimor et al., 2001).

15     Several studies (Sommer and Olesen, 1991; Vandre et al., 1997; Misselbrook et al., 2005b) have shown a positive correlation between the DM content and $NH_3$ volatilization. The suspended solids cause slurry to infiltrate soil slowly compared to water or urine, and consequently, large initial volatilization losses occur from broadcast slurry unless the slurry is mechanically incorporated to the soil (Pain et al., 1989; Van Der Molen et al., 1990b; Sommer et al., 2003; Meisinger and Jokela, 2000).

    To capture this effect, FANv2 includes the age class S0 representing soil patches with slurry partly remaining on the soil 20 surface. Conceptually, S0 corresponds to the first phase of ammonia volatilization in slurry as described by Sommer et al. (2003). The age extent $\Delta a$ of S0 defines the transition time to the second phase where the slurry can be considered incorporated into the soil matrix. The rate of infiltration depends on hydraulic properties of both the slurry and soil (Misselbrook et al., 2005b; Sommer et al., 2006). However, this level of detail is not feasible to simulate in a global model, as the uncertainties related to slurry composition and application methods are too large. While a major simplification, we assume that the infiltration occurs 25 in fixed time defined by the age extent of S0.

    The transport and transformation of N species in slurry is modeled following the overall approach described for soils in Section 2.2. However, due to the presence of slurry on the soil surface, the resistances in Eq. (8) for pool S0 need to be modified from those given in Eq. (9). To derive the resistance for the slurry-covered soil, we first consider the generic situation depicted in Fig. 4, where a fraction of the slurry remains on the surface while the infiltrated fraction forms a water-saturated 30 layer immediately below the soil surface. Instead of assuming a fixed layer thickness $\Delta z$, the fluxes for S0 are evaluated for the partly infiltrated slurry layer, and the layer thickness depends on infiltration and evaporation of the slurry. We do not track the distribution of TAN between the fractions above and below the surface, but do consider the two-layer structure when evaluating the resistances, as described below.





**Figure 4.** Schematic description and the corresponding resistance chart for modeling a partly infiltrated slurry layer. The resistance within the slurry remaining on surface is $R_{sl}$, the resistances within saturated soil are denoted by $R_{ss\uparrow}$ and $R_{ss\downarrow}$; other resistances are as in Fig. 1. Labels a) to d) refer to TAN concentrations: a) [NH$_3$ (g)] at atmospheric reference height ; b) [NH$_3$ (g)] and [TAN(aq)] at the slurry surface; c) [TAN (aq)] in the slurry and saturated soil; d) [TAN (aq)] and [NH$_3$ (g)] at the bottom of the saturated soil layer. Thicknesses of the slurry and soil layers are denoted by $d_{sl}$, $d_{sat}$ and $d_{1/2}$ as described in the text.





Following the resistance scheme in Fig. 4, the diffusive transport between the slurry-containing layer and surface is governed by the resistances $R_{ss\uparrow}$ and $R_{sl}$, which represent the aqueous phase diffusion in the saturated soil and in the slurry remaining on the soil surface. The downwards transport into soil is governed by the resistance $R_{ss\downarrow}$ (aqueous phase diffusion in saturated soil) and the parallel resistances $R_{aq\downarrow}$ and $R_{gas\downarrow}$, which represent aqueous and gaseous diffusion in the unsaturated soil layer 5 immediately below the saturated layer.

Denote the depth of the layer remaining on surface by $d_{sl}$ (m), and the depth of the saturated soil by $d_{sat}$. We assume that the volume of solid matter in slurry can be neglected. The total water volume (m$^3$ m$^{-2}$) within the two layers is therefore

$$W = d_{sl} + d_{sat}\theta_s. \tag{21}$$

As in Section 2.2, the resistances have the form $R = L_D/D$, where $D$ is diffusivity and $L_D$ denotes the length of the 10 diffusion path. Normally in FANv2 (Eq. 9), $L_D$ is defined as half of the geometric thickness of the layer, $\Delta z/2$. However, when the water content $\theta$ within the TAN-containing layer changes rapidly, the mean TAN concentration $N_{TAN}/W$ is a better approximation to the concentration at a depth $d_{1/2}$ such that

$$\int_0^{d_{1/2}} \theta(z)dz = \frac{1}{2}W, \tag{22}$$

where $z = 0$ corresponds the slurry surface. It is understood that $\theta = 1$ within the uninfiltrated slurry, and $d_{1/2}$ thus divides the 15 water volume $W$ into equal fractions. Fig. 4 assumes that $d_{sl} \leq W/2$, since, as shown below, this will always be the case in FANv2.

Following the notation defined above, the resistances in the slurry and the saturated layer are given as follows:

$$\begin{aligned} R_{sl} &= \min(W/2, d_{sl})/D_{NH4} \\ R_{ss\uparrow} &= \frac{\max(W/2 - d_{sl}, 0)}{\theta_s\xi(\theta_s)D_{NH4}} \\ R_{ss\downarrow} &= \frac{W}{2\theta_s\xi(\theta_s)D_{NH4}}, \end{aligned} \tag{23}$$

where the tortuosity factor $\xi$ is applied to the molecular diffusivity $D_{NH4}$ within soil but not to the slurry on surface ($R_{sl}$). The remaining resistances ($R_a$, $R_b$, $R_{gas\downarrow}$, and $R_{aq\downarrow}$), and subsequently the nitrogen fluxes, are evaluated as in Section 2.2.

The depths $d_{sl}$ and $d_{sat}$ need to be determined for evaluating the resistances. At $a = 0$, $d_{sl}$ equals the slurry depth $d_0$, and at $a = \Delta a$, $d_{sl} = 0$. We assume that at $a = \Delta a/2$, half of the initial volume has infiltrated into the soil, so that

$$25 \quad d_{sat}(a = \Delta a/2) = \frac{d_0}{2\varepsilon}. \tag{24}$$

The depth $d_{sl}$ is obtained by subtracting the evaporation loss over $\Delta a/2$ from the remaining half of $d_0$:

$$d_{sl}(a = \Delta a/2) = \max\left(\frac{d_0 - \Delta a q_e}{2}, 0\right), \tag{25}$$

which justifies the implicit assumption $d_{sl} \leq W/2$ in Fig. 4 and the Eqs. (23).





The evaporation rate $q_e$ (m s$^{-1}$) for slurry is evaluated as

$$q_e = \frac{\rho_{air}}{\rho_w} \frac{Q_{sat} - Q_{atm}}{R_a + R_b},$$ (26)

where $\rho_{air}$ and $\rho_w$ are the densities of air and water, $Q_{atm}$ is specific humidity at the atmospheric reference height, $Q_{sat}$ is the specific humidity at saturation, and $R_a$ and $R_b$ are as in Eq. (7). The initial slurry depth $d_0$ is given by the slurry application rate (m$^3$ m$^{-2}$), and in the global simulations we assume $d_0 = 5$ mm, equal to 50 m$^3$ ha$^{-1}$.

The infiltration time, as needed to define $\Delta a$ for S0, may be difficult to determine in practice, since a fraction of the water may be retained by the slurry solids for several days (Petersen and Andersen, 1996). Few observations are available to constrain $\Delta a$; Sommer and Jacobsen (1999) found 3 mm of pig slurry to infiltrate within 24 hours from application, while Misselbrook et al. (2005a) reported 20-30 % of cattle slurry and up 80 % of pig slurry to infiltrate within 1 hour. For the global simulations in this study, the infiltration time is set to 6 h, however, the effect of varying $\Delta a$ of S0 will be investigated in Section 2.6.

The other nitrogen fluxes from S0 are evaluated with only minor modifications compared to the other pools. The moisture flux $q_p$, required to evaluate the leaching flux (Eq. 10), is evaluated from the fraction of water in excess of $\Delta z\theta_s$ when the infiltration is complete,

$$q_p = \max\left(\frac{d_0 - \Delta a q_e - \Delta z\theta_s}{\Delta a}, 0\right).$$ (27)

where the cumulative evaporation is subtracted from the initial water volume. In addition, slurry remaining on the soil surface is exposed to enhanced runoff losses (Jarvis et al., 1987; Smith et al., 2001); this is simulated by evaluating the runoff flux $Q_r$ for S0 directly from the bulk concentration as $Q_r = q_r \times N_{\text{TAN}}/W$ instead of diagnosing the surface concentration as in Eq. (10).

The volatilization from fully infiltrated slurry is evaluated following the approach for manure on pastures. The remaining slurry age classes S1, S2 and S3 are defined similarly to the classes G1 through G3, with the difference that the evaporation loss $q_e\Delta a$ is subtracted from the initial water content $d_0$ in Eq. (19). The pH of slurry tends to increase after application due to volatilization of $CO_2$; a constant value 8.0 is used for pools S1 and S2 based on the data published by Sommer and Olesen (1991), Bussink et al. (1994) and Sherlock et al. (2002). Similar to G3, the pH for S3 is taken from the HWSD database.

### 2.4.4 Synthetic fertilizers

In FANv2, the nitrogen applied in synthetic fertilizers is split between urea N, nitrate N, and ammonium N. Urea N is simulated in the greatest detail due to its significance in the total $NH_3$ emissions (e.g. Bouwman et al., 2002). The ammonium N includes the $NH_4^+$-nitrogen in mineral fertilizers such as ammonium nitrate (AN), ammonium sulfate (AS), and ammonium phosphates. Volatilization losses from these fertilizers are normally low compared to urea (Whitehead and Raistrick, 1990; Sommer et al., 2004). An exception is ammonium bicarbonate (ABC), which is subject to similar volatilization losses as urea (Sommer et al., 2004; Bouwman et al., 2002). In FANv2, ABC is simply treated as urea. The nitrate N is not emitted as $NH_3$, and therefore not tracked further in this study.

Three TAN age classes (F1, F2 and F3) and two urea age classes (U1 and U2) are used to evaluate the volatilization losses for urea fertilizers (Fig. 3). The TAN formed from urea hydrolysis in each age urea class (U1 and U2) is added to the corresponding





TAN age class (F1 and F2). As in FANv1, the urea hydrolysis is modeled as a first order process with a time constant of 2.4 days (independent of soil temperature or moisture) adapted from the observations of Agehara and Warncke (2005).

As the fertilized patches age, TAN is transferred from F1 to F2 to F3 (Eq. 17), and urea N is transferred from U1 to U2. The transition between U1 and U2 matches the time scale for urea hydrolysis, and thus, little urea remains unhydrolyzed by the end

of U2. To avoid the need for a third urea pool, the remaining urea N in U2 is transferred directly to F3.

Since FAN assumes that fertilizers are applied in dry, granular form, no soil moisture perturbation is assumed for the fertilizer N pools. However, similar to urine, the formation of ammonium in urea hydrolysis increases the soil pH. The peak pH following urea application is often between 8 and 9 (Black et al., 1985; Whitehead and Raistrick, 1990; Sommer, 2013), and pHs of 7.0, 8.5 and 8.0 were chosen for F1, F2 and F3.

Other ammonium-based fertilizers do not form a strongly basic solution when applied on soil, which explains the smaller volatilization losses (Whitehead and Raistrick, 1990; Sommer et al., 2004), but also makes the losses more sensitive to the soil pH. In FANv2, this is modeled by assigning the ammonium N to the single TAN pool F4 with pH taken from the HWSD database. Although this neglects the variations in soil chemistry between different types of fertilizers, the effect on total $NH_3$ emissions is small due to the generally low volatilization losses. Since arable soils are frequently amended for pH, the pH for

F4 is restricted between 5.5 and 7.5, which includes the preferred range for most field crops (Spurway, 1941).

## 2.5 Agricultural systems

The final step in the global application of FAN is linking the process model with datasets describing global agricultural practices. For synthetic fertilizers, this task is simplified by using the fertilization rates included in the CLM5 surface dataset (Lawrence et al., 2016), which is the dataset used within the Coupled Model Intercomparison Project Phase 6 (CMIP6).

However, for manure, additional input data are needed to describe global patterns of livestock production, and additional parameterizations are needed to account for N losses in stored manure.

### 2.5.1 Livestock production systems and manure N

As described in Section 2.4.1, the volatilization losses differ between manure excreted on pastures and manure spread mechanically. To distribute the manure N between the two pathways we follow Seré et al. (1996), Bouwman et al. (2005) and

Beusen et al. (2008), and classify the global livestock into (i) pastoral and (ii) landless and mixed production systems. Pastoral systems are based on animal grazing in pastures, while in mixed and landless systems animals are typically confined to barns or feedlots. A significant fraction of $NH_3$ emissions in mixed/landless systems occurs during storage and handling of manure (Beusen et al., 2008).

Since the currently available datasets of global manure N excretion do not differentiate between production systems, we

compiled a new gridded dataset of yearly manure N excretion divided between these two systems. The global livestock density was obtained mainly from the Gridded Livestock of World (GLW) v2.01 dataset (Robinson et al., 2014), which includes the population densities of cattle, sheep, goats, pigs and poultry for the year 2010. The density of buffalo was taken from an earlier version of the same dataset with the base year 2005. The animal densities were converted to nitrogen excretion rates using the





coefficients recommended by IPCC (2006). The excretion coefficients depend on the animal and the region and are listed in the Suppl. Section 2.1. The total N excretion was 120 TgN for 2010, which is within 10% of the estimates of Zhang et al. (2017a; 129 TgN for 2010s), Potter et al. (2010; 128 TgN for 2007), and Beusen et al. (2008; 112 TgN for 2000). The N excretion was evaluated at 0.5 degree spatial resolution.

The manure N in each grid cell was divided between the pastoral and mixed/landless production systems as follows: all poultry and pig manure was assigned to mixed systems, while the ruminant manures (cattle, sheep, goats and buffalo) were split between the two systems using the FAO Global Livestock Production Systems dataset (version 5, Robinson et al. (2011)), which classifies the global land area into 12 livestock production categories. For each grid cell in the N excretion map, the fraction of ruminant manures attributed to pastoral systems was set equal to the fraction of the grid cell covered by grassland-

based (categories LGY, LGH, LGA and LGT) production systems. The remainder, about 75 % of the manure N globally, was assigned to the mixed/landless production systems.

In pastoral systems, all manure is assumed to be excreted in pastures while grazing, while in mixed/landless systems, ruminants are assumed to graze seasonally. The fraction $f_{grz}$ of ruminant manure excreted while grazing in mixed/landless production systems is evaluated dynamically as

$$
\quad f_{grz} = \begin{cases} f_{grz}^{max}, & T_{10}^{min} > +10° \text{ C} \\ 0, & \text{otherwise,} \end{cases} \tag{28}
$$

where $T_{10}^{min}$ is the 10-day running average of daily minimum temperature and $f_{grz}^{max} = 0.65$. The threshold temperature of $+10°$ C was used by Pinder et al. (2004) for modeling $NH_3$ emissions from dairy farms in the US; the temperature threshold also explains some of the geographical variations in grazing reported in European survey data (Klimont and Brink, 2004, Suppl. Section 3), although regional differences are large. For pigs and poultry, $f_{grz}$ is zero. Under these assumptions, about 60 %

of the manure N in mixed/landless systems was assigned to barns in the 2010–2015 simulations, which is a similar to that as estimated by Beusen et al. (2008).

The manure N remaining after subtracting the fraction $f_{grz}$ is excreted in animal housings (e.g., barns) and then stored prior to being spread. The volatilization losses of ammonia in animal housings and manure stores cannot be described as a soil process, and instead, we adopted a simpler mass flow scheme with empirical factors for the nitrogen losses based on the work

of Gyldenkærne et al. (2005). The same parameterization was used by Paulot et al. (2014).

We assume that manure is removed from storage and applied to soil at a constant rate. While this assumption neglects seasonal patterns in manure spreading, manure management practices generally depend on local regulations, availability of workforce, and other factors that remain difficult to represent in a global model. Our approach furthermore assumes that the ammonia emissions at a given time in housings are proportional to the TAN produced in housings, and that the amount of

ammonia volatilized from storage is proportional to the TAN entering storage.

Under these assumptions, the $NH_3$ emission from stores and housings is

$$
F_{\text{NH3}} = (1 - f_{grz}) F_{\text{TAN,excr}} \left( f_{\text{barn}} + f_{\text{store}} (1 - f_{\text{barn}}) \right), \tag{29}
$$



where $F_{\text{TAN,excr}}$ is the rate of TAN excretion, $f_{\text{barn}}$ is the fraction of TAN emitted in barns and $f_{\text{store}}$ is the fraction emitted in storage. The flux of TAN and organic N applied on soil is evaluated as

$$
\begin{aligned}
F_{\text{TAN,appl}} &= F_{\text{TAN,excr}} - F_{\text{NH3}} \\
F_{\text{org,appl}} &= F_{\text{org,excr}}
\end{aligned}
\tag{30}
$$

where $F_{\text{org,excr}}$ is the organic N excreted in barns. The loss of organic nitrogen from housings and during storage is assumed negligible.

The fractions $f_{\text{barn}}$ and $f_{\text{store}}$ are evaluated using the parameterization of Gyldenkærne et al. (2005). In the parameterization, the emissions from both housings and stores have the form

$$
f = CT^a V^b,
\tag{31}
$$

where $T$ is the temperature in barns or stores, $V$ is the effective ventilation rate, and $a$ and $b$ are constants. The values for $a$, $b$ and expressions of $T$ and $V$ are given by Gyldenkærne et al. (2005); the parameterization for naturally ventilated (open) barns are used for ruminants, and the values for mechanically ventilated (closed) barns are used for other livestock. The normalization constants $C$ are set to 0.03 open barns and 0.025 for closed barns and storage. The values were chosen to approximately reproduce the EMEP/EEA default emission factors (EEA, 2016) under European conditions.

Some of the stored manure may be used as fertilizer on croplands and some may be spread on grasslands. Volatilization losses from manure applied on crops and grasslands may differ due to differences in timing, vegetation cover, and method of manure application (Sommer and Hutchings, 2001). Since these details are not included in the model, for simplicity, our implementation applies all manure N on the natural soil column, which in the CLM subgrid structure includes the grasslands plant functional type. The current CLM version does not include an explicit representation of pastures, and consequently, the
natural soil column is also used to represent pastures in FANv2.

### 2.5.2 Synthetic fertilizers

The data for N fertilization rates in CLM5 do not specify the fertilizer type. Consequently, we used the country-level consumption statistics provided by the International Fertilizer Association (www.fertilizer.org) to disaggregate the total fertilization rates into fractions of nitrate, urea, and ammonium N as discussed in Section 2.4.4. The N in ammonium nitrate, calcium ammonium
nitrate and NPK compound fertilizers was split equally between ammonium and nitrate N; nitrogen solutions were assumed to contain 75 % of the nitrogen as ammonium and the remainder as nitrate. For China, the N reported under "other straight N" was attributed to ammonium bicarbonate following Bouwman et al. (2002) and, as described in Section 2.4.4, treated as urea.

The synthetic fertilizers are assumed to be applied exclusively on crop columns. The fertilization timing is determined by the CLM crop model, which applies the fertilizer according to the phenological stage of the crop, which in turn is parameterized
based on thresholds for growing degree days and air temperature (Badger and Dirmeyer, 2015; Levis et al., 2018). The fertilizer is applied during the 20 days following the leaf emergence. As discussed in Lawrence et al. (2018a), this choice is inherited





from earlier CLM versions which were found to overestimate denitrification loss. However, for the purposes of FANv2, the 20-day window provides a useful representation of the variability of fertilization timing within a grid cell.

NH$_3$ losses from fertilizers can be reduced substantially by placing or incorporating the fertilizer deeper into soil. Although mechanical incorporation is a standard practice for some crops and regions, the global fertilization practices are not well

characterized, and therefore we have not attempted to simulate the incorporation in detail. Instead, in FANv2 the effect of incorporation is simulated by reducing the fertilizer N available for volatilization by a constant 25 %. This assumes a typical 50 % reduction (Bouwman et al., 2002) applied to 50 % of the fertilizer N.

## 2.6 Model Evaluation

The simulated volatilization rates using FANv2 were compared with the results from 21 studies published in peer-reviewed

literature, with a total of 107 data points. The comparisons presented here are obtained by first performing a single-point CLM simulation for the time and site of the experiment, and subsequently using the simulated soil temperature, moisture and other parameters as the input for FAN. The CLM simulations were run in the satellite phenology mode and forced with the Global Soil Wetness Project Phase 3 (GSWP3) meteorological data set (http://hydro.iis.u-tokyo.ac.jp/GSWP3).

The experimental studies were selected to provide a geographically representative dataset covering volatilization from broad-

cast slurry applications, pastures, and from synthetic fertilizers. The experiments on pastures include both simulated urine patches and pastures with grazing livestock. For fertilizers, only experiments using surface application were included, and the 25 % reduction due to incorporation (Section 2.5.2) was therefore not used.

Preference was given for measurements based on micrometeorological techniques. However, the enclosure-based measurements of Vallis et al. (1982) were included due to the scarcity of volatilization observations in warm (subtropical) conditions.

Also, the measurements of Black et al. (1985) for ammonium sulfate, nitrate and phosphates, based on a similar enclosure method, were included in order to better represent fertilizers other than urea. For the measurements of Black et al. (1985), the total atmospheric resistance ($R_a + R_b$) was replaced with

$$R_{encl} = A/Q, \tag{32}$$

where $A$ is the soil area covered by the measurement chamber and $Q$ is the air flux (m$^3$s$^{-1}$) through the chamber. In the

measurements of Vallis et al. (1982), the flow rate was adjusted to follow the near-surface wind speed, and the $R_a$ and $R_b$ from CLM were used as for all other experiments. Whenever several replicate measurements were reported for the same time and site, only the averaged losses were compared to the model.

Generally, the experiments represented the local ambient conditions. The only exception was the experiment of Holcomb et al. (2011), which evaluates the effect of varying irrigation rates on NH$_3$ emissions. The irrigation was introduced to the CLM

simulations as precipitation; a separate CLM simulation was run for each irrigation experiment.

The simulated volatilization rates were unavoidably affected by the uncertainties in the variables simulated by the CLM and in the meteorological forcing. However, most of the experimental studies did not characterize the atmospheric and soil





conditions sufficiently to provide input for the FANv2 model. Furthermore, running FAN in combination with the CLM can be expected to give a more realistic assessment of the model's performance in its intended application.

## 2.7 Setup for global simulations

The global ammonia emissions analyzed below (Section 3.2) are based on a six-year simulation using the Community Earth
System Model (CESM), which couples the CLM with the Community Atmospheric Model CAM. As a part of CLM, the FANv2 ammonia emissions were evaluated interactively at each time step using the meteorological forcing from the atmospheric model. The simulation covered the years from 2010 to 2015. The year 2009 was run as spinup.

The model was run on a global longitude-latitude grid with a 2.5×1.9 degree spacing and a 30 minute coupling time step. The CAM version 5.4 was used, configured with the CAM4 physics package and run in the "offline" mode (Lamarque et al.,
2012) with the atmospheric dynamics prescribed by the MERRA reanalysis fields.

In addition to the 6-year simulations coupled to CAM, a set of 2-year (2010 and 2011) simulations was run evaluate the model's parameter sensitivity. To reduce the computational burden, these simulations were run in land-only mode with the atmospheric forcing given by the GSWP3 dataset.

## 3 Results

### 3.1 Evaluation against field measurements

A comparison of the modeled and measured volatilization rates (cumulative emission flux divided by the N input) is shown for grazed pastures in Fig. 5, panel a. The correlation between the model and measurements was $R = 0.57$. FANv2 captures the tendency towards higher volatilization at the warmer sites (Vallis et al., 1982; Laubach et al., 2012, 2013) reaching 30%, although one of the measurements of Vallis et al. (1982) is overestimated by the model. This measurement had the highest air
and soil temperature (up to +36° C) among the three measurements in Vallis et al. (1982), yet the lowest volatilization loss.

The measurements of Bussink (1992) and Jarvis et al. (1989) evaluate volatilization losses on pastures under varying N fertilization rates. Since the effect of fertilization prior to grazing cannot be simulated by FANv2, the replicates with different N fertilization were averaged when possible. However, this was not possible with most of the data in Bussink (1992), because the different treatments were applied at different times, which likely explains why the model did not reproduce most of the vari-
ability within the Bussink (1992) dataset. Nevertheless, the average losses taken over the Bussink (1992) data were reproduced reasonably well.

Similar to the pastures, in the comparison for synthetic fertilizers (Fig. 5b) the model has small average bias ($< 1$ % of the applied N), although the correlation between the model and the data is moderate ($R = 0.53$). The contrast between urea (blue markers) and and other fertilizers (purple markers) is captured. Also the decrease of volatilization with increasing irrigation
in the measurements of Holcomb et al. (2011), is reproduced, although the simulated volatilization is underestimated in the lightly irrigated treatments with the measured volatilization losses up to 60 %.



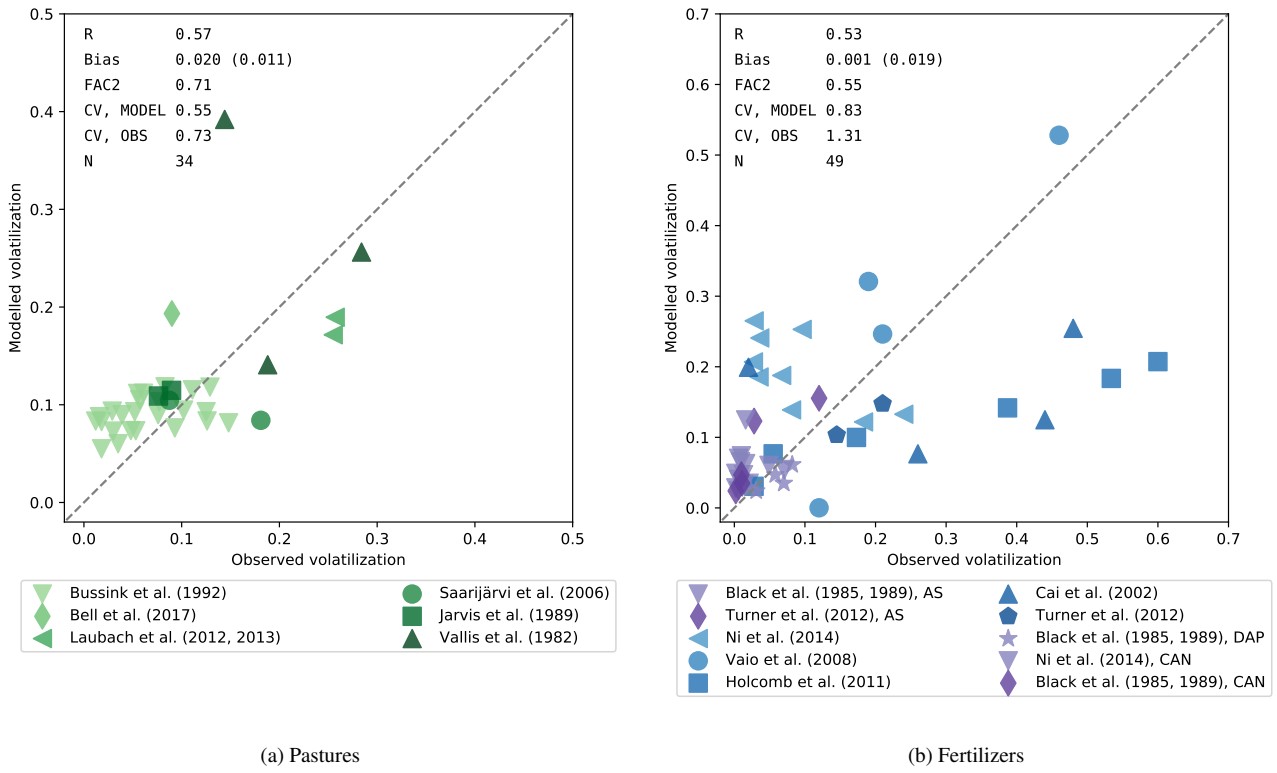

**Figure 5.** Modeled volatilization losses (relative to applied N) compared with field observations for urine patches (left) and for synthetic fertilizers (right). The data for fertilizers include urea, shown with blue markers, and diammonium phosphate (DAP), ammonium sulfate (AS), calcium ammonium nitrate (CAN), shown with purple markers. Abbreviations used for statistical indicators: R – Pearson's correlation coefficient, FAC2 – fraction of values within factor of 2, CV – Coefficient of Variation, N – number of points.

Finally, Fig. 6 compares the simulated volatilization losses with observations for surface-applied slurry. In panel a, the model was run with a constant application rate of 50 m$^3$ ha$^{-1}$ and infiltration time ($\Delta a$ for S0, Section 2.4.3) $\tau_{\mathrm{infl}} = 6$ h, which are the default values chosen for the global simulations. In this configuration, the model captures the average volatilization losses, which are higher than for urea or pastures, but the observations of Spirig et al. (2010) and Sintermann et al. (2011) are strongly
5  overestimated, and the model is not significantly correlated with observations ($R = 0.26$, $p = 0.2$). The modest agreement with the observations suggests that a significant fraction of the variation might not be related to the variations in ambient conditions.

The experiments of Spirig et al. (2010) and Sintermann et al. (2011) were carried out using mixtures of cattle and swine slurries with DM contents mostly between 1 and 3 %, while the other studies include slurries with up to 12 % DM. Similarly, the application rate varied from 30 m$^3$ ha$^{-1}$ up to 100 m$^3$ ha$^{-1}$ (3–10 mm) in the various studies. While the application rate is
10  an input parameter for FANv2 as noted in Section 2.4.3, the DM content is not directly related to any of the model parameters. However, the DM content is related to the infiltration rate of slurry (Misselbrook et al., 2005b; Sommer et al., 2006), and by



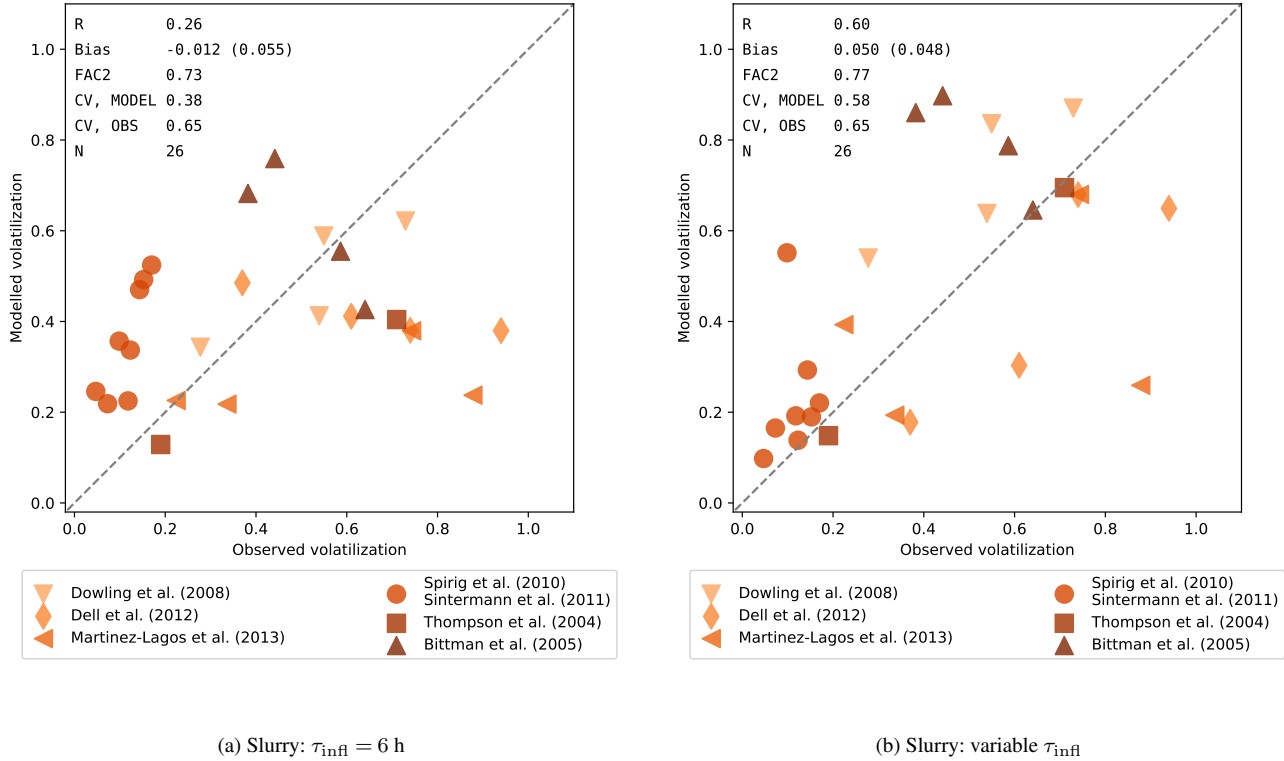

(a) Slurry: $\tau_{\mathrm{infl}} = 6\,\mathrm{h}$          (b) Slurry: variable $\tau_{\mathrm{infl}}$

**Figure 6.** Modeled volatilization losses compared with field observations for slurry. Left: results with 6 hour infiltration time and no adjustment for application rate. Right: results using reported application rates and infiltration times adjusted based on dry matter content. Abbreviations used are as in Fig. 5.

assuming a simple relation between the DM content and the infiltration rate, it was possible to tune the model to provide a better match to the observations.

The comparison in Fig. 6b is obtained by setting the initial slurry depth $d_0$ equal to the reported application rate, and setting the infiltration time $\tau_{\mathrm{infl}} = d_0/q_s$, where the slurry infiltration rate $q_s$ decreases linearly from 5 mm/h at DM $\leq 1$ % to 6 mm/day
5  at DM $\geq 4$ %. This adjustment effectively causes the model to treat the dilute slurries similarly to urine. When adjusted for the DM content and application rate, the modeled volatilization losses are significantly correlated with the observations ($R = 0.60$, $p < 0.01$). Thus, for the datasets included in this study, the variations of DM and application rate indeed appear to explain a considerable fraction of the variation in the observations. Especially the data of Spirig et al. (2010), Sintermann et al. (2011) and Thompson and Meisinger (2004) are well reproduced after adjusting for slurry characteristics. The slurry characteristics
10  also appear to explain the variations between measurements of Dell et al. (2012), although the model tends to underestimate the volatilization loss in these measurements.





Parameters like DM content and application rate are not available for global simulations. Similar to the case for slurry, the evaluations for pastures and fertilizers are likely to be affected by insufficiently known parameters, such as the urine volume $d_0$ and the layer thickness $\Delta z$, which for fertilizers can be interpreted as the depth of application. The model sensitivity to these parameters is discussed with regard to the global simulations in Section 3.3. However, globally, even more substantial

variations may arise from different application methods. When applied on arable land, both fertilizers and manure are frequently incorporated mechanically, which results in a large reduction of volatilization losses (Sommer, 2013; Pan et al., 2016). Further uncertainty arises from various types of manure, such as deep litter or farmyard manure, which are currently not implemented in the model. With sufficient observational data, these practices could also be included to the model.

If the data from all experiments are pooled together, and the default parameters are assumed for slurry, the modeled volatiliza-

tion loss was within factor of 2 of the observed in 64 % of the cases, and the model reproduces the observed losses with $R = 0.65$ and mean bias of $\sim$1 % of the applied N. Thus, the model captures variations in volatilization losses associated with different forms of nitrogen application with small overall bias. The modeled coefficient of variation was for all categories lower than observed, as could be expected in absence of site-specific adaptations.

### 3.2   Global NH$_3$ emissions

The simulated global agricultural ammonia emissions for 2010-2015 were 47 Tg N/year consisting of 36 Tg N from manure and 11 Tg N from use of synthetic fertilizers. The manure emissions include 12 Tg N from grazed pastures, 18 Tg N from barns and stores, and 6.3 Tg N from manure application. The fertilizer emissions consist of 8.1 Tg N from urea and ammonium bicarbonate and 2.9 Tg N from all other synthetic fertilizers.

Geographically, the highest emissions for urea and other fertilizers (Fig. 7) occur in China and India. The highest emissions

from manure (Fig. 8) partly coincide with those from fertilizers, however, significant emissions occur also in regions such as Equatorial Africa and South America where fertilizer usage is low. The highest relative volatilization losses for both fertilizers and manure (Figs. 9 and 10) are associated with regions with warm and often arid climates. The losses in equatorial regions are relatively low due to high precipitation, with the exception of losses in barns and manure stores, where the emissions are assumed to be unaffected by rain.

The volatilization losses are shown as fractions of the N inputs in Table 1. The losses from manure application are shown with respect to both applied TAN and total (organic and ammoniacal) nitrogen. Since the higher losses in housings and storage result in lower TAN fractions in the applied manure, normalizing the losses by the TAN applied reveals a much higher regional variability than is apparent from the losses calculated with respect to total N. It should be noted that the fraction normalized by the applied TAN is not exactly equal to the real fraction of TAN volatilized, since some of the emission actually originates

from the organic fraction (Section 2.2).

The predominant process limiting the volatilization loss was diffusion and leaching of TAN deeper into the soil; for both manure and fertilizers, about 55 % of the input N was removed from the FANv2 pools via this pathway (data not shown). The role of nitrification was generally smaller: about 12 (15) % of the manure (fertilizer) N was nitrified within FANv2. The loss





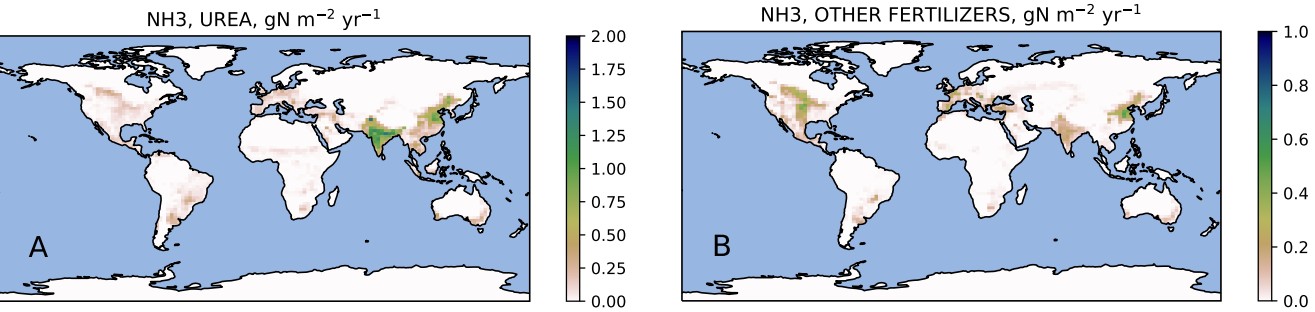

**Figure 7.** Simulated ammonia emission (gN m$^{-2}$ yr$^{-1}$) from urea (left) and other synthetic fertilizers (right) averaged over 2010–2015. Note different color scales.

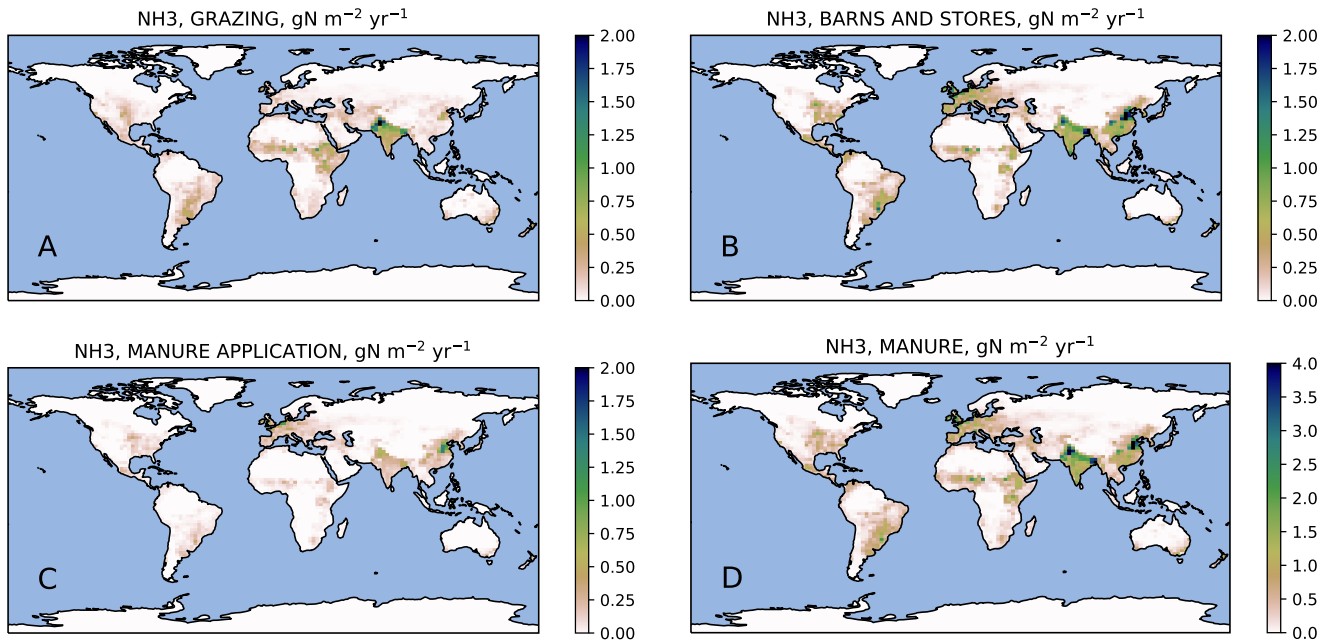

**Figure 8.** Simulated ammonia emission (gN m$^{-2}$ yr$^{-1}$) from manure: pastures (panel a) barns and storage (panel b), manure application and total from manure (panels c and d), averaged over 2010–2015. Note the different color scale for panel d.



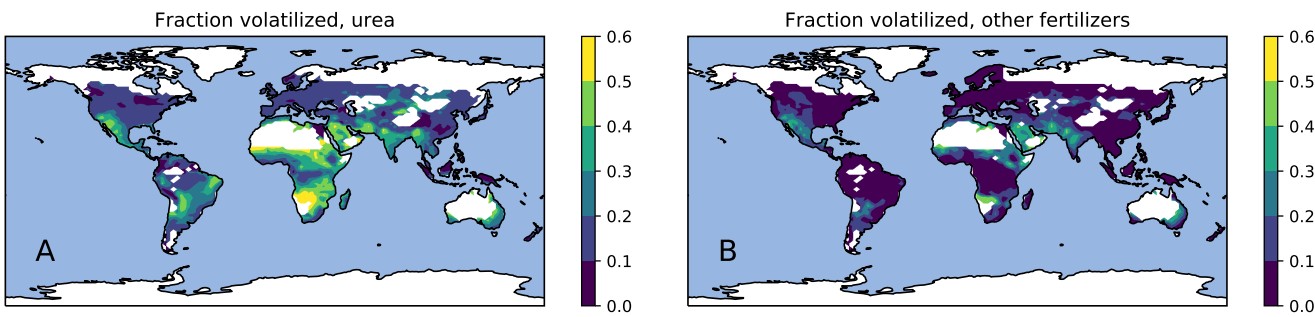

**Figure 9.** Fraction of fertilizer N lost due to volatilization, average for 2010–2015: urea (left), other synthetic fertilizers (right).

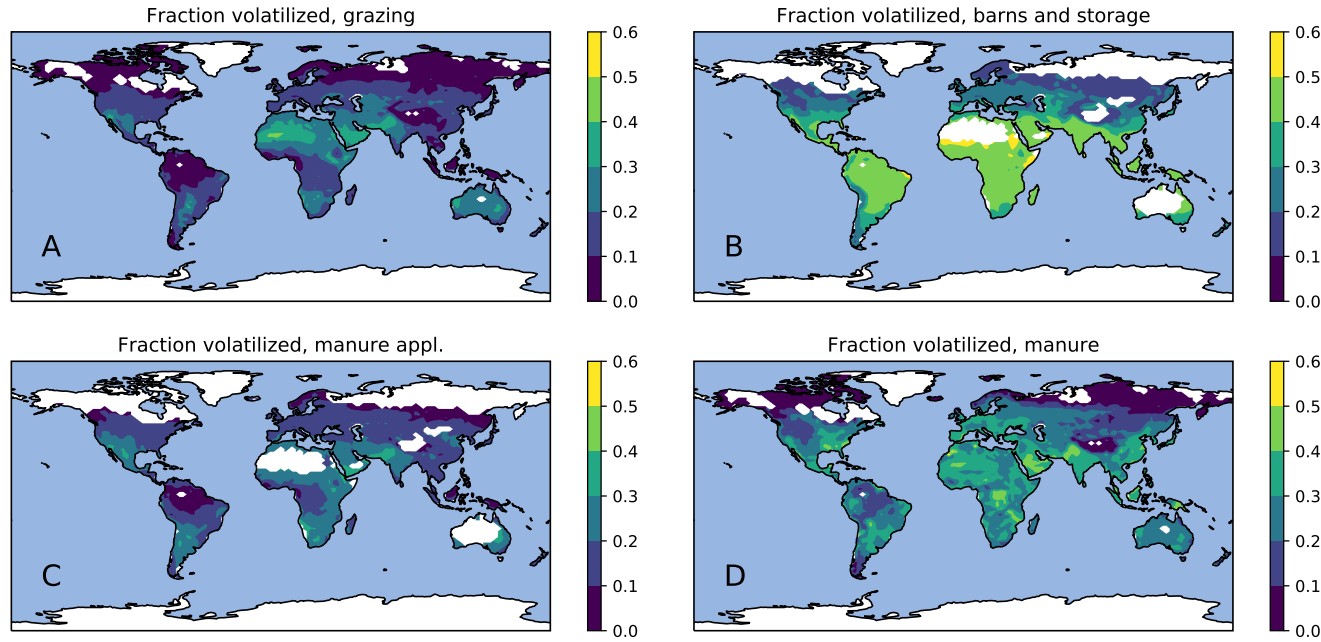

**Figure 10.** Fraction of manure N lost due to volatilization, average for 2010–2015: Panel a – grazing, panel b – barns and storage, panel c – manure application, panel d – all manure.





**Table 1.** Global and regional averages of volatilization losses in agricultural activities. The losses are given as fractions of total (organic and inorganic) manure or fertilizer nitrogen unless stated otherwise. The total volatilization loss for manure includes emissions from all individual processes normalized by the total manure N produced in the region. The average loss for synthetic fertilizers consists of the emissions urea and other fertilizers normalized by the total fertilizer N applied.

| Region | Barns, stores[1] | Grazing[2] | Manure Spreading, of total N[3] | Spreading, of TAN[4] | Synth. fertilizer Total | Urea | Others | Average |
|---|---|---|---|---|---|---|---|---|
| Africa | 0.44 | 0.21 | 0.22 | 0.77 | 0.31 | 0.30 | 0.11 | 0.20 |
| Asia except China and India | 0.36 | 0.21 | 0.19 | 0.49 | 0.33 | 0.18 | 0.08 | 0.14 |
| China | 0.29 | 0.12 | 0.17 | 0.39 | 0.32 | 0.14 | 0.06 | 0.11 |
| Europe | 0.25 | 0.13 | 0.15 | 0.33 | 0.30 | 0.15 | 0.04 | 0.06 |
| India | 0.44 | 0.21 | 0.23 | 0.81 | 0.35 | 0.27 | 0.19 | 0.26 |
| Latin America | 0.42 | 0.14 | 0.17 | 0.55 | 0.27 | 0.23 | 0.10 | 0.17 |
| Oceania | 0.33 | 0.15 | 0.19 | 0.47 | 0.23 | 0.25 | 0.15 | 0.22 |
| US and Canada | 0.28 | 0.15 | 0.15 | 0.34 | 0.27 | 0.14 | 0.07 | 0.09 |
| World | 0.34 | 0.18 | 0.18 | 0.46 | 0.30 | 0.19 | 0.07 | 0.13 |

[1] As fraction of N excreted in barns; [2] As fraction of N excreted while grazing; [3] As fraction of N remaining after losses in storage and housings; [4] As fraction of TAN remaining after losses in storage and housings.

due to surface runoff as $NH_4^+$ or urea was 1.7 % for fertilizer and 0.8 % for manure N. Note that the runoff loss evaluated by FANv2 does not include subsurface leaching, or any runoff or leaching of nitrate N.

Figure 11 compares the FANv2 emissions regionally and globally with the version 4.3.2 of the EDGAR emission inventory (Crippa et al., 2018). Globally, the FANv2 emissions (47 TgN y$^{-1}$) are about 15 % greater than the EDGAR emissions (41

TgN y$^{-1}$ from the agricultural sector). The regional comparison shows that the difference is largely due to the emissions in Africa, India, and Latin America, while for China, the EDGAR emissions are about 50 % higher than FANv2. For Europe and North America, FANv2 and EDGAR are in good agreement.

The EDGAR emissions are split into two reporting categories: "manure management", which includes emissions from animal housings and stored manure, and "agricultural soils" which includes emissions from soils (from both manure or synthetic

fertilizer application and grazing). As seen in Fig. 11, the split between the categories is similar between FANv2 and EDGAR for Europe and North America, where the total emissions are also similar. Conversely, the regions where FANv2 and EDGAR differ most also have large differences in the contributions from the two emission categories. In particular, a significant fraction of manure in Africa, India and Latin America is attributed to mixed production systems in FANv2. This leads to large emissions from housings and manure stores in FANv2, while in EDGAR, manure management contributes only minimally to the

emissions in these regions.





**Table 2.** Simulated $NH_3$ emissions by region averaged for years 2010–2015 and compared with existing inventories. The total emission is equal to manure management + agricultural soils, or total manure + synthetic fertilizer. For FANv2, the manure management emissions are equal to the emissions from barns and storage.

| Region | Inventory | Base year | NH3 emission, TgN/yr | | | | |
|---|---|---|---|---|---|---|---|
| | | | Total | Manure manag. | Agr. soils | Manure, total | Synth. fertilizer |
| China | *Range*[1] | 2008-2010 | 6.6–12.3 | 1.4–2.0 | 7.7–9.3 | 4.1–7.1 | 2.4–5.2 |
| | EDGAR4.3 | 2010 | 11.3 | 2 | 9.3 | | |
| | FANv2 | 2010-2015 | 7.5 | 3.1 | 4.3 | 5.2 | 2.3 |
| Europe | EMEP[2] | 2010 | 3.9 | 2.3 | 1.6 | | |
| | EDGAR4.3 | 2010 | 4.8 | 2.1 | 2.7 | | |
| | FANv2 | 2010-2015 | 4.8 | 2.5 | 2.3 | 4.0 | 0.7 |
| India | *Range*[3] | 2003-2010 | 4.8–5.9 | 0.3–1.4 | 3.9–5.0 | 1.5–3.1 | 2.2–3.3 |
| | EDGAR4.3 | 2010 | 5.4 | 0.3 | 5 | | |
| | FANv2 | 2010-2015 | 7.4 | 2.3 | 5.1 | 4.7 | 2.7 |
| North America | EC[4]/EPA[5] | 2010/2011 | 3.3 | | | 2.2 | 1.1 |
| | EDGAR4.3 | 2010 | 3.6 | 1.2 | 2.4 | | |
| | FANv2 | 2010-2015 | 3.5 | 1.1 | 2.4 | 2.2 | 1.3 |
| World | B2008[6] | 2000 | 32 | 9.2 | 23 | 21 | 11 |
| | EDGAR4.3 | 2010 | 41 | 9 | 32 | | |
| | MASAGE_NH3[7] | 2005-2008 | 34 | | | 24 | 9.4 |
| | FANv1[8] | 2000 | 33 | | | 21 | 12 |
| | FANv2 | 2010 | 47 | 18 | 29 | 36 | 11 |

[1] Kang et al. (2016); Xu et al. (2018); Kurokawa et al. (2013); Zhang et al. (2018, 2017b)

[2] EMEP/CEIP 2018, http://www.ceip.at/webdab_emepdatabase/emissions_emepmodels

[3] Aneja et al. (2012); Kurokawa et al. (2013); Xu et al. (2018)

[4] https://pollution-waste.canada.ca/air-emission-inventory

[5] https://www.epa.gov/air-emissions-inventories/2011-national-emissions-inventory-nei-data

[6] Beusen et al. (2008); [7] Paulot et al. (2014); [8] Riddick et al. (2016)





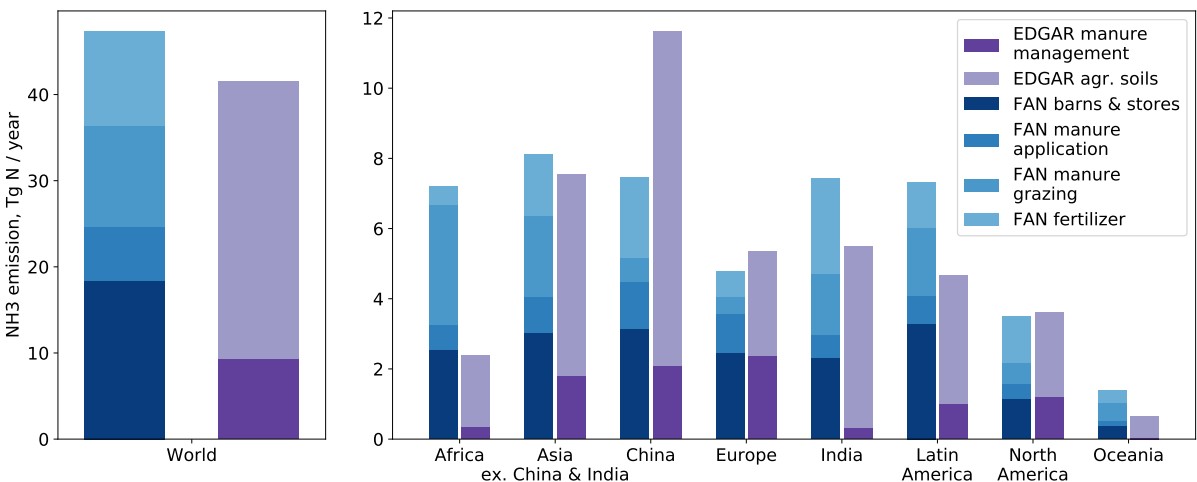

**Figure 11.** Global and regional ammonia emissions from agricultural sources in FANv2 and EDGARv4.3.2, Tg N / year. The EDGAR manure management emissions correspond to barns and stores in FANv2.

Table 2 compares FANv2 with additional regional and global emission inventories. FANv2 and EDGAR agree within 10% with the national emission inventories for the US and Canada (EPA/EC); also the split between manure and synthetic fertilizers is similar in FANv2 and the EPA/EC inventories. For Europe, the FANv2 emissions are in agreement with EDGAR, but 23 % higher than those reported in the EMEP emission inventory, mainly due to larger emissions in the "agricultural soils" category.

The ammonia emissions in China have been studied intensively, and only the studies with the base year 2008 or later are included in Table 2. The FANv2 emissions (7.5 Tg N) are within the range of the published estimates, albeit on the lower end, mainly due to the lower emissions from fertilizer application. In contrast, the FANv2 emissions for India are about 25– 50 % higher than in previously published global and regional inventories, mainly due to increased emissions from manure management and grazing.

We are not aware of regional emission inventories covering all of South and Central America, but national inventories have been compiled for Chile (Muñoz et al., 2016, livestock only) and Argentina (Castesana et al., 2018). For Chile, the estimate of Muñoz et al. (2016) of 57 Gg N (69 Gg $NH_3$) from livestock for 2013 is comparable to the FANv2-simulated emission of 70 Gg N for 2010–2015. For Argentina, Castesana et al. (2018) estimated annual emissions of 139 Gg N (169 Gg $NH_3$) from manure and 119 Gg N (145 Gg $NH_3$) from mineral fertilizers in 2010–2012 – far less than the corresponding FAN emissions of

760 and 260 Gg N. The higher fertilizer emissions in Argentina simulated by FANv2 are largely explained by higher fertilizer use in the CLM dataset (1400 Gg N compared to 400–900 Gg N reported by Castesana et al. (2018)). The fertilizer use of Castesana et al. (2018) are consistent with the IFA statistics for 2010-2015. However, the difference in manure $NH_3$ emissions appears to be caused by a much higher emission factor implied by the FANv2 simulation.





In Africa, the FANv2 emissions from grazing alone (3.4 Tg N) exceed the total $NH_3$ emission (2.4 Tg N) reported in the EDGAR inventory. Comprehensive regional $NH_3$ emission inventories for Africa are not available. However, assuming a fixed 30 % volatilization loss, Delon et al. (2010) estimated 1.5 ±0.8 TgN/yr emitted within the Sahel region, which is consistent with the FANv2 emission of 1.2 TgN/yr for the same region.

Compared to FANv1, the total emissions in FANv2 are about 40 % higher. This difference is mainly caused by the volatilization loss from manure, which is 30 % of manure N in FANv2, but only 17 % in FANv1. As a consequence, the total emissions in FANv1 were relatively low especially for China (5.2 TgN) and Europe (1.9 Tg N), and for these regions, the emissions simulated by FANv2 (Table 2) are closer to the available regional inventories. It is difficult to attribute the difference between FANv1 and FANv2 to specific model features, because FANv1 does not differentiate between emissions in storage and hous-

ing, manure application, and grazing. However, as discussed in Section 2.2, a key difference between FANv1 and FANv2 is that the emission fluxes in FANv1 include a 60 % canopy capture fraction, while FANv2 does not currently assume any canopy capture.

Also losses for fertilizers differ between the versions. FANv1 treated all fertilizers as urea, resulting in a higher total volatilization rate (19 %) for synthetic fertilizers than FANv2 (13 %). However, the mean volatilization rate for urea in FANv2

is 19 %, which is similar to FANv1.

### 3.3 Sensitivity to model parameters

As a process model FANv2 uses a number of poorly constrained parameters. A set of 2-year simulations were run to investigate the model's sensitivity to its parameters as described in Suppl. Section 4. The sensitivity experiments used a different meteorological forcing than the main simulations (GSWP3 instead of the CAM simulation), which increased the global emissions for

2010–2011 by 2 %. On global level, the model therefore appears fairly robust with regard to the meteorological input.

Overall, the model was also relatively insensitive (<10 % change in global emission, $\sim$ 0.1–0.2 % per % change in parameter) to parameters affecting any individual process, such as slurry infiltration, urea hydrolysis or timing of fertilization (Suppl. Table 2). The parameters with a more systematic effect, and therefore higher sensitivity, included thickness of the model layer ($\Delta z$), the adsorption parameter $K_d$, the manure TAN fraction $f_{\mathrm{TAN}}$ and the maximum grazing fraction ($f_{grz}^{max}$,

Section 2.5.1). A 10 % change in the TAN fraction or the grazing fraction changes the global manure $NH_3$ emission by 3–4 and 8 %, respectively.

The sensitivity for both $\Delta z$ and $K_d$ was higher for fertilizers than for manure. Varying $K_d$ between 0 and 10 times the default changed the manure emissions by $-29 - +11$ %, while for fertilizers, the range was $-55 - +30$ %. For manure, varying $\Delta z$ (by default 2 cm) between 4 and 1 cm changed the emission by $-19 - +6$ %. However, for fertilizers, doubling the $\Delta z$ to 4

cm reduced the emissions by 52 %, while halving $\Delta z$ increased the emissions by 41 %. This response is roughly comparable with the observed effect of incorporating urea into soil as evaluated in the literature survey of Rochette et al. (2013); in the polynomial fit of Rochette et al. (2013) increasing the incorporation depth from 2 to 4 cm reduces emissions by $\sim$40 %, while reducing the depth to 1 cm increases the emission by $\sim$23 %.





## 4 Discussion

Agricultural ammonia emissions are determined both by agricultural activity and environmental conditions. Both these of these aspects of ammonia emissions have been incorporated into the process model FANv2, which embedded within the CESM simulates agricultural ammonia emissions globally. An advantage of using a process model is that the simulated emissions

respond to changes in both agricultural practices and environmental conditions in a physically realistic manner. The model can simulate how the emissions from various agricultural processes respond to variations in meteorological forcing on timescales from hours to centuries, and thus, FANv2 can capture the changes in emissions occurring as a result of anomalous weather events as well as due to changes in climate.

Global datasets have been used to quantify some regional agricultural practices in FANv2. For example, regional nitrogen

excretion rates and synthetic fertilizer usage and type have been included. Regional agricultural practices also reflect variations in local meteorology, and these variations can be parameterized within an Earth system model. In FANv2 we use the local meteorological conditions to parameterize the timing of fertilizer application and the extent to which domestic animals excrete manure on pastures. The advantage of these meteorological-dependent parameterizations are that impacts of climate change on these aspects of agricultural management are built implicitly into the model; the disadvantage is that these meteorological

parameterizations do not always conform to regional agricultural practices.

Some regional aspects of agriculture remain simplified in the model. In particular, livestock manure is treated everywhere as a slurry, which is likely to lead to uncertainties in developing countries where handling manure as slurry is uncommon, although the emissions from manure applications constitute only 10–15 % of the simulated total emissions outside Europe, North America and China. Nevertheless, with globally available information, FANv2 could be configured to include further

details on regional agricultural practices and their changes.

Distinct from FANv2, most other available ammonia emission inventories make use of empirical factors relating ammonia emissions to livestock N excretion and fertilizer usage. The disadvantage of this approach is that it does not fully take into account variations in the environmental parameters that partially govern the ammonia emissions. On the other hand, many emission inventories take regional and local agricultural practices into account. Over North America and Europe, the FANv2

NH$_3$ emissions (3.5 and and 4.8 Tg N yr$^{-1}$, respectively) are within ∼25 % of established emission inventories (Table 2). This is perhaps not very surprising, as some of the simulated processes, such as handling manure as slurry, reflect primarily North American and European agricultural practices. Furthermore, some of the model parameters, such as average losses from animal housings and manure storage, were explicitly chosen to reproduce emission factors used in Europe. In contrast, for most other parts of the world, the FANv2 simulations differ from previous emission estimates.

In China, the FANv2 emissions (7.5 Tg N/yr) are lower than the majority of recent global and regional estimates. The difference appears to be caused by the relatively low simulated emission losses from fertilizer in FANv2. FANv2 predicts that the fractions of urea or other fertilizers volatilized in China is similar to those in Europe or North America, in contrast to regional studies such as Zhang et al. (2017b) and Zhang et al. (2018) which use higher emission factors as compiled from





empirical studies. The emission factor implied by FANv2 for China (Table 1) is 11 % for total fertilizer N and 14 % for urea. In comparison, the average emission factor for fertilizer N was 18.1 % in Zhang et al. (2017b), and 16.6 % in Zhang et al. (2018).

It is difficult to isolate any particular factor that causes FANv2 to underestimate the Chinese emission factors compared to the other inventories. Based on a sensitivity analysis (Section 3.3 and Suppl. Section 4), differences in soil adsorption, soil

pH and fertilizer incorporation could explain some of the discrepancy. The explicit consideration of rice paddies could also be potentially important; FANv2 and the CLM do not explicitly simulate rice paddies, even though the processes controlling $NH_3$ volatilization in paddies are likely different from those in upland crops. For Chinese rice paddies, Wang et al. (2018) report an average emission factor of about 18 % for urea, which is higher than the factor calculated from FANv2, but not significantly higher than the overall emission factors used by Zhang et al. (2017b) and Zhang et al. (2018). This suggests that the omission

of rice paddies in FANv2 is not solely responsible for the discrepancy in emissions.

In contrast to China, in India FANv2 predicts higher $NH_3$ emissions (7.3 Tg N/yr) than previous inventories. In FANv2, the total volatilization loss of manure N is 35 % in India, which is ∼50 % higher than the 23 % emission factor used by Xu et al. (2018). The 27 % loss simulated for urea is also higher than the 19 % loss evaluated by Xu et al. (2018), but nevertheless similar to the the 25 % emission factor used by Aneja et al. (2012). While the emissions from agricultural soils are similar

in India in FANv2 and EDGAR, the emissions from manure management in FANv2 are seven times as high as than those in EDGAR. A higher fraction of grazing in FANv2 would act to reduce the overall emissions, since the volatilization loss (Table 1) is 21 % for manure N excreted on pastures in FANv2, while the joint loss for barns, stores and manure application is 57 % of the N excreted in barns. In FANv2 about 8 Tg out of the total 13 Tg of manure N is excreted on pastures in India, which is limited by the maximum grazing fraction for mixed production systems in FANv2. Increasing the grazing fraction could

reduce the simulated $NH_3$ emission by up to 1.8 Tg N, which would result in emissions similar to those in EDGAR.

It is unclear if a grazing fraction this high would be realistic, given that Mohini et al. (2016, cited by Prasad et al. (2017)) report that in India the fraction of manure input on grazed fields is 35–45 % depending on the type of livestock. However, Prasad et al. (2017) note that a similar fraction of manure is used for fuel, and thus removed from the agricultural system. This is not taken into account in FANv2, but since the manure N in fuel is likely to be mainly in organic form with a low potential

for ammonia volatilization, the reduction in $NH_3$ emissions would likely be lower than the fraction of N in fuel.

The fractional volatilization losses (Table 1) were generally more variable regionally for synthetic fertilizers than for manure. The volatilization loss from synthetic fertilizers ranges between 6 % in Europe to 26 % in India. The different climates in Europe and India result in a variation between ∼15–27 % in the volatilization loss for urea. However, in Europe ∼20 % of fertilizer N was applied as urea, and ∼30 % as nitrate according to the IFA fertilizer consumption data, while in India, the corresponding

fractions were ∼85 % as urea and < 2 % as nitrate. Thus, the climate-driven difference in volatilization rates is amplified by the strong contrast in usage of different fertilizers.

For manure, the overall fractional volatilization in FANv2 ranges from 23 to 35 %. Manure emissions are split between the emissions from grazing and the emissions from manure housing, storage and the subsequent spreading. Large regional differences are apparent in the emissions from manure spreading, ranging from approximately 30 % of the TAN applied under

cooler conditions (North America and Europe) to ∼75 % of the TAN applied in the warmest regions (Africa and India, Table 1).





However, these regional variations in emissions are compensated by regional variations in the extent of grazing which ranges from $\sim 30$ % in Europe (not shown) up to 75 % in Africa. The higher fraction of grazing in Africa compared to Europe is due to longer grazing season, smaller proportion of non-grazing livestock (pigs, poultry), and larger proportion of pastoral livestock systems. Thus in cooler regions animals spend more time in housing, where the overall emissions factors due to housing, storage and spreading are relatively high, while in warmer regions animals spend more time grazing where the emission factors are relatively low. The combination of regional practices and meteorological conditions acts to mute regional differences in manure $NH_3$ emissions – contrary to the fertilizer $NH_3$ emissions.

Globally, in FANv2, the simulated volatilization loss for fertilizers (13 %) is similar to the central estimate (14 %) given by Beusen et al. (2008). However, the average volatilization loss for manure ($\sim 30$ %) is about 60 % higher than the 19 % loss in the study of Beusen et al. (2008), where the manure emissions were based on the emission factors in Bouwman et al. (1997), hereafter B97. The difference stems largely from assumptions regarding geographical differences in $NH_3$ volatilization from manure, which in B97 is represented by two aggregated regions.

In both FANv2 and B97 the overall volatilization loss from grazing animals is significantly less than the losses from barns and manure storage and spreading. For the region I countries (the developed countries), the 36 % volatilization loss from manure N excreted by cattle, pigs and poultry in barns in B97 is similar to that in FANv2 (38 % in the same region, including losses in housing, storage and spreading). The higher total volatilization loss (28 % in FANv2 vs. 21 % in B97) for manure N in the region I countries is therefore explained by the higher volatilization rate for manure on pastures (15 % vs. 7 %) and the lower proportion of N excreted on pastures in Europe (28 % in FANv2 vs. 51 % for Region I in B97). In North America, the fraction of N excreted on pastures is 50 % in FANv2, in agreement with B97.

In the region II countries (the developing countries), the fraction of N excreted on pastures in FANv2 (59 %) agrees with B97 (62 %). The corresponding volatilization rate is higher in FANv2 (18 %) than in B97 (13 %), but this difference alone is not enough to explain the difference in the total manure N volatilization. However, B97 assumed that the effect of higher average temperatures in animal housings in region II is compensated by lower TAN content in manure, and therefore used the same emission factors for manure N excreted in barns for regions I and II. This resulted in a 21 % overall volatilization loss for both regions. In contrast, the TAN fraction in FANv2 is fixed at 60 % and therefore does not compensate for the higher volatilization rate (50 % for barns, storage and spreading in region II). Together with the higher volatilization loss for grazing, this explains the higher volatilization loss of (31 % vs. 21 %) in areas corresponding to the region II in B97.

Although only present-day emissions were evaluated in this study, the simulated geographical variation in volatilization rates can be used to derive a crude estimate of how the $NH_3$ emission respond to changes in mean temperature and rainfall. This evaluation implicitly includes how agricultural management practices change across temperature and rainfall. We first categorize the model cells by yearly rainfall, then for each category linearly regress with temperature (Suppl. Section 5). This yields an overall temperature sensitivity of ammonia emissions of $\sim 3$ % $K^{-1}$, which is in the lower end of the 3–7 % range given by Sutton et al. (2013) for fertilizer and slurry application. However, the FANv2 estimate also includes the effect of increased grazing and earlier planting dates in warmer climates, which reduces the effective temperature sensitivity. Especially for synthetic fertilizers, the temperature sensitivity varies with annual rainfall, and generally decreases towards more humid





climates. This suggests that assessments of $NH_3$ emissions in future climate should consider patterns of precipitation in addition to the mean temperature.

## 5   Conclusions

We have described a process-based model for evaluating ammonia volatilization losses from synthetic fertilizers and livestock
wastes, evaluated the model with experimental data, and presented simulated global ammonia emissions obtained by coupling the process model into the land component of the Community Earth System Model (CESM). Compared to the initial version (Riddick et al., 2016), FANv2 improves the representation of soil processes and fertilization and manure management practices. The model evaluates ammonia emissions interactively with the simulated atmosphere, and responds to both short and long-term variations in the meteorological forcing. The impacts of different agricultural practices and their changes have also been
incorporated into the model. Thus, FANv2, embedded within an Earth system model, represents a platform with which to investigate how ammonia emissions change as agricultural practices and climate change as we head into the future.

Comparison with data from 21 volatilization experiments shows that FANv2 successfully reproduces variations in volatilization between different types of manures and fertilizers. The model also reproduced variations stemming from environmental factors, albeit with a higher uncertainty. The mean model bias was small both within the categories and over the whole data
set.

Based on global simulations for 2010–2015, we estimate an average yearly $NH_3$ emission of 47 Tg N consisting of 36 Tg from manure and 11 Tg from synthetic fertilizers. The volatilization losses correspond to 30 % of excreted manure N and 13 % of applied fertilizer N. The simulated total emission is 30–40 % larger than previous estimates for 2010, which is mainly caused by higher simulated emissions from livestock wastes in Africa, India and Latin America. The simulated emissions are
in agreement with regional inventories for Europe and North America, and within the range of previous estimates for China.

In a preliminary estimate, based on a statistical regression on geographical variations of the simulation $NH_3$ volatilization, the global $NH_3$ was estimated to increase by ~3 % for a 1 K increase in global mean temperature. This sensitivity includes the effect of increasing grazing and earlier crop planting dates in warmer climates.

The global $NH_3$ emissions and their geographic distribution was sensitive to assumptions regarding livestock N excretion
and the prevalence of grazing in mixed livestock production systems. Differences in these assumptions may explain some of the differences between FANv2 and earlier emission inventories.

The simulated emission were coupled to the CAM-Chem chemistry-climate model, which allows further evaluation of the emission estimates via comparison with atmospheric observations. This path will be taken in a subsequent paper (Vira et al., 2019, in prep.), which compares the atmospheric simulation with datasets of ammonia and ammonium concentrations and wet
depositions.





*Code and data availability.* The Community Earth System Model, including the Community Land Model (CLM) is available at www.cesm.ucar.edu. The modified version of CLM used in this paper is available at https://doi.org/10.5281/zenodo.3373497. The full modified version of CESM, including changes to CAM and the coupler interface, requires access to the CAM development repository which can be granted by UCAR upon agreement with the terms of use. Potential users are suggested to contact the authors for the latest version of the code.

5   The model input data prepared for this study are included in the supplement. The simulated monthly NH3 emissions are included in the supplement; other model outputs are available on request.

*Author contributions.* JV, PH and JM formulated the model. JV and WRW implemented the model in CLM. JV designed and performed the simulation experiments and analyzed the output with contributions from PH. JV and PH prepared the manuscript with contributions from JM and WRW.

10  *Competing interests.* The authors declare no competing interests.

*Acknowledgements.* This work was funded in part by the Department of Energy (#DE-SC0016361) and in part supported by the National Center for Atmospheric Research, which is a major facility sponsored by the National Science Foundation under Cooperative Agreement No. 1852977. Computing resources (doi:10.5065/D6RX99HX) were provided by the Climate Simulation Laboratory at NCAR's Computational and Information Systems Laboratory, sponsored by the National Science Foundation and other agencies. WRW was supported by the US Department of Agriculture NIFA Award number 2015-67003-23485, the Environmental Protection Agency's National Center for Environmental Assessment, through an Interagency Agreement with the National Science Foundation and the National Center for Atmospheric Research (#DW-49-92447301-0). The authors thank Susan Cheng, Susan Riha and Jinyun Tang for valuable discussions, and Marje Prank for comments on the manuscript.





## Appendix A:  Model equations and parameters

| Variable | Equation |
| --- | --- |

**TAN**

$$\frac{dN_{TAN}}{dt} = I_{TAN} - F_{atm} \tag{A1}$$
$$+ \quad k_U N_U + k_A N_A + k_R N_R - k_N N_{TAN} - k_m N_{TAN}$$
$$- \quad F_{\downarrow}^{TAN} - Q_r^{TAN} - Q_p^{TAN}$$

**Urea**

$$\frac{dN_{\text{urea}}}{dt} = I_{\text{urea}} \tag{A2}$$
$$- \quad k_U N_U - k_m N_{\text{urea}}$$
$$- \quad F_{\downarrow}^{\text{urea}} - Q_r^{\text{urea}} - Q_p^{\text{urea}}$$

$N_A$ and $N_R$

$$\frac{dN_{A,R}}{dt} = I_{N_A,N_R} - k_{A,R}N_{A,R} - k_m N_{A,R} \tag{A3}$$

Diagnostic concentrations

| Quantity | Unit | Description |
| --- | --- | --- |
| [urea (aq,srf)] | gN m$^{-3}$ | Dissolved urea at surface |
| [TAN (aq, srf)] | gN m$^{-3}$ | Dissolved TAN at surface |

$$[\text{urea (aq,srf)}] = \frac{N_{\text{urea}}}{\Delta z \theta (R_{aq\uparrow} q_r + 1)} \tag{A4}$$

$$[\text{TAN (aq, srf)}] = \frac{R_{ab}\chi_s(R_{gas\uparrow} + R_{aq\uparrow}K_{\text{NH3}}) + R_{gas\uparrow}R_{aq\uparrow}\chi_a(K_d + \theta + x_0 - x_1 - x_2)}{R_{aq\uparrow}\varepsilon K_{\text{NH3}}^2 x_3 + K_d x_4 + K_d x_5 + \theta x_4 + \theta x_5 + x_0 x_5 - x_1 x_4 - x_1 x_5 - x_2 x_4 - x_2 x_5}$$

where

$$\chi_s = N_{TAN}/\Delta z \tag{A5}$$
$$\chi_a = [\text{NH3 (g, atm)}]$$
$$R_{ab} = R_a + R_b$$
$$x_0 = \varepsilon K_{\text{NH3}}$$
$$x_1 = \varepsilon K_d$$
$$x_2 = K_d\theta$$
$$x_3 = R_{ab} + R_{gas\uparrow}$$
$$x_4 = R_{aq\uparrow}K_{\text{NH3}}x_3$$
$$x_5 = R_{ab}R_{gas\uparrow}(R_{aq\uparrow}q_r + 1)$$





| Symbol | Unit | Description | Equation | |
|---|---|---|---|---|
| **Diffusion** | | | | |
| $\xi_{gas}$ | | Tortuosity for gas phase diffusion | $\xi_{gas}(\theta) = \dfrac{(\theta - \theta_s)^{\frac{10}{3}}}{\theta_s^2}$ | (A6) |
| | | | Millington and Quirk (1961) | |
| $\xi_{aq}$ | | Tortuosity for aqueous-phase diffusion | $\xi_{aq}(\theta) = \dfrac{\theta^{\frac{10}{3}}}{\theta_s^2}$ | (A7) |
| | | | Millington and Quirk (1961) | |
| $D_{\mathrm{NH4}}^{\mathrm{aq}}$ | $\mathrm{m^2 s^{-1}}$ | Molecular diffusivity of $NH_4^+$ in water | $D_{\mathrm{NH4}} = 9.8 \cdot 10^{-10} \cdot 1.03^{T_g - 273.15}$ | (A8) |
| | | | Van Der Molen et al. (1990a) | |
| $D_{\mathrm{NH3}}^{\mathrm{g}}$ | $\mathrm{m^2 s^{-1}}$ | Molecular diffusivity of $NH_3$ in air | | |
| | | | $D_{\mathrm{NH3}}^{\mathrm{g}} = \dfrac{0.001 \times T_g^{1.75}(1/M_{air} + 1/M_{\mathrm{NH3}})^{1/2}}{p[(\Sigma_{air}v_i)^{1/3} + (\Sigma_{\mathrm{NH3}}v_i)^{1/3}]^2},$ | (A9) |
| | | | where $M_{air} = 29.0$, $M_{\mathrm{NH3}} = 17.0$, $\Sigma_{air}v_i = 20.1$, $\Sigma_{\mathrm{NH3}}v_i = 14.9$ and $p = 1.0$ (Fuller et al., 1966). | |
| **Equilibrium constants** | | | | |
| $K_{\mathrm{NH4}}$ | $\mathrm{mol\,l^{-1}}$ | $NH_3 + H_2O \rightleftharpoons NH4^+ + OH$ | $K_{\mathrm{NH4}} = 5.67 \times 10^{-10} e^{-6286(1/T_g - 1/T_{\mathrm{ref}})},$ | (A10) |
| | | | (Sutton et al., 1994), where $T_{\mathrm{ref}} = 298.15$ K. | |
| $K_H$ | | $NH_3\,(aq) \rightleftharpoons NH_3\,(g)$ | $K_H = (4.59\ K^{-1})T_g e^{4092(1/T_g - 1/T_{\mathrm{ref}})},$ | (A11) |
| | | | (Sutton et al., 1994), where $T_{\mathrm{ref}}$ is as in Eq. (A10). | |
| $K_{\mathrm{NH3}}$ | | [NH3 (g)] / [TAN (aq)] | $K_{\mathrm{NH3}} = \dfrac{1}{K_H + [\mathrm{H^+}]/K_{\mathrm{NH4}}}$ | (A12) |
| $K_d$ | | [TAN (s)] / [TAN (aq)] | $K_d = 1.0$ | (A13) |





| Symbol | Unit | Description | Equation |
|--------|------|-------------|----------|

**Decomposition rates**

$k_{\mathrm{NO3}}$    $\mathrm{s}^{-1}$    Nitrification rate

$$k_{\mathrm{NO3}} = \frac{2r_{\max}}{1/\Sigma(T_g) + 1/\Pi(\theta_g)}, \tag{A14}$$

where $r_{\max} = 1.16 \cdot 10^{-6} \ \mathrm{s}^{-1}$ (Riddick et al., 2016) and the gravimetric soil moisture

$$\theta_g = \frac{\theta \rho_{\mathrm{water}}}{(1 - \theta_s)\rho_{\mathrm{soil}}}.$$

$\Sigma(T_g)$      Temperature response function

$$\Sigma(T_g) = \left(\frac{t_{\max} - T_g}{t_{\max} - t_{\mathrm{opt}}}\right)^{a_\Sigma} \exp\left(a_\Sigma\left(\frac{T_g - t_{\mathrm{opt}}}{t_{\max} - t_{\mathrm{opt}}}\right)\right), \tag{A15}$$

where $t_{\mathrm{opt}} = 301$ K and $t_{\max} = 313$ K, and $a_\Sigma = 2.4$ (Stange and Neue, 2009).

$\Pi(\theta)$      Moisture response function

$$\Pi(\theta_g) = 1 - e^{-(\theta_g/m_{\mathrm{crit}})^b}, \tag{A16}$$

where $m_{\mathrm{crit}} = 0.12$ (Stange and Neue, 2009) and $\theta_g$ is as in Eq. (A14).

$k_a, k_r$    $\mathrm{s}^{-1}$    Decomposition rate for $N_A$ and $N_R$

$$k_{a,r} = B_{a,r} T_R(T_g) P_\psi(\psi), \tag{A17}$$

where $B_a = 8.94 \times 10^{-7} \ \mathrm{s}^{-1}$ and $B_r = 6.38 \times 10^{-8} \ \mathrm{s}^{-1}$ (Gilmour et al., 2003; Vigil and Kissel, 1995).

$T_R$      Temperature dependence of $k_a$ and $k_b$

$$T_R(T_g) = t_{r1} \exp(t_{r2}(T_g - 273.15)), \tag{A18}$$

where $t_{r1} = 0.0106$ and $t_{r2} = 0.12979 \ \mathrm{K}^{-1}$ (Vigil and Kissel, 1995).





| Symbol | Unit | Description | Equation |
|--------|------|-------------|----------|
| Decomposition rates, continued | | | |
| $P_\psi$ | | Soil moisture dependency $k_a$ and $k_b$ | |
| | | | $$P_\psi(\psi) = \frac{\log(\psi_{\min}/\psi)}{\log(\psi_{\min}/\psi_{\max})}, \qquad \text{(A19)}$$ where $\psi$ is the soil matric potential (MPa), $\psi_{\min} = -2.5$ MPa and $\psi_{\max} = -0.002$ MPa (Lawrence et al., 2018a). |
| $k_u$ | s$^{-1}$ | Decomposition rate for $N_U$ | $$k_u = 4.83 \times 10^{-6} \qquad \text{(A20)}$$ Agehara and Warncke (2005) |





## Appendix B: Measurement data used in model evaluation

| Reference | Type | Region |
|---|---|---|
| Bell et al. (2017) | Pasture | France |
| Bussink (1992) | Pasture | Netherlands |
| Jarvis et al. (1989) | Pasture | Great Britain |
| Laubach et al. (2012) | Pasture | New Zealand |
| Laubach et al. (2013) | Pasture | New Zealand |
| Saarijärvi et al. (2006) | Pasture | Finland |
| Vallis et al. (1982) | Pasture | Queensland, Australia |
| Bittman et al. (2005) | Slurry | British Columbia, Canada |
| Dowling et al. (2008) | Slurry | Ireland |
| Dell et al. (2012) | Slurry | Pennsylvania, USA |
| Martínez-Lagos et al. (2013) | Slurry | Chile |
| Spirig et al. (2010) | Slurry | Switzerland |
| Sintermann et al. (2011) | Slurry | Switzerland |
| Thompson and Meisinger (2004) | Slurry | Maryland, USA |
| Black et al. (1985) | DAP, AS, CAN | New Zealand |
| Black et al. (1989) | AS | New Zealand |
| Cai et al. (2002) | Urea | Henan, China |
| Holcomb et al. (2011) | Urea | Oregon, USA |
| Ni et al. (2014) | Urea, CAN | Germany |
| Turner et al. (2010) | Urea, AS | Victoria, Australia |
| Vaio et al. (2008) | Urea | Georgia, USA |

The synthetic fertilizers are abbreviated as AS (ammonium sulfate), CAN (calcium ammonium nitrate) and DAP (diammonium phosphate). The measurements of Bittman et al. (2005) and Dowling et al. (2008) were extracted from the ALFAM2
5  database (Hafner et al., 2018).





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
