# Peer review of "An improved mechanistic model for ammonia volatilization in Earth system models: Flow of Agricultural Nitrogen, version 2 (FANv2)"

_Geoscientific Model Development, 2019_

## Referee Comment (RC1) · Anonymous Referee #1 · 23 Sep 2019

This manuscript aims to update the process model FAN to version 2 (FANv2) through including more detailed information on both physical and agricultural processes and use FANv2 to simulate ammonia volatilization from agricultural systems globally. Through modeling both fertilizer and manure application in agricultural lands, the authors provide updated numbers compared with that reported by FAN. Although this study is not innovative and with some flaws, it does make a significant contribution to the simulation of agricultural NH3 emission.

Main comments: 1. Why did not you consider plants (crop or grasses) in your model? If you have ignored the interaction between crops, does it mean NH3 emission from

different crops is only related to the fertilizer or manure application and its soil property? For different crops, do you use different parameters to calibrate your NH3 emission module? In page 2, line 29: 'FANv2 makes use of the interactive crop model included in CLM (Lawrence et al., 2019; Lombardozzi et al., 2019) to determine the timing and amounts of fertilization appropriate to each crop.' Please at least indicate the total amount of annual N fertilizer application in the main text. Also, does the timing of fertilization you mention vary with crop type? Only consider one-time fertilization? In page 3, line 20: 'Each crop type, in contrast, corresponds to a single soil column. Since the primary input variables in FANv2 are related to N cycling and hydrology in the CLM, FANv2 is introduced into the CLM sub-grid structure on the soil column level.' I cannot understand what you mean. Also, it seems that your NH3 emission module is kind of isolated even though you use existing parameters in the CLM (e.g., soil column, the timing of fertilization, etc.). In page 4, line 1: 'FANv2 does currently not interact with the base nitrogen cycle in CLM, and the effects of microbial immobilization and plant uptake of fertilizer and manure N are therefore not simulated.' If there is no interaction with crops when you apply fertilizer or manure, how can your FANv2 realistically simulate agricultural nitrogen flows? In page 9, line 1: 'FANv1 applies a 60 % reduction to the emissions to account for the NH3 captured by plant canopy; this reduction is included in the flux shown in Fig. 2a.' I expect how the updated FANv2 deals with N flows when considering the interaction with plant dynamics.

2. The authors miss the description of input datasets. In page 2, line 33: 'In this study FANv2 is run globally within the CLM for the six-year period 2010–2015 to simulate the present-day NH3 emissions, which are then compared with existing global and regional inventories.' Moreover, you mention NH3 emission is sensitive to climate change, but I did not find any information on climate conditions. The authors should add one more table listing the input datasets that were used to drive FANv2. Meanwhile, you need to mention that N input data are only for one-year. Even though you only cover 6 years' simulation, the temporal changes in N input amount, especially at the regional scale, may introduce the uncertainty when you compare your results with existing global and

regional inventories.

More specific questions: 1. Page 2, line 5: Here, you should at least cite articles to support your statement. Moreover, in Xu et al. (2019), they have considered the interaction between atmosphere and soils to investigate the effect of meteorological forcing.

2. Page 2, line 10: Please cite Xu et al. (2019) as well since they also emphasized the importance of environmental factors.

3. Page 4, line 16: Why 2.4 days? Any explanation for this assumption? You need to add citation here.

4. Page 5, line 21: Actually, it is not reasonable to use a constant Kd to deal with ammonium adsorption in soils.

5. Page 11: What is the time step for your simulation? Is it in second (s)? How did you get hourly, daily, monthly, and annual NH3 emission from s? Is there any assumption behind it? By the way, you mentioned an assumed 2.4 days for urea hydrolysis. How does the model deal with the 2.4 days reaction since your simulation is at second time step, right?

6. Page 18, line 6: Do you mean FANv2? I am confused with this sentence 'Since FAN assumes that fertilizers are applied in dry, granular form, no soil moisture perturbation is assumed for the fertilizer N pools. However, similar to urine, the formation of ammonium in urea hydrolysis increases the soil pH.' If there is no soil moisture effect, how does soil water content affect the NH3 volatilization?

7. Page 20, line 31: One time only for each crop?

8. Page 21, line 12: 'The CLM simulations were run in the satellite phenology mode and forced with the Global Soil Wetness Project Phase 3 (GSWP3) meteorological data set (http://hydro.iis.u-tokyo.ac.jp/GSWP3).' The model comparison project covers the time period of 1850-2010; however, this study focuses on the period of 2010-2015. Did

you use climate data in 2010 to run the model for 2010-2015? Please clarify it.

9. Page 23, figure 5: Did you check the effect of different levels of fertilizer or manure? Why did you only include one site in China? What is the unit in this figure?

10. Page 27, figure 9: Can you explain why the fraction volatilized in urea is highest in Southern Africa as well as Northern Africa?

11. Page 27, figure 10a: Why is it shown the highest fraction of NH3 volatilization in the desert Africa? As I see, it seems there is little manure there in Figure 1 in the supplement. Why is it shown the highest fraction of NH3 volatilization in Africa and South America in figure 10b? Moreover, it seems the fraction is fixed for most of land areas. Please explain it.

12. Page 30, figure 11: Please indicate the time period. Also, please indicate the version of EDGAR that you used for comparison and add the citation here.

13. Page 30, line 9: 'mainly due to increased emissions from manure management and grazing'. Please check it carefully. Is there any data to support it?

14. Page 30, line 13: 'Also losses for fertilizers differ between the versions. FANv1 treated all fertilizers as urea, resulting in a higher total volatilization rate (19 %) for synthetic fertilizers than FANv2 (13 %). However, the mean volatilization rate for urea in FANv2 is 19 %, which is similar to FANv1.' You described how FANv1 deals with no crop interaction. In page 9, line 1: 'FANv1 applies a 60 % reduction to the emissions to account for the NH3 captured by plant canopy; this reduction is included in the flux shown in Fig. 2a.' Did you consider this percentage (60%) when you compared your current results with that from FANv1? If no, why yours are similar to that in FANv1 for NH3 emission from urea? Please clarify.

15. Page 31, line 10: Your higher emission is from manure. The canopy capture may be not the explanation.

16. Page 32, line 5-8: In this study, you simulated NH3 volatilization only covering 6

years. Climate change refers to a long-term change in weather. Thus, the author needs to put this into your future research even though you have conducted the sensitivity analysis to show how temperature and soil moisture can affect the NH3 volatilization processes. Please reorganize this sentence to describe what your model has done at the current stage.

17. Page 32, line 12: You mentioned that you used the local meteorological conditions to parameterize your model. Where did these data come from? It is necessary to add this information in Appendix B. Please add more detailed information on each site, such as longitude/latitude, soil type, crop type, annual temperature, annual precipitation, etc.

18. Page 32, line 17: 'In particular, livestock manure is treated everywhere as a slurry, which is likely to lead to uncertainties in developing countries where handling manure as slurry is uncommon'. Any citation?

19. Page 33, line 3: 'It is difficult to isolate any particular factor that causes FANv2 to underestimate the Chinese emission factors compared to the other inventories'. First, you only include one site in China, which may cause the underestimate. Second, did you check fertilizer and manure data and compare them with the previous study? It is possible that the amount of N inputs to Chinese agricultural soils differ a lot in different studies. Please at least state this here.

20. Page 33, line 10: Omission of rice paddies may lead to the underestimate. It has been reported that 90% of rice is cultivated in East, South, and Southeast Asia. Xu et al. (2019) claims that rice cultivation has become the largest source for NH3 emission due to its high rate of fertilization and warm temperature. Please at least discuss the important role of rice here.

21. Page 35, line 8-9: 'The model evaluates ammonia emissions interactively with the simulated atmosphere, and responds to both short and long-term variations in the meteorological forcing.' Throughout the main text, I did not see any long-term estimates of NH3 volatilization. You only reported the average annual amount for the period of

2010-2015. Also, the author did not report intra-annual estimates of NH3 volatilization. Please carefully state 'both short and long-term variations in the meteorological forcing' since these were not fully reported in the current study. By the way, I am curious about the intra-annual results. Is it possible to present your intra-annual results in the main text?

22. Page 35, line 23: It is odd to mention 'This sensitivity includes the effect of increasing grazing and earlier crop planting dates in warmer climates.' What is the relationship between this sentence and the sensitivity analysis? You did not consider these factors in the sensitivity test, right? Please reorganize it.

Reference Xu, R., Tian, H., Pan, S., Prior, S.A., Feng, Y., Batchelor, W.D., Chen, J., Yang, J. (2019) Global ammonia emissions from synthetic nitrogen fertilizer applications in agricultural systems: Empirical and process‐based estimates and uncertainty. Global Change Biology 25, 314-326.

---

## Short Comment (SC1) · 4 Mar 2020

This paper describes a process based model FAN (Flow of Agricultural Nitrogen), which evaluates the NH3 emissions interactively within an Earth system model for use in the Community Earth System Model (CESM). The paper is an advancement to FANv1. However, FANv2 largely inherits its parameterizations for chemical and biological processes from FANv1 but adds a more detailed description of the processes which transport TAN within the soil. The updated version (FANv2) includes 5 more detailed treatment of both physical and agricultural processes, which allows the model to differentiate between the volatilization losses from animal housings, manure storage, grazed

pastures, and from application of manure and different types of mineral fertilizers. The FANv2 model is connected to the interactive crop model within the land component of CESM, which determines the amount and timings of fertilizer applications for major types of crops. The model is first evaluated at local scale against experimental data for various types of fertilizers and manure, and subsequently run globally to evaluate present-day NH3 emissions.

The authors i.e. Vira, Hess, Melkonian and Wieder are all highly regarded. They have made a very good attempt at advancing the science of estimating the science of ammonia emissions from agricultural soils. Moreover, they have compared their model output to measured values; and extended the analysis to global prediction. I believe the manuscript needs to address my concerns before I will recommend it for publication. Major Comments: 1. The manuscript is very lengthy; and a lot of the modeling framework background should be shortened. 2. The assumptions made in the manuscript need to be explained and or justified. For example pg 3, line 30 "the soil below the topmost layer is treated as a sink——and all the N transport below 2 cm layer is assumed to be permanently unavailable—-" needs to be justified. 3. Most of the References are old i.e. prior to 2015. More recent references i.e. 2018 and 2019 should be provided and their results discussed. 4. Pg 4, line 2 The role of microbial activity is not simulated. This is a very important component of the N biogeochemical cycling and needs to be addressed. 5. Pg 2, line 33 "In this study FANv2 is run globally within the CLM for the six-year period 2010–2015 to simulate the present-day NH3 emissions, which are then compared with existing global and regional inventories". This is fundamentally incorrect since ammonia emissions and thus ammonia concentrations calculated for 2015 will be different from present-day i.e. 2019 or 2020. 6. Pg 4, Eq 2. The rates are all first order. This assumption needs to both be explained and justified. 7. Pg 5, Line 2. "The diffusion and leaching fluxes are not evaluated for the available and resistant organic N". This reviewer does not understand what is being suggested. 8. Pg. 23, Figure 5 b. Model results and observations do not agree well. This needs to be explained in detail.

---

## Author Response (AR1)

We thank the reviewers for the detailed feedback on the manuscript. We have revised the manuscript to address the concerns raised, and below we give point by point response to each comment. In the revised manuscript we have followed the recommendation of Reviewer 2 and moved technical content from Section 2 to the supplement.

In addition, we found a coding error affecting the infiltration time for slurry (Section 2.4.3). In effect, the infiltration time was approximately 12 hours instead of 6 h as reported in the initial submission. We have repeated the simulations and evaluation with the 12 h infiltration time resulting in an overall increase of global NH3 emissions by 0.6 % from 47.3 Tg N to 47.6 Tg N.

*Reviewer comments are shown in italic.*

**Response to reviewer 1**

*Why did not you consider plants (crop or grasses) in your model? If you have ignored the interaction between crops, does it mean NH3 emission from different crops is only related to the fertilizer or manure application and its soil property? For different crops, do you use different parameters to calibrate your NH3 emission module?*

The NH3 emission is determined by the fertilizer/manure type, application and timing and the simulated soil processes (Section 2.1). The fertilizer type, timing and amount is determined from external datasets and by the CESM crop model. The FANv2 model itself does not use crop-specific parameters. However, the FANv2 model is designed to interface with the CLM-crop model which does consider specific crop properties.

FANv2 is not intended as a replacement of the N cycling simulated by the CLM, but rather as a module for evaluating the short-term volatilization loss of NH3. In this first set of simulations we run FANv2 in a partially coupled mode (as stated in 2.1) where FANv2 is coupled to the atmospheric model but not to the CLM biogeochemical cycling of N. A full coupling with the model biogeochemistry and the crop model coupling will be described in subsequent papers. However, at present, coupling FANv2 to the atmospheric model allows using atmospheric measurements of ammonia and ammonium to provide strong observational constraints on the NH3 emissions.

We have revised the manuscript to describe the scope of these simulations more clearly throughout the Introduction.

*Please at least indicate the total amount of annual N fertilizer application in the main text.*

The amounts of total fertilizer N (79 Tg N per year in 2010 increasing to 87 Tg N per year in 2015) applied have been added to Section 2.5.2.

*Also, does the timing of fertilization you mention vary with crop type? Only consider one-time fertilization?*

We assume fertilization only occurs once per growing season. Fertilization in CLM5 occurs during the leaf emergence phenological stage of the crop model and lasts for 20 days (Section 2.5.2). The phenological stage is determined dynamically and depends on the crop type. Fertilization rates are prescribed by crop type, country, and year based on the Land-Use Harmonization 2 dataset (Hurtt et al., 2011; Lawrence et al. 2019).

*'Each crop type, in contrast, corresponds to a single soil column. Since the primary input variables in FANv2 are related to N cycling and hydrology in the CLM, FANv2 is introduced into the CLM sub-grid structure on the soil column level.' I cannot understand what you mean.*

The "soil column" has a specific meaning in the structure of CLM: for example, a grid cell could be covered by a mix of different crop types each represented by a separate "soil column" corresponding to a given fraction of surface area.

We have revised the paragraph as follows:

"The CLM uses a hierarchical structure to represent sub-grid scale heterogeneity in land cover, and in particular this allows biogeochemical processes to be simulated independently for each crop type within a given grid cell. FANv2 conforms to the CLM sub-grid structure and evaluates the NH3 volatilization separately for the grasslands and each managed crop present in a grid cell."

*If there is no interaction with crops when you apply fertilizer or manure, how can your FANv2 realistically simulate agricultural nitrogen flows? I expect how the updated FANv2 deals with N flows when considering the interaction with plant dynamics.*

Among the limitations in FANv1 were the inability to differentiate between different types of fertilizers, and between different forms of manure. We prioritized these details in the development of FANv2, since both factors are known to be critical in determining the global variations in NH3 losses in crop and livestock systems.

As stated above, we are currently running FANv2 in a partially coupled mode where it is coupled to the atmosphere but not to the CLM soil nitrogen cycle, nor is it explicitly coupled to the crops through the CLM crop model. We intend to simulate the biogeochemical interactions in a future version by leveraging the processes existing in the CLM. In the fully coupled model we expect NH3 emissions to be similar to those analyzed in the current study. Since the CLM currently applies the fertilizers in a single application during an early phenological stage, both root uptake of $NH4^+$ and leaf uptake of NH3 during volatilization are expected to be small due to the small root density and leaf area. As discussed in the manuscript, the interaction with vegetation might be more significant for pastures and other grasslands. Even then, the effect of root uptake would remain small, since the plant uptake of N is slow compared to the NH3 volatilization, and since most of the root biomass is typically located below the 2 cm depth.

The processes incorporated in FANv2 are now more explicitly outlined in Section 2.2 just prior to equation (1).

*The authors miss the description of input datasets. In page 2, line 33: 'In this study FANv2 is run globally within the CLM for the six-year period 2010–2015 to simulate the present-day NH3 emissions, which are then compared with existing global and regional inventories.' Moreover, you mention NH3 emission is sensitive to climate change, but I did not find any information on climate conditions. The authors should add one more table listing the input datasets that were used to drive FANv2*

A table has been added to Appendix B.

*Meanwhile, you need to mention that N input data are only for one-year. Even though you only cover 6 years' simulation, the temporal changes in N input amount, especially at the regional scale, may introduce the uncertainty when you compare your results with existing global and regional inventories.*

The fertilizer inputs changed annually, as explained in our response to the comment about fertilizer amounts above, while the manure inputs were for 2010. We agree that this introduces some uncertainty to the comparisons between inventories. The uncertainty can be investigated using available multi-year emission inventories: for example, in the EDGAR4.3.2 inventory, the global NH3 emission increases by about 1.3 % per year in 2007-2012 (the last five years available). For India and China, the increase is up to ~3 % per year. This suggests that the different base years would have a relatively small effect compared to the overall differences between the inventories.

The temporal evolution of the fertilizer input has been clarified in Section 2.5.2.

*Here, you should at least cite articles to support your statement. Moreover, in Xu et al. (2019), they have considered the interaction between atmosphere and soils to investigate the effect of meteorological forcing*

Xu et al. (2019) provide interesting results regarding the role of atmospheric drivers in the NH3 emission. However, the study does not consider the transport and deposition of NH3 in the atmosphere. We recognize that the statement could be understood ambiguously, and have revised the introduction accordingly.

*2. Page 2, line 10: Please cite Xu et al. (2019) as well since they also emphasized the importance of environmental factors.*

Done.

*3. Page 4, line 16: Why 2.4 days? Any explanation for this assumption? You need to add citation here.*

The 2.4 days e-folding times were used in the previous FAN version (Riddick et al., 2016) based on the results of Agehara and Warncke (2005). The global sensitivity to this parameter was low (Section 3.3).

The citation (Agehara and Warncke, 2005) has been added.

*4. Page 5, line 21: Actually, it is not reasonable to use a constant Kd to deal with ammonium adsorption in soils.*

As noted recently by Pleim et al. (2019), a physically faithful description of the ammonium adsorption would require a nonlinear isotherm depending on several soil-dependent parameters which are generally not available. We agree that FANv2 and other models would benefit from a better characterization of the adsorption. However, given the uncertainties in global datasets of soil properties together with the uncertainty due to the linearized adsorption model, we believe that assuming a constant Kd is reasonable for the current version.

*5. Page 11: What is the time step for your simulation? Is it in second (s)? How did you get hourly, daily, monthly, and annual NH3 emission from s? Is there any assumption behind it? By the way, you mentioned an assumed 2.4 days for urea hydrolysis. How does the model deal with the 2.4 days reaction since your simulation is at second time step, right?*

The time step was 30 minutes (Section 2.7). The hourly, daily, monthly and annual emissions were obtained by averaging the emission flux. As noted above, the urea hydrolysis is evaluated as a first order process (Eq. 1), and the 2.4 days refers to an e-folding time, which corresponds to the rate of $4.83e-6$ $s^{-1}$ (Eq. A20).

*6. Page 18, line 6: Do you mean FANv2? I am confused with this sentence 'Since FAN assumes that fertilizers are applied in dry, granular form, no soil moisture perturbation is assumed for the fertilizer N pools. However, similar to urine, the formation of ammonium in urea hydrolysis increases the soil pH.' If there is no soil moisture effect, how does soil water content affect the NH3 volatilization?*

Thank you for the correction, the text should indeed read FANv2.

We mean that the fertilizer application is not assumed to change the soil moisture, contrary to urine patches or slurry. The existing soil moisture does affect the volatilization as discussed in section 2.2. The text has been changed to "Contrary to urine and slurry, fertilizer application is not assumed to change the soil moisture".

*7. Page 20, line 31: One time only for each crop?*

Yes. This follows from the parameterization used in the CLM crop model; we have clarified the text.

*8. Page 21, line 12: 'The CLM simulations were run in the satellite phenology mode and forced with the Global Soil Wetness Project Phase 3 (GSWP3) meteorological data set (http://hydro.iis.u-tokyo.ac.jp/GSWP3).' The model comparison project covers the time period of 1850-2010; however, this study focuses on the period of 2010-2015. Did you use climate data in 2010 to run the model for 2010-2015? Please clarify it.*

The GSWP3 dataset has been extended to 2010-2014. The statement on p. 21 refers to the single-point CLM runs which were performed separately for each site and the time of measurement. One set of the measurements (Bell et al., 2017) were made in 2015, and the CRUNCEP data were used instead of GSWP3. The remaining experiments were performed in late 1970s until early 2010s.

The global simulations for 2010-2015 were performed by coupling CLM with the Community Atmospheric Model (CAM; Section 2.7), which provided the meteorological forcing. The CAM simulation was forced by the dynamical fields from the MERRA meteorological reanalysis for 2010-2015.

We have revised the paragraph to emphasize the distinction between the point simulations and the main run for 2010-2015.

*9. Page 23, figure 5: Did you check the effect of different levels of fertilizer or manure? Why did you only include one site in China? What is the unit in this figure?*

Fig. 5 shows the volatilization losses as a fraction of the N application. The FANv2 model is linear with respect to the N application rate, and thus the absolute fertilization level would not affect the simulated loss fraction.

Apart from the exceptions noted in Section 2.6, we included measurements which (i) were published in international literature and (ii) were performed using micrometeorological techniques (as opposed to chambers and wind tunnels), and (iii) reported the location and time of the observation at sufficient detail. Unfortunately, we found only one site in China that satisfied these constraints.

*10. Page 27, figure 9: Can you explain why the fraction volatilized in urea is highest in Southern Africa as well as Northern Africa?*

Near the Equator, the volatilization rate is suppressed by high precipitation. The less humid climates typical to subtropical Africa are predicted to cause higher volatilization losses.

*11. Page 27, figure 10a: Why is it shown the highest fraction of NH3 volatilization in the desert Africa? As I see, it seems there is little manure there in Figure 1 in the supplement. Why is it shown the highest fraction of NH3 volatilization in Africa and South America in figure 10b? Moreover, it seems the fraction is fixed for most of land areas. Please explain it.*

The GLW2 dataset allocates a very small but nonzero livestock density to desert areas. The desert conditions favor large volatilization losses, which is visible in Fig. 10a. However, as shown in Fig. 8a, the absolute NH3 emission is negligible in this region.

Fig. 10b shows the fractional volatilization loss in barns and manure storage. As described in Section 2.5.1, the geographical variation of these losses is determined by the temperature-driven parameterization of Gyldenkaerne et al. (2005), and the volatilization rate in Fig. 10b therefore largely follows the global pattern of mean surface temperature. In contrast, the emissions from soils (Fig. 9 and Figs. 10a and 10c) depend on precipitation, soil moisture and other variables besides the temperature and therefore give a much higher variability.

*12. Page 30, figure 11: Please indicate the time period. Also, please indicate the version of EDGAR that you used for comparison and add the citation here.*

Done.

*13. Page 30, line 9: 'mainly due to increased emissions from manure management and grazing'. Please check it carefully. Is there any data to support it?*

The manure-originated emissions are higher than in any of the inventories included in Table 2, and the difference is especially large compared to EDGAR (Fig. 11). We consider this difference in closer detail in Section 4. However, we acknowledge that it is difficult to verify this prediction, since the literature and observations about nitrogen flows in Indian livestock systems are scarce.

*14. Page 30, line 13: 'Also losses for fertilizers differ between the versions. FANv1 treated all fertilizers as urea, resulting in a higher total volatilization rate (19 %) for synthetic fertilizers than FANv2 (13 %). However, the mean volatilization rate for urea in FANv2 is 19 %, which is similar to FANv1.' You described how FANv1 deals with no crop interaction. In page 9, line 1: 'FANv1 applies a 60 % reduction to the emissions to account for the NH3 captured by plant canopy; this reduction is included in the flux shown in Fig. 2a.' Did you consider this percentage (60%) when you compared your current results with that from FANv1? If no, why yours are similar to that in FANv1 for NH3 emission from urea? Please clarify.*

FANv1 applies the 60 % reduction for all NH3 emissions, and all results we cite for FANv1 include this reduction. However, the formulation of FANv1 does not include a soil resistance. The analysis in Section 2.2 as well as the experience from other models (Pleim et al., 2013) indicates that the emissions are often limited by the soil resistance, and therefore the 60 % reduction in FANv1 has to be understood to implicitly include some of the effects of soil. This makes the parameter specific to FANv1, and it would not be meaningful to apply the reduction to FANv2, or conversely, to evaluate the FANv1 emission without the reduction.

*15. Page 31, line 10: Your higher emission is from manure. The canopy capture may be not the explanation.*

As noted above, FANv1 uses the 60 % reduction for all emissions including those from manure (see comment above).

*16. Page 32, line 5-8: In this study, you simulated NH3 volatilization only covering 6 years. Climate change refers to a long-term change in weather. Thus, the author needs to put this into your future research even though you have conducted the sensitivity analysis to show how temperature and soil moisture can affect the NH3 volatilization processes. Please reorganize this sentence to describe what your model has done at the current stage.*

We have revised the paragraph as follows:

"Agricultural ammonia emissions are determined both by agricultural activity and environmental conditions. Both of these aspects of ammonia emissions have been incorporated into the process model FANv2, which embedded within the CESM simulates agricultural ammonia emissions globally. While we simulated the response of emissions from various agricultural processes to meteorological forcing on yearly level, FANv2 could be used to estimate how the emissions respond to climate change on decadal to century timescales, or how the emissions respond to weather anomalies on hourly to daily timescales."

*17. Page 32, line 12: You mentioned that you used the local meteorological conditions to parameterize your model. Where did these data come from? It is necessary to add this information in Appendix B. Please add more detailed information on each site, such as longitude/latitude, soil type, crop type, annual temperature, annual precipitation, etc.*

We have added the longitude and latitude for each site, and the soil pH and cation exchange capacity are shown if available. Most of the studies did not report annual temperature or precipitation.

*18. Page 32, line 17: 'In particular, livestock manure is treated everywhere as a slurry, which is likely to lead to uncertainties in developing countries where handling manure as slurry is uncommon'. Any citation?*

We have changed the sentence as follows: "In particular, livestock manure is treated everywhere as a slurry and applied on land. This is likely to lead to uncertainties where handling manure as slurry is uncommon (e.g. Ndambi et al. (2019) for sub-Saharan Africa), or where a significant fraction of manure is discharged to waterways (e.g. Strokal et al. (2016) for China and IAEA (2008) for Southeast Asia)."

*19. Page 33, line 3: 'It is difficult to isolate any particular factor that causes FANv2 to underestimate the Chinese emission factors compared to the other inventories'. First, you only include one site in China, which may cause the underestimate. Second, did you check fertilizer and manure data and compare them with the previous study? It is possible that the amount of N inputs to Chinese agricultural soils differ a lot in different studies. Please at least state this here.*

See comment above with regard to the Chinese sites. The difference in total emissions is caused by the emission from fertilizer application; the manure NH3 emissions are well within the range of previous estimates. The fertilizer application rates do differ between those used in FANv2 and the previous studies, and this does explain some of the difference in the NH3 emission. However, as discussed in the text, a considerable difference remains even when comparing the emission factors (NH3-N volatilized divided by N fertilizer applied). We have added to the text a discussion about estimates of total fertilizer use in China.

*20. Page 33, line 10: Omission of rice paddies may lead to the underestimate. It has been reported that 90% of rice is cultivated in East, South, and Southeast Asia. Xu et al. (2019) claims that rice cultivation has become the largest source for NH3 emission due to its high rate of fertilization and warm temperature. Please at least discuss the important role of rice here.*

We state in the text: "The explicit consideration of rice paddies could also be potentially important; FANv2 and the CLM do not explicitly simulate rice paddies, even though the processes controlling NH3 volatilization in paddies are likely different from those in upland crops. For Chinese rice paddies, Wang et al. (2018) report an average emission factor of about 18 % for urea, which is higher than the factor calculated from FANv2, but not significantly higher than the overall emission factors used by Zhang et al. (2017b) and Zhang et al. (2018). This suggests that the omission of rice paddies in FANv2 is not solely responsible for the discrepancy in emissions." Thus we believe the text states the importance of rice paddies, but also the fact that they may be unlikely to cause the difference. We have revised the text to further emphasize the role of rice cultivation in NH3 emissions as pointed out in Xu et al. (2019).

*21. Page 35, line 8-9: 'The model evaluates ammonia emissions interactively with the simulated atmosphere, and responds to both short and long-term variations in the meteorological forcing.' Throughout the main text, I did not see any long-term estimates of NH3 volatilization. You only reported the average annual amount for the period of 2010-2015. Also, the author did not report intra-annual estimates of NH3 volatilization. Please carefully state 'both short and long-term variations in the meteorological forcing' since these were not fully reported in the current study. By the way, I am curious about the intra-annual results. Is it possible to present your intra-annual results in the main text?*

We have changed the sentence to read "The model evaluates ammonia emissions interactively with the simulated atmosphere and therefore responds to variations in the meteorological forcing".

Due to the length of the manuscript we prefer to not include additional material. Seasonal variations of the simulated emissions will be analyzed in an upcoming manuscript, where they are evaluated using measurements of ammonium wet deposition. In brief, the evaluation indicates that the model reproduces the main seasonality of NH3 emissions, but also that the model would benefit from a more detailed representation of fertilizer application and timing.

*22. Page 35, line 23: It is odd to mention 'This sensitivity includes the effect of increasing grazing and earlier crop planting dates in warmer climates.' What is the relationship between this sentence and the sensitivity analysis? You did not consider these factors in the sensitivity test, right? Please reorganize it.*

This refers to the last paragraph in Section 4, which discusses diagnosing the temperature response from the geographical variations in the volatilization rate as described in the Suppl. Section 5 (Section S6 in the revised manuscript). To make the manuscript more coherent, we have expanded the discussion of the results in the main text and made it a new Subsection 3.4 in Results.

**Response to reviewer 2**

*1. The manuscript is very lengthy; and a lot of the modeling framework background should be shortened.*

We have shortened the manuscript by moving the more technical material in Section 2 to the supplementary information.

*2. The assumptions made in the manuscript need to be explained and or justified. For example pg 3, line 30 "the soil below the topmost layer is treated as a sink——and all the N transport below 2 cm layer is assumed to be permanently unavailable——" needs to be justified.*

We have added the following explanation to the text:

"Ammoniacal nitrogen is generally transported and distributed within the soil column by molecular diffusion and movement of soil water. However, after a surface application of synthetic fertilizers or manure, the slow molecular diffusion within soil pores initially confines the ammoniacal N to the first few centimeters of the soil column (Pang et al., 1973; Sadeghi et al., 1989). This allows the ammonia volatilization to be evaluated using a single model layer similar to the earlier models of Sherlock and Goh (1985), Li et al. (2012) and Móring et al. (2016). In FANv2, this layer covers the topmost $\Delta z$ = 2 cm of the soil profile, which coincides with the topmost soil layer in CLM5; different values for $\Delta z$ are tested in Section 3.3. Since the TAN concentration in the topmost layer is much higher than in the soil below, the underlying soil is not assumed to contribute to the emission, and the TAN transported below the 2 cm layer is assumed unavailable for volatilization."

*3. Most of the References are old i.e. prior to 2015. More recent references i.e. 2018 and 2019 should be provided and their results discussed.*

We have added two new references (Battye et al., 2017 and Pleim et al., 2019). The manuscript currently cites 33 papers published in 2015 or later.

*4. Pg 4, line 2 The role of microbial activity is not simulated. This is a very important component of the N biogeochemical cycling and needs to be addressed.*

We agree on the importance of microbial processes in N cycling. Several microbial processes affecting the NH3 emission (urea decomposition, nitrification and N mineralization from manure) are implicitly represented in FANv2, but since FANv2 is designed to evaluate NH3 emissions exclusively from agricultural soils following a fertilizer or manure application, a general representation of the microbial N cycling is outside the scope of FANv2. Interaction with the soil N cycling could nevertheless be improved in a future version by integrating FANv2 more closely with the soil biogeochemistry simulated elsewhere in the CLM5. We have added a brief discussion of this aspect into Section 2.2.

*5. Pg 2, line 33 "In this study FANv2 is run globally within the CLM for the six-year period 2010–2015 to simulate the present-day NH3 emissions, which are then compared with existing global and regional inventories". This is fundamentally incorrect since ammonia emissions and thus ammonia concentrations calculated for 2015 will be different from present-day i.e. 2019 or 2020.*

We have changed the sentence to read "In this study FANv2 is run globally within the CLM to evaluate the NH3 emissions for the six-year period 2010--2015, which are then compared with existing global and regional inventories.".

*6. Pg 4, Eq 2. The rates are all first order. This assumption needs to both be explained and justified.*

First order kinetics has been found to provide an adequate description of the organic matter turnover in CLM and many other biogeochemical models (see e.g. Manzoni and Porporato 2009).

*7. Pg 5, Line 2. "The diffusion and leaching fluxes are not evaluated for the available and resistant organic N". This reviewer does not understand what is being suggested.*

We have changed the sentence as follows: "The organic N fractions (resistant and available organic N) are assumed to be transported only by the mechanical disturbances described by the rate coefficient $k_m$, and the molecular diffusion in gas or aqueous phase is not evaluated."

*8. Pg. 23, Figure 5 b. Model results and observations do not agree well. This needs to be explained in detail.*

We have added the following discussion:

"The simulated volatilization losses were evaluated against data from experimental studies, which consist of one or more experiment typically spanning a period of several weeks. The observations are therefore local in both space and time, which makes them challenging to reproduce with a model intended for continental or global scales. Difficulties may arise

particularly due to the emissions' complex response to soil moisture (Section 2.2), which can be strongly affected by local-scale orography and drainage conditions as well as unresolved precipitation patterns. The evaluation presented here therefore focuses on the model's ability to mechanistically reproduce the differences in the volatilization rates from different types of fertilizers and manure."

[revised manuscript text omitted]

**S1  Patch aging scheme**

The upscaling scheme used in FANv2 is based on a generic approach for evaluating the aggregate N fluxes for a distribution of patches based on the fluxes of a single patch. The main underlying assumptions are that

- Each patch has a well-defined age $a > 0$ which is equal to the time elapsed since the patch was created.

- The state of each patch is uniquely determined by its age $a$ and the current time $t$.

- The patches are aggregated over an area large enough that the distribution of the state variables can be approximated by integrable functions of $a$ and $t$.

Under these assumptions, the upscaling approach aims to link the evolution of the nitrogen content $N(t)$ of a single patch, given by a differential equation in the form

$$\frac{dN}{dt} = f(N, t), \tag{S1}$$

to an age and time dependent function $n(a, t)$ which describes how the nitrogen is distributed between the patches of different age.

We start by defining the density function $w(a, t)$ such that the surface area occupied by patches aged between $a$ and $a + \Delta a$ is given by

$$W(t) = \int_a^{a+\Delta a} w(a', t) da'. \tag{S2}$$

Since the quantities simulated by FANv2 are expressed as area densities, $W(t)$ is a fraction (area of patches / total area) and $w(a, t)$ has the unit $s^{-1}$.

Any feature of the patches is now expressed as two-dimensional functions $\phi(a, t)$, and for every fixed age $a_0$, the single-variable functions

$$\Phi(t) = \phi(a_0 + t, t) \tag{S3}$$

define the evolution (Eq. S1) of an individual patch aged $a_0$ at $t = 0$. The total derivative of $\phi(a, t)$ can therefore be identified with the tendency of $\Phi$,

$$\frac{\partial \phi}{\partial a} + \frac{\partial \phi}{\partial t} = \frac{d\Phi}{dt}, \tag{S4}$$

where we substituted $\partial a / \partial t = 1$. Since FANv2 assumes that the area of a patch does not change with $a$, application of (S4) to the density function $w(a, t)$ yields a "continuity equation" for $w$:

$$\frac{\partial w}{\partial a} + \frac{\partial w}{\partial t} = 0. \tag{S5}$$

The equation (S1) governing the nitrogen density of a single patch can then be understood in the sense of Eq. (S4):

$$\frac{\partial N}{\partial t} + \frac{\partial N}{\partial a} = f(N, a, t). \tag{S6}$$

Combining Eqs. (S5) and (S6) yields and equation for the area-weighted N density $n(a, t) = w(a, t)N(a, t)$:

$$\frac{d}{dt}(wN) = \frac{\partial n}{\partial a} + \frac{\partial n}{\partial t} = w(a, t)f(N(a, t), a, t). \tag{S7}$$

If $f$ is linear with respect to $N$, Eq. (S7) simplifies to

$$\frac{\partial n}{\partial a} + \frac{\partial n}{\partial t} = f(n(a, t), a, t). \tag{S8}$$

Eq. (S7) is a hyperbolic first-order partial differential equation. Eq. (**??**) (aging) in the main manuscript is obtained as a first order finite volume approximation of $\partial n/\partial a$. Applying a first-order, forward-in-time discretization yields a two-step numerical scheme implemented in FANv2:

1. For each $i$, update $n_i(t)$ according to Eq. (S1) as

$$n_i(t') = n_i(t) + w_i(t)\overline{f(N_i(t), t)}\Delta t \tag{S9}$$

where $\Delta t$ denotes the time step and the tendency $\overline{f(N_i(t), t)}$ is evaluated as a mean over the $i$th age class.

2. Transfer nitrogen from the younger to older age classes according to

$$n_i(t + \Delta t) = \begin{cases} n_i(t') - \Delta t \frac{n_i(t')}{\Delta a_i}, i = 1 \\ n_i(t') - \Delta t \left( \frac{n_i(t')}{\Delta a_i} - \frac{n_{i-1}(t')}{\Delta a_{i-1}} \right), i > 1 \end{cases} \tag{S10}$$

where the ages $a_{i-1}$ and $a_i$ define the $i$th age class and $\Delta a_i = a_i - a_{i-1}$.

In FANv2, the tendency $f$ in Eq. (S1) is linear with respect to $N$. Substituting $N_i = n_i/w_i$ simplifies Eq. (S9) to

$$n_i(t') = n_i(t) + \overline{f(n_i(t), t)}\Delta t, \tag{S11}$$

and the area fractions $w_i$ are therefore not needed.

The fertilizer or manure N is initially introduced to the youngest age class, and subsequently transferred through the sequence of age classes as described by Eq. (S10), until reaching the final class $i*$. By Eq. (S10), nitrogen is removed from the final age class at a rate equal to $1/\Delta a_{i*}$, which can be made arbitrarily small by the choice of $\Delta a_{i*}$. In FANv2, the final bins have $\Delta a_{i*} = 360$ days, which sets the maximum age of the N patches considered. Although not implemented in the current version, the nitrogen aged beyond $\Delta a_{i*}$ could be transferred into the soil N pools in the CLM.

If the tendency $f(N, a, t)$ is nonlinear for $N$, Eq. (S7) requires evaluating the density function $w(a, t)$. This can be obtained from the exact solution to Eq. (S5),

$$w(a, t) = w(0, t - a), \tag{S12}$$

which is the area fraction occupied by fresh patches at time $t - a$. Alternatively, $w(a, t)$ could be evaluated numerically with a similar procedure as for $n(a, t)$, which has the advantage of guaranteeing consistency between the evolution of $w(a, t)$ and $n(a, t)$.

**S1.1 Application to moisture content**

The framework described above can be used to aggregate the water budget for a class of patches. This is based on the assumption that the volumetric soil moisture of the patches can be expressed as a sum of a $t$-dependent background and an $a$-dependent perturbation:

$$\theta(a, t) = \theta_b(t) + \delta\theta(a). \tag{S13}$$

Applying Eq. (S4) to the single-patch water budget yields

$$\frac{d\theta}{dt} = \frac{\partial\theta}{\partial t} + \frac{\partial\theta}{\partial a} = \frac{q_{top} - q_{bot}}{\Delta z}, \tag{S14}$$

where $q_{top}$ and $q_{bot}$ are the water fluxes at the top and bottom of the layer with thickness $\Delta z$. Substituting Eq. (S13) and integrating Eq. (S14) over the age range $(a_i, a_{i+1})$ yields the budget averaged over the age class, as in Eq. (**??**) in the main text:

$$\frac{\overline{q_{top} - q_{bot}}}{\Delta z} = \frac{\overline{d\theta_b}}{dt} - \frac{\delta\theta(a_{i+1}) + \delta\theta(a_i)}{a_{i+1} - a_i}, \tag{S15}$$

where the lines denote $a$-averages, such as

$$\overline{q_{top}} = \frac{\int_{a_i}^{a_{i+1}} q_{top} \, da}{a_{i+1} - a_i}. \tag{S16}$$

**S2 Sub-models for pastures, slurry and synthetic fertilizers**

**S2.1 Grazed pastures**

[revised manuscript text omitted]

**S3 Nitrogen inputs**

**S3.1 Manure N excretion**

The estimated nitrogen excretion of livestock is based on the coefficients and animal weights in IPCC (2006). The yearly N excretion rates are listed in Table S1. For cattle, the coefficients are given for dairy and other cattle; the average is evaluated assuming 26 % fraction of dairy cattle for Europe, North America and Oceania , 35 % for India, and 14 % for other regions. The dairy cattle fraction for India is based on statistics published by the Indian Ministry of Agriculture and Farmers Welfare (2015), for other regions, the fraction is adapted from Bouwman et al. (1997). Geographical distributions of manure N in pastoral and landless production systems are shown in Fig. S2.

**S3.2 Fertilizers**

As noted in the main text (Section 2.5.2), the N fertilizer application in the simulations is prescribed in the CLM input dataset (Lawrence et al., 2016). The disaggregation to urea, ammonium, and nitrate N was based on the consumption statistics of the International Fertilizer Association (IFA) for the year 2013. The country-level values were used to derive gridded maps of fraction of urea and nitrate N. For countries with missing data for urea, the fraction was extrapolated from the neighboring areas. To ensure that the sum of urea and nitrate fraction remains below 1, the nitrate fraction was not extrapolated but assumed zero when missing. Maps of urea and other fertilizer use are shown in Fig. S3.

**S4 Grazing and housing periods in mixed systems**

Ruminants in mixed/landless systems are assumed to graze when the 10-day mean daily minimum temperature exceeds $+10°$ C. At other times the livestock is assumed to remain in animal housings. Fig. S4 shows the number of yearly housing days estimated using the temperature threshold and

Table S1: Nitrogen excretion coefficients for livestock, kgN yr$^{-1}$ head $^{-1}$.

| | North America | Western Europe | Eastern Europe | Oceania | Latin America | Africa | Middle East | Asia | India |
|---|---|---|---|---|---|---|---|---|---|
| Cattle, average | 58.0 | 65.0 | 55.3 | 65.5 | 44.2 | 42.6 | 52.7 | 42.4 | 25.4 |
| Dairy cattle | 97.0 | 105.1 | 70.3 | 80.3 | 70.1 | 60.2 | 70.3 | 60.0 | 47.2 |
| Other cattle | 44.0 | 50.6 | 50.0 | 60.2 | 40.1 | 39.8 | 49.9 | 39.6 | 13.7 |
| Pigs, average | 11.2 | 16.1 | 17.0 | 15.6 | 16.8 | 16.8 | 16.8 | 5.1 | 5.1 |
| Market swine | 7.1 | 9.3 | 10.0 | 8.7 | 16.0 | 16.0 | 16.0 | 4.3 | 4.3 |
| Breeding swine | 17.3 | 30.4 | 30.2 | 30.2 | 5.6 | 5.6 | 5.6 | 2.5 | 2.5 |
| Poultry | 0.5 | 0.5 | 0.5 | 0.5 | 0.5 | 0.5 | 0.5 | 0.5 | 0.5 |
| Sheep | 7.4 | 15.0 | 15.9 | 20.0 | 12.0 | 12.0 | 12.0 | 12.0 | 12.0 |
| Goats | 6.3 | 18.0 | 18.0 | 20.0 | 15.0 | 15.0 | 15.0 | 15.0 | 15.0 |
| Buffalo | 44.4 | 44.4 | 44.4 | 44.4 | 44.4 | 44.4 | 44.4 | 44.4 | 34.5 |

[Figure]

Figure S2: Manure N production (gN m$^{-2}$yr$^{-1}$) in pastoral (left) and mixed/landless systems (right).

[Figure]

Figure S3: Yearly application (gN m$^{-2}$yr$^{-1}$) of urea (left) and other N fertilizers (right).

global temperature data from the NCEP reanalyses for 1980-2010. Fig. S5 compares the yearly housing days for Europe with the survey results (for cattle) reported by Klimont and Brink (2004). The values shown for European countries are weighted averages; the distribution of housing days was weighted by the population distribution of cattle within each country.

**S5 Sensitivity experiments**

The sensitivity to model parameters was examined with experiments consisting of 2-year CLM simulations forced by the GSWP3 dataset. The experiments, the modified parameters, and changes in $NH_3$ emission are listed in Table S2. Switching the meteorological forcing from the CAM simulation to GSWP3 changed to total emission by $\sim$2 %; all other changes in emissions are reported with regard to the GSWP3-driven control simulation.

In the experiments evaluating sensitivity to the layer thickness $\Delta z$, only the thickness used in FAN computations was changed; the soil layers used in elsewhere in CLM were not changed. The water content used in FAN computations was taken for the topmost CLM layer in all experiments.

The sensitivity to manure TAN fraction by was evaluated by computing the emissions at 0 and 100 % TAN fractions. Since the model is linear with respect to N input, this allows calculating the $NH_3$ volatilization for any TAN fraction $f_{TAN}$ as

$$F_M(f_{TAN}) = f_{TAN} \times F_M(f_{TAN} = 1) + (1 - f_{TAN}) \times F_M(f_{TAN} = 0), \quad (S26)$$

where $F_M$ denotes the total $NH_3$ emission from manure. In the experiments, $F_M(0) = 6.8$ Tg N and $F_M(1) = 56$ Tg N. Varying the TAN fraction between 40 and 80 % would therefore result in about $\pm 10$ TgN variation in the global $NH_3$ emission.

[Figure]

Figure S4: Average number of housing days per year estimated using the 1980-2010 NCEP temperature reanalyses.

[Figure]

Figure S5: Yearly housing days reported by Klimont and Brink (2004) (left) and estimated using a temperature threshold (right).

Table S2: Relative changes in NH$_3$ emissions in the sensitivity experiments. The change for the control run is relative to the main run; for other experiments the change with respect to the control run is shown. Percent change in emission per percent change in parameter is shown in parentheses when applicable.

| Parameter | Value | Africa | Asia except China and India | China | Europe | India | Latin America | North America | Oceania | World Total | World Fertilizer | World Manure |
|---|---|---|---|---|---|---|---|---|---|---|---|---|
| Control[1] | | +8 | +2 | −0 | −1 | +1 | +4 | +4 | +1 | +2 | +6 | +1 |
| $\tau_{\mathrm{infl}}$[2] | ×0.5 | −2 | −2 | −3 | −5 | −2 | −3 | −3 | −3 | −3 (+0.1) | +0 | −3 (+0.1) |
| | ×2.0 | +2 | +3 | +5 | +6 | +2 | +3 | +4 | +4 | +3 (+0.0) | +0 | +4 (+0.0) |
| $d_0$ for urine [3] | −2 mm | +7 | +4 | +2 | +1 | +4 | +5 | +2 | +4 | +4 (−0.1) | +0 | +5 (−0.1) |
| | +2 mm | −6 | −3 | −2 | −1 | −3 | −5 | −2 | −5 | −3 (−0.1) | +0 | −4 (−0.1) |
| pH for manure[4] | +0.5 | +9 | +8 | +8 | +12 | +6 | +9 | +9 | +19 | +9 | +0 | +11 |
| | −0.5 | −8 | −6 | −6 | −10 | −4 | −7 | −7 | −11 | −7 | +0 | −9 |
| $f_{\mathrm{TAN}}$[5] | 0.0 | −70 | −62 | −57 | −77 | −46 | −72 | −54 | −64 | −63 (+0.6) | +0 | −81 (+0.8) |
| | 1.0 | +46 | +42 | +38 | +52 | +31 | +48 | +36 | +43 | +42 (+0.6) | +0 | +54 (+0.8) |
| pH for other fert.[6] | 7.0 | +1 | +2 | +1 | +8 | +0 | +3 | +11 | +0 | +3 | +12 | +0 |
| pH for urea[7] | | −0 | −0 | +2 | −1 | +1 | −2 | −0 | +2 | +0 | +1 | +0 |
| Urea decomp.[8] | ×0.5 | +0 | +2 | +4 | +1 | +3 | +1 | +3 | +2 | +2 (−0.0) | +9 (−0.2) | +0 |
| | ×2.0 | −0 | −3 | −5 | −2 | −4 | −2 | −3 | −2 | −3 (−0.0) | −11 (−0.1) | +0 |
| Fert. timing[9] | 1 day | +0 | −2 | −0 | +2 | −4 | +0 | +1 | −1 | −1 | −3 | +0 |
| | 90 days | −1 | +4 | −6 | −2 | −1 | −1 | +2 | +5 | −0 | −2 | +0 |
| $\Delta z$[10] | ×0.5 | +5 | +12 | +22 | +19 | +13 | +13 | +29 | +24 | +15 (−0.3) | +41 (−0.8) | +7 (−0.1) |
| | ×2.0 | −28 | −26 | −27 | −20 | −30 | −24 | −31 | −33 | −27 (−0.3) | −52 (−0.5) | −19 (−0.2) |
| $K_d$[11] | 0 | +9 | +12 | +20 | +17 | +13 | +16 | +21 | +24 | +15 (−0.2) | +30 (−0.3) | +11 (−0.1) |
| | ×10.0 | −35 | −31 | −38 | −32 | −36 | −34 | −40 | −43 | −35 (−0.0) | −55 (−0.1) | −29 (−0.0) |
| Fraction of grazing[12] | +0.30 | −17 | −12 | −8 | −6 | −19 | −20 | −5 | −15 | −14 (−0.3) | +0 | −18 (−0.4) |
| | −0.30 | +17 | +12 | +8 | +7 | +19 | +21 | +5 | +15 | +14 (−0.3) | +0 | +18 (+0.4) |

[1]Control run with GSWP3 forcing. [2]Slurry infiltration time. [3]See Section 2.4.2. [4]Add or subtract from pH of N classes G1–G3 and S0–S3 . [5]Fraction of TAN in manure. [6]Fixed pH for age class F4. [7]pH for N classes F1, F2, F3 set to soil pH + 0.5, 2, 1.5 units. [8]Time constant $1/k_U$. [9]Fertilizer application window, days from leaf emergence. [10]Soil layer thickness. [11]Adsorption constant. [12]Maximum fraction of ruminants grazing in mixed production systems.

**S6 Sensitivity to mean temperature and precipitation**

The sensitivity of $NH_3$ emissions to mean temperature was investigated with a linear regression on the geographical distribution of the simulated emissions. The model grid cells were first categorized by yearly rainfall, and then, the normalized volatilization loss ($NH_3$ emitted / N applied) within each category was fit with a linear function of the yearly mean temperature, assuming that for each grid cell,

$$NH_3/N_{appl} = a + bT + r \tag{S27}$$

where $a$ and $b$ are the regression parameters, $T$ is the temperature (°C) and the residual $r$ represents the temperature-independent effects. The temperature sensitivity is obtained from Eq. (S27) as

$$\Delta(NH_3) = bN_{appl}\Delta T. \tag{S28}$$

The temperature sensitivity of the total emission is then obtained by summing Eq. (S28) over the gridcells and precipitation categories.

The regression was evaluated separately for manure, urea, and other synthetic fertilizers. Exponential fits were tested in addition to the linear fits, however, the exponential fits invariably had lower $R^2$ than the linear fits and thus were not analyzed further. Results of the regression and the temperature response are shown in Table S3.

Table S3: Parameters of linear fits of the normalized volatilization loss (N emitted / N applied) as function of local mean temperature in °C. Present day manure N production or fertilizer N application and the corresponding $NH_3$ emissions are shown for each category.

| Source | Precipitation, mm | Intercept | Slope, $K^{-1}$ | $R^2$ | N applied, Tg | $NH_3$ emitted, Tg N | Temperature sensitivity, % $K^{-1}$ |
|---|---|---|---|---|---|---|---|
| Manure | $< 200$ | 0.13 | 0.009 | 0.87 | 8.5 | 2.9 | 2.5 |
| | $200 - 500$ | 0.15 | 0.008 | 0.65 | 20.8 | 6.4 | 2.5 |
| | $500 - 1000$ | 0.18 | 0.006 | 0.40 | 41.1 | 12.5 | 1.9 |
| | $1000 - 2000$ | 0.19 | 0.005 | 0.18 | 36.4 | 11.2 | 1.7 |
| | $> 2000$ | 0.15 | 0.006 | 0.07 | 12.8 | 3.6 | 2.0 |
| | Total | | | | 119.6 | 36.5 | 2.0 |
| Urea | $< 200$ | 0.20 | 0.005 | 0.05 ($p = 0.01$) | 3.1 | 0.6 | 2.4 |
| | $200 - 500$ | 0.00 | 0.013 | 0.48 | 8.5 | 1.6 | 7.1 |
| | $500 - 1000$ | -0.03 | 0.014 | 0.70 | 13.1 | 2.4 | 7.5 |
| | $1000 - 2000$ | 0.00 | 0.010 | 0.45 | 12.8 | 2.8 | 4.7 |
| | $> 2000$ | 0.04 | 0.005 | 0.09 | 4.7 | 0.7 | 3.2 |
| | Total | | | | 42.2 | 8.1 | 5.7 |
| Other fert. | $< 200$ | -0.01 | 0.010 | 0.29 | 1.0 | 0.2 | 6.1 |
| | $200 - 500$ | -0.05 | 0.010 | 0.33 | 7.7 | 0.8 | 9.3 |
| | $500 - 1000$ | -0.05 | 0.007 | 0.34 | 20.9 | 1.4 | 10.8 |
| | $1000 - 2000$ | -0.04 | 0.005 | 0.22 | 8.2 | 0.5 | 9.0 |
| | $> 2000$ | 0.00 | 0.001 | 0.03 ($p = 0.02$) | 2.6 | 0.1 | 4.1 |
| | Total | | | | 40.4 | 2.9 | 9.7 |
| All sources | Total | | | | 202.1 | 47.5 | 3.1 |

**S1  Patch aging scheme**

The upscaling scheme used in FANv2 is based on a generic approach for evaluating the aggregate N fluxes for a distribution of patches based on the fluxes of a single patch. The main underlying assumptions are that

- Each patch has a well-defined age $a > 0$ which is equal to the time elapsed since the patch was created.

- The state of each patch is uniquely determined by its age $a$ and the current time $t$.

- The patches are aggregated over an area large enough that the distribution of the state variables can be approximated by integrable functions of $a$ and $t$.

Under these assumptions, the upscaling approach aims to link the evolution of the nitrogen content $N(t)$ of a single patch, given by a differential equation in the form

$$\frac{dN}{dt} = f(N, t),\tag{S1}$$

to an age and time dependent function $n(a, t)$ which describes how the nitrogen is distributed between the patches of different age.

We start by defining the density function $w(a, t)$ such that the surface area occupied by patches aged between $a$ and $a + \Delta a$ is given by

$$W(t) = \int_a^{a+\Delta a} w(a', t)da'.\tag{S2}$$

Since the quantities simulated by FANv2 are expressed as area densities, $W(t)$ is a fraction (area of patches / total area) and $w(a, t)$ has the unit $\text{s}^{-1}$.

Any feature of the patches is now expressed as two-dimensional functions $\phi(a, t)$, and for every fixed age $a_0$, the single-variable functions

$$\Phi(t) = \phi(a_0 + t, t)\tag{S3}$$

define the evolution (Eq. S1) of an individual patch aged $a_0$ at $t = 0$. The total derivative of $\phi(a, t)$ can therefore be identified with the tendency of $\Phi$,

$$\frac{\partial \phi}{\partial a} + \frac{\partial \phi}{\partial t} = \frac{d\Phi}{dt},\tag{S4}$$

where we substituted $\partial a/\partial t = 1$. Since FANv2 assumes that the area of a patch does not change with $a$, application of (S4) to the density function $w(a, t)$ yields a "continuity equation" for $w$:

$$\frac{\partial w}{\partial a} + \frac{\partial w}{\partial t} = 0.\tag{S5}$$

The equation (S1) governing the nitrogen density of a single patch can then be understood in the sense of Eq. (S4):

$$\frac{\partial N}{\partial t} + \frac{\partial N}{\partial a} = f(N, a, t).\tag{S6}$$

Combining Eqs. (S5) and (S6) yields and equation for the area-weighted N density $n(a, t) = w(a, t)N(a, t)$:

$$\frac{d}{dt}(wN) = \frac{\partial n}{\partial a} + \frac{\partial n}{\partial t} = w(a, t)f(N(a, t), a, t).\tag{S7}$$

If $f$ is linear with respect to $N$, Eq. (S7) simplifies to

$$\frac{\partial n}{\partial a} + \frac{\partial n}{\partial t} = f(n(a, t), a, t).\tag{S8}$$

Eq. (S7) is a hyperbolic first-order partial differential equation. Eq. (**??**) (aging) in the main manuscript is obtained as a first order finite volume approximation of $\partial n/\partial a$. Applying a first-order, forward-in-time discretization yields a two-step numerical scheme implemented in FANv2:

1. For each $i$, update $n_i(t)$ according to Eq. (S1) as

$$n_i(t') = n_i(t) + w_i(t)\overline{f(N_i(t), t)}\Delta t \tag{S9}$$

where $\Delta t$ denotes the time step and the tendency $\overline{f(N_i(t), t)}$ is evaluated as a mean over the $i$th age class.

2. Transfer nitrogen from the younger to older age classes according to

$$n_i(t + \Delta t) = \begin{cases} n_i(t') - \Delta t \frac{n_i(t')}{\Delta a_i}, i = 1 \\ n_i(t') - \Delta t \left( \frac{n_i(t')}{\Delta a_i} - \frac{n_{i-1}(t')}{\Delta a_{i-1}} \right), i > 1 \end{cases} \tag{S10}$$

where the ages $a_{i-1}$ and $a_i$ define the $i$th age class and $\Delta a_i = a_i - a_{i-1}$.

In FANv2, the tendency $f$ in Eq. (??) (time integration) is a first-order forward-in-time discretization for $\partial n/\partial t$. S1) is linear with respect to $N$. Substituting $N_i = n_i/w_i$ simplifies Eq. (S9) to

$$n_i(t') = n_i(t) + \overline{f(n_i(t), t)}\Delta t, \tag{S11}$$

and the area fractions $w_i$ are therefore not needed.

The fertilizer or manure N is initially introduced to the youngest age class, and subsequently transferred through the sequence of age classes as described by Eq. (S10), until reaching the final class $i*$. By Eq. (S6)is analogous to one-dimensional tracer transport, where $w$ becomes the fluid density, $N$ becomes the tracer mixing ratio, and $n$ becomes the tracer concentration. S10), nitrogen is removed from the final age class at a rate equal to $1/\Delta a_{i*}$, which can be made arbitrarily small by the choice of $\Delta a_{i*}$. In FANv2, the final bins have $\Delta a_{i*} = 360$ days, which sets the maximum age of the N patches considered. Although not implemented in the current version, the nitrogen aged beyond $\Delta a_{i*}$ could be transferred into the soil N pools in the CLM.

If the tendency $f(N, a, t)$ is nonlinear for $N$, Eq. (S7) requires evaluating the density function $w(a, t)$. This can be obtained from the exact solution to Eq. (S5),

$$w(a, t) = w(0, t - a), \tag{S12}$$

which is the area fraction occupied by fresh patches at time $t - a$. Alternatively, $w(a, t)$ could be evaluated numerically with a similar procedure as for $n(a, t)$, which has the advantage of guaranteeing consistency between the evolution of $w(a, t)$ and $n(a, t)$.

**S1.1   Application to moisture content**

As noted in Section 2.4.2 in the main text, the The framework described above can be used to aggregate the water budget for a class of patches. This is based on the assumption that the volumetric soil moisture of the patches can be expressed as a sum of a $t$-dependent background and an $a$-dependent perturbation:

$$\theta(a, t) = \theta_b(t) + \delta\theta(a). \tag{S13}$$

Applying Eq. (S4) to the single-patch water budget yields

$$\frac{d\theta}{dt} = \frac{\partial\theta}{\partial t} + \frac{\partial\theta}{\partial a} = \frac{q_{top} - q_{bot}}{\Delta z}, \tag{S14}$$

where $q_{top}$ and $q_{bot}$ are the water fluxes at the top and bottom of the layer with thickness $\Delta z$. Substituting Eq. (S13) and integrating Eq. (S14) over the age range $(a_i, a_{i+1})$ yields the budget averaged over the age class, as in Eq. (??) in the main text:

$$\frac{\overline{q_{top} - q_{bot}}}{\Delta z} = \frac{\overline{d\theta_b}}{dt} - \frac{\delta\theta(a_{i+1}) + \delta\theta(a_i)}{a_{i+1} - a_i}, \tag{S15}$$

where the lines denote $a$-averages, such as

$$\overline{q_{top}} = \frac{\int_{a_i}^{a_{i+1}} q_{top}\, da}{a_{i+1} - a_i}. \tag{S16}$$

**S2 Sub-models for pastures, slurry and synthetic fertilizers**

**S2.1 Grazed pastures**

[revised manuscript text omitted]

$$R_{\mathrm{sl}} \equiv \min(W/2, d_{sl})/D_{\mathrm{NH4}} \tag{S21}$$

$$R_{\mathrm{ss\uparrow}} \equiv \frac{\max(W/2 - d_{sl}, 0)}{\theta_s \xi(\theta_s) D_{\mathrm{NH4}}}$$

$$R_{\mathrm{ss\downarrow}} \equiv \frac{W}{2\theta_s \xi(\theta_s) D_{\mathrm{NH4}}},$$

where the tortuosity factor $\xi$ is applied to the molecular diffusivity $D_{\mathrm{NH4}}$ within soil but not to the slurry on surface ($R_{\mathrm{sl}}$). The remaining resistances ($R_a$, $R_b$, $R_{\mathrm{gas\downarrow}}$, and $R_{\mathrm{aq\downarrow}}$), and subsequently the nitrogen fluxes, are evaluated as in Section ??.

The depths $d_{sl}$ and $d_{sat}$ need to be determined for evaluating the resistances. At $a = 0$, $d_{sl}$ equals the slurry depth $d_0$, and at $a = \Delta a$, $d_{sl} = 0$. We assume that at $a = \Delta a/2$, half of the initial volume has infiltrated into the soil, so that

$$d_{sat}(a = \Delta a/2) = \frac{d_0}{2\varepsilon}. \tag{S22}$$

The depth $d_{sl}$ is obtained by subtracting the evaporation loss over $\Delta a/2$ from the remaining half of $d_0$:

$$d_{sl}(a = \Delta a/2) = \max\left(\frac{d_0 - \Delta a q_e}{2}, 0\right), \tag{S23}$$

[Figure]

Figure S1: Schematic description and the corresponding resistance chart for modeling a partly infiltrated slurry layer. The resistance within the slurry remaining on surface is $R_{\mathrm{sl}}$, the resistances within saturated soil are denoted by $R_{\mathrm{ss\uparrow}}$ and $R_{\mathrm{ss\downarrow}}$; other resistances are as in Fig. **??**. Labels a) to d) refer to TAN concentrations: a) [NH$_3$ (g)] at atmospheric reference height ; b) [NH$_3$ (g)] and [TAN(aq)] at the slurry surface; c) [TAN (aq)] in the slurry and saturated soil; d) [TAN (aq)] and [NH$_3$ (g)] at the bottom of the saturated soil layer. Thicknesses of the slurry and soil layers are denoted by $d_{sl}$, $d_{sat}$ and $d_{1/2}$ as described in the text.

which justifies the implicit assumption $d_{sl} \leq W/2$ in Fig. S1 and the Eqs. (S21).

The evaporation rate $q_e$ (m s$^{-1}$) for slurry is evaluated as

$$q_e = \frac{\rho_{air}}{\rho_w} \frac{Q_{sat} - Q_{atm}}{R_a + R_b}, \tag{S24}$$

where $\rho_{air}$ and $\rho_w$ are the densities of air and water, $Q_{atm}$ is specific humidity at the atmospheric reference height, $Q_{sat}$ is the specific humidity at saturation, and $R_a$ and $R_b$ are as in Eq. (??). The initial slurry depth $d_0$ is given by the slurry application rate (m$^3$ m$^{-2}$), and in the global simulations we assume $d_0 = 5$ mm, equal to 50 m$^3$ ha$^{-1}$.

The moisture flux $q_p$, required to evaluate the leaching flux (Eq. ??), is evaluated from the fraction of water in excess of $\Delta z \theta_s$ when the infiltration is complete,

$$q_p = \max \left( \frac{d_0 - \Delta a q_e - \Delta z \theta_s}{\Delta a}, 0 \right). \tag{S25}$$

where the cumulative evaporation is subtracted from the initial water volume.

The pH of slurry tends to increase after application due to volatilization of $CO_2$; a constant value 8.0 is used for pools S1 and S2 based on the data published by Sommer and Olesen (1991), Bussink et al. (1994) and Sherlock et al. (2002). Similar to G3, the pH for S3 is taken from the HWSD database.

**S2.3 Synthetic fertilizers**

Three TAN age classes (F1, F2 and F3) and two urea age classes (U1 and U2) are used to evaluate the volatilization losses for urea fertilizers (Fig. ??). The peak pH following urea application is often between 8 and 9 (Black et al., 1985; Whitehead and Raistrick, 1990; Sommer, 2013), and pHs of 7.0, 8.5 and 8.0 were chosen for F1, F2 and F3.

**S3 Nitrogen inputs**

**S3.1 Manure N excretion**

The estimated nitrogen excretion of livestock is based on the coefficients and animal weights in IPCC (2006). The yearly N excretion rates are listed in Table S1. For cattle, the coefficients are given for dairy and other cattle; the average is evaluated assuming 26 % fraction of dairy cattle for Europe, North America and Oceania , 35 % for India, and 14 % for other regions. The dairy cattle fraction for India is based on statistics published by the Indian Ministry of Agriculture and Farmers Welfare (2015), for other regions, the fraction is adapted from Bouwman et al. (1997). Geographical distributions of manure N in pastoral and landless production systems are shown in Fig. S2.

**S3.2 Fertilizers**

As noted in the main text (Section 2.5.2), the N fertilizer application in the simulations is prescribed in the CLM input dataset (Lawrence et al., 2016). The disaggregation to urea, ammonium, and nitrate N was based on the consumption statistics of the International Fertilizer Association (IFA) for the year 2013. The country-level values were used to derive gridded maps of fraction of urea and nitrate N. For countries with missing data for urea, the fraction was extrapolated from the neighboring areas. To ensure that the sum of urea and nitrate fraction remains below 1, the nitrate fraction was not extrapolated but assumed zero when missing. Maps of urea and other fertilizer use are shown in Fig. S3.

Table S1: Nitrogen excretion coefficients for livestock, kgN yr$^{-1}$ head $^{-1}$.

| | North America | Western Europe | Eastern Europe | Oceania | Latin America | Africa | Middle East | Asia | India |
|---|---|---|---|---|---|---|---|---|---|
| Cattle, average | 58.0 | 65.0 | 55.3 | 65.5 | 44.2 | 42.6 | 52.7 | 42.4 | 25.4 |
| Dairy cattle | 97.0 | 105.1 | 70.3 | 80.3 | 70.1 | 60.2 | 70.3 | 60.0 | 47.2 |
| Other cattle | 44.0 | 50.6 | 50.0 | 60.2 | 40.1 | 39.8 | 49.9 | 39.6 | 13.7 |
| Pigs, average | 11.2 | 16.1 | 17.0 | 15.6 | 16.8 | 16.8 | 16.8 | 5.1 | 5.1 |
| Market swine | 7.1 | 9.3 | 10.0 | 8.7 | 16.0 | 16.0 | 16.0 | 4.3 | 4.3 |
| Breeding swine | 17.3 | 30.4 | 30.2 | 30.2 | 5.6 | 5.6 | 5.6 | 2.5 | 2.5 |
| Poultry | 0.5 | 0.5 | 0.5 | 0.5 | 0.5 | 0.5 | 0.5 | 0.5 | 0.5 |
| Sheep | 7.4 | 15.0 | 15.9 | 20.0 | 12.0 | 12.0 | 12.0 | 12.0 | 12.0 |
| Goats | 6.3 | 18.0 | 18.0 | 20.0 | 15.0 | 15.0 | 15.0 | 15.0 | 15.0 |
| Buffalo | 44.4 | 44.4 | 44.4 | 44.4 | 44.4 | 44.4 | 44.4 | 44.4 | 34.5 |

[Figure]

Figure S2: Manure N production (gN m$^{-2}$yr$^{-1}$) in pastoral (left) and mixed/landless systems (right).

[Figure]

Figure S3: Yearly application (gN m$^{-2}$yr$^{-1}$) of urea (left) and other N fertilizers (right).

**S4 Grazing and housing periods in mixed systems**

Ruminants in mixed/landless systems are assumed to graze when the 10-day mean daily minimum temperature exceeds $+10°$ C. At other times the livestock is assumed to remain in animal housings. Fig. S4 shows the number of yearly housing days estimated using the temperature threshold and global temperature data from the NCEP reanalyses for 1980-2010. Fig. S5 compares the yearly housing days for Europe with the survey results (for cattle) reported by Klimont and Brink (2004). The values shown for European countries are weighted averages; the distribution of housing days was weighted by the population distribution of cattle within each country.

**S5 Sensitivity experiments**

The sensitivity to model parameters was examined with experiments consisting of 2-year CLM simulations forced by the GSWP3 dataset. The experiments, the modified parameters, and changes in $NH_3$ emission are listed in Table S2. Switching the meteorological forcing from the CAM simulation to GSWP3 changed to total emission by $\sim 2$ %; all other changes in emissions are reported with regard to the GSWP3-driven control simulation.

In the experiments evaluating sensitivity to the layer thickness $\Delta z$, only the thickness used in FAN computations was changed; the soil layers used in elsewhere in CLM were not changed. The water content used in FAN computations was taken for the topmost CLM layer in all experiments.

The sensitivity to manure TAN fraction by was evaluated by computing the emissions at 0 and 100 % TAN fractions. Since the model is linear with respect to N input, this allows calculating the  $NH_3$ volatilization for any TAN fraction $f_{\text{TAN}}$ as

$$F_M(f_{\text{TAN}}) = f_{\text{TAN}} \times F_M(f_{\text{TAN}} = 1) + (1 - f_{\text{TAN}}) \times F_M(f_{\text{TAN}} = 0), \qquad \text{(S26)}$$

where $F_M$ denotes the total  $NH_3$ emission from manure. In the experiments, $F_M(0) = 6.8$ Tg N and $F_M(1) = 56$ Tg N. Varying the TAN fraction between 40 and 80 % would therefore result in about $\pm 10$ TgN variation in the global  $NH_3$ emission.

[Figure]

Figure S4: Average number of housing days per year estimated using the 1980-2010 NCEP temperature reanalyses.

[Figure]

Figure S5: Yearly housing days reported by Klimont and Brink (2004) (left) and estimated using a temperature threshold (right).

Table S2: Relative changes in  NH$_3$ emissions in the sensitivity experiments. The change for the control run is relative to the main run; for other experiments the change with respect to the control run is shown. Percent change in emission per percent change in parameter is shown in parentheses when applicable.

| Parameter | Value | Africa | Asia except China and India | China | Europe | India | Latin America | North America | Oceania | World Total | World Fertilizer | World Ma... |
|---|---|---|---|---|---|---|---|---|---|---|---|---|
| Control[1] | | +8 | +2 | −0 | −1 | +1 | +4 | +4 | +1 | +2 | +6 | +1 |
| $\tau_{\mathrm{infl}}$[2] | ×0.5 | −2 | −2 | −3 |  −5 | −2 | −3 | −3 | −3 | −3 (+0.1) | +0 | −3 |
| | ×2.0 | +2 | +3 | +5 | +6 | +2 | +3 | +4 | +4 | +3 (+0.0) | +0 | +4 |
| $d_0$ for urine[3] | −2 mm | +7 | +4 | +2 | +1 | +4 |  +5 | +2 | +4 | +4 (−0.1) | +0 | +5 |
| | +2 mm | −6 | −3 | −2 | −1 | −3 | −5 | −2 | −5 |  −3 (−0.1) | +0 | −4 |
| pH for manure[4] | +0.5 | +9 | +8 |  +8 |  +12 | +6 | +9 | +9 | +19 |  +9 | +0 | +1 |
| | −0.5 | −8 | −6 | −6 | −10 | −4 |  −7 | −7 | −11 | −7 | +0 | −9 |
| $f_{\mathrm{TAN}}$[5] | 0.0 | −70 | −62 | −57 | −77 | −46 | −72 | −54 | −64 |  −63 (+0.6) | +0 | −8 |
| | 1.0 | +46 | +42 | +38 |  +52 | +31 | +48 | +36 | +43 | +42 (+0.6) | +0 | +54 |
| pH for other fert.[6] | 7.0 | +1 | +2 | +1 | +8 | +0 | +3 | +11 | +0 | +3 | +12 | +0 |
| pH for urea[7] | | −0 | −0 | +2 | −1 | +1 | −2 | −0 | +2 | +0 | +1 | +0 |
| Urea decomp.[8] | ×0.5 | +0 | +2 | +4 | +1 | +3 | +1 | +3 | +2 | +2 (−0.0) | +9 (−0.2) | +0 |
| | ×2.0 | −0 | −3 | −5 | −2 | −4 | −2 | −3 | −2 | −3 (−0.0) | −11 (−0.1) | +0 |
| Fert. timing[9] | 1 day | +0 | −2 | −0 | +2 | −4 | +0 | +1 | −1 | −1 | −3 | +0 |
| | 90 days | −1 | +4 | −6 | −2 | +0 | −1 | +2 | +5 | −0 | −2 | +0 |
| $\Delta z$[10] | ×0.5 | +5 |  +12 |  +22 |  +19 |  +13 |  +13 |  +29 |  +24 |  +15 (−0.3) | +41 (−0.8) |  |
| | ×2.0 | −28 |  −26 | −27 |  −20 | −30 | −24 | −31 | −33 | −27 (−0.3) | −52 (−0.5) | −19 |
| $K_d$[11] | 0 | +9 | +12 | +20 | +17 | +13 | +16 | +21 |  +24 | +15  (−0.2) | +30 (−0.3) | +11 |
| | ×10.0 | −35 | −31 | −38 | −32 | −36 | −34 | −40 |  −43 | −35 (−0.0) | −55 (−0.1) | −29 |
| Fraction of grazing[12] | +0.30 | −17 | −12 | −8 | −6 | −19 | −20 | −5 | −15 | −14 (−0.3) | +0 | −18 |
| | −0.30 |  +17 |  +12 | +8 | +7 |  +19 |  +21 |  +5 |  +15 | +14  (−0.3) | +0 | +16 |

[1]Control run with GSWP3 forcing. [2]Slurry infiltration time. [3]See Section 2.4.2. [4]Add or subtract from pH of N classes G1–G3 and S0–S3 . [5]Fraction of TAN in manure. [6]Fixed pH for age class F4. [7]pH for N classes F1, F2, F3 set to soil pH + 0.5, 2, 1.5 units. [8]Time constant $1/k_U$. [9]Fertilizer application window, days from leaf emergence. [10]Soil layer thickness. [11]Adsorption constant. [12]Maximum fraction of ruminants grazing in mixed production systems.

**S6 Sensitivity to mean temperature and precipitation**

The sensitivity of  $NH_3$ emissions to mean temperature was investigated with a linear regression on the geographical distribution of the simulated emissions. The model grid cells were first categorized by yearly rainfall, and then, the normalized volatilization loss ( $NH_3$ emitted / N applied) within each category was fit with a linear function of the yearly mean temperature, assuming that for each grid cell,

$$NH_3/N_{appl} = a + bT + r \tag{S27}$$

where $a$ and $b$ are the regression parameters, $T$ is the temperature ($°C$) and the residual $r$ represents the temperature-independent effects. The temperature sensitivity is obtained from Eq. (S27) as

$$\Delta(NH_3) = bN_{appl}\Delta T. \tag{S28}$$

The temperature sensitivity of the total emission is then obtained by summing Eq. (S28) over the gridcells and precipitation categories.

The regression was evaluated separately for manure, urea, and other synthetic fertilizers. Exponential fits were tested in addition to the linear fits, however, the exponential fits invariably had lower  $R^2$ than the linear fits and thus were not analyzed further.

Results of the regression and  the temperature response are shown in Table S3.

Table S3: Parameters of linear fits of the normalized volatilization loss (N emitted / N applied) as function of local mean temperature in °C. Present day manure N production or fertilizer N application and the corresponding  $NH_3$ emissions are shown for each category.

| Source |  Precipitation, mm | Intercept | Slope, $K^{-1}$ | $R^2$ | N applied, Tg | $NH_3$ emitted, Tg N |
|---|---|---|---|---|---|---|
| Manure | < 200 | 0.13 | 0.009 | 0.87 | 8.5 | 2.9 |
|  | 200 − 500 | 0.15 | 0.008 | 0.65 | 20.8 | 6.4 |
|  | 500 − 1000 | 0.18 | 0.006 |  0.40 | 41.1 |  12.5 |
|  | 1000 − 2000 |  0.19 | 0.005 | 0.18 | 36.4 |  11.2 |
|  | > 2000 |  0.15 | 0.006 | 0.07 | 12.8 |  3.6 |
|  | Total |  |  |  | 119.6 |  36.5 |
| Urea | < 200 | 0.20 | 0.005 | 0.05 $(p = 0.01)$ | 3.1 | 0.6 |
|  | 200 − 500 | 0.00 | 0.013 | 0.48 | 8.5 | 1.6 |
|  | 500 − 1000 | -0.03 | 0.014 | 0.70 | 13.1 | 2.4 |
|  | 1000 − 2000 | 0.00 | 0.010 | 0.45 | 12.8 | 2.8 |
|  | > 2000 | 0.04 | 0.005 | 0.09 | 4.7 | 0.7 |
|  | Total |  |  |  | 42.2 | 8.1 |
| Other fert. | < 200 | -0.01 | 0.010 | 0.29 | 1.0 | 0.2 |
|  | 200 − 500 | -0.05 | 0.010 | 0.33 | 7.7 | 0.8 |
|  | 500 − 1000 | -0.05 | 0.007 | 0.34 | 20.9 | 1.4 |
|  | 1000 − 2000 | -0.04 | 0.005 | 0.22 | 8.2 | 0.5 |
|  | > 2000 | 0.00 | 0.001 | 0.03 $(p = 0.02)$ | 2.6 | 0.1 |
|  | Total |  |  |  | 40.4 | 2.9 |
| All sources | Total |  |  |  | 202.1 |  47.5 |

---

## Author Response (AR2)

We thank the reviewers for their feedback and for notifying us for the missing cross references between the supplement and the main text. Below we give responses to the comments of reviewer #3.

*Page 5, Equation 3: I'm Confused by units. Do brackets mean moles/L? The line above says NH3 g is in gN m-3*

Thank you for pointing this out. The concentrations in Eq. (4) should be molar. However, since $K_{NH4}$ is in practice given by Eq. (A11), only the hydrogen ion concentration needs to be expressed in mol $L^{-1}$ in the model. Therefore, gN $m^{-3}$ can be used for all prognostic concentrations.

To avoid the inconsistency, we have removed Eq. (4) and instead refer to Eq. (A11).

*Page 5, line 27: Wouldn't the conversion of Kd to ml g-1 depend on soil moisture content?*

Buss et al. (2004) define the partitioning coefficient as $K_d = C^*/C$, where C* is the adsorbed concentration (mg contaminant $kg^{-1}$ soil) and C is the aqueous phase concentration (mg $L^{-1}$ water). Since the $K_d$ in FANv2 is also formulated using the aqueous phase concentration [TAN (aq)], the conversion is independent of the soil moisture content. Soil particle density is needed to convert the adsorbed concentration from per-volume to per-mass.

Some models define Kd using the concentration in bulk soil volume instead of the aqueous phase volume, and in this case, the conversion would depend on moisture content.

*Page 7, line 22: Should mention here that the tortuosity equation is shown in Appendix A.*

Done.

*Page 17, line 31: It is stated that the ammonia emissions are linear wrt N input in FANv2, but shouldn't it depend temperature, soil moisture, and other meteorological forcings that might make it nonlinear?*

We mean that for a given meteorological forcing, the NH3 emission scales linearly with respect to the N input, which is a consequence of the linear partitioning coefficients (Eq. 3) and the linear reaction rates (Eq. 2) together with the resistance formulation (Eqs. 6 and 7).

[revised manuscript text omitted]
_{\text{NH3}})^{1/2}}{p[(\Sigma_{air}v_i)^{1/3} + (\Sigma_{\text{NH3}}v_i)^{1/3}]^2}, \quad \text{(A9)}$$ where $M_{air} = 29.0$, $M_{\text{NH3}} = 17.0$, $\Sigma_{air}v_i = 20.1$, $\Sigma_{\text{NH3}}v_i = 14.9$ and $p = 1.0$ (Fuller et al., 1966). |
| **Equilibrium constants** | | | |
| $K_H$ | | NH$_3$ (aq) $\rightleftharpoons$ NH$_3$ (g) | $$K_H = (4.59\ K^{-1})T_g e^{4092(1/T_g - 1/T_{\text{ref}})}, \quad \text{(A10)}$$ (Sutton et al., 1994), where  $T_{\text{ref}} = 298.15$ K. |
| $K_{\text{NH4}}$ | mol l$^{-1}$ | NH$_3$ + H$_2$O $\rightleftharpoons$ NH4$^+$ + OH | $$K_{\text{NH4}} = 5.67 \times 10^{-10} e^{-6286(1/T_g - 1/T_{\text{ref}})}, \quad \text{(A11)}$$ (Sutton et al., 1994), where  $T_{\text{ref}}$ is as in Eq. (A10). |
| $K_{\text{NH3}}$ | | [NH3 (g)] / [TAN (aq)] | $$K_{\text{NH3}} = \frac{1}{K_H + [\text{H}^+]/K_{\text{NH4}}} \quad \text{(
[revised manuscript text omitted]

**S1 Patch aging scheme**

The upscaling scheme used in FANv2 is based on a generic approach for evaluating the aggregate N fluxes for a distribution of patches based on the fluxes of a single patch. The main underlying assumptions are that

- Each patch has a well-defined age $a > 0$ which is equal to the time elapsed since the patch was created.

- The state of each patch is uniquely determined by its age $a$ and the current time $t$.

- The patches are aggregated over an area large enough that the distribution of the state variables can be approximated by integrable functions of $a$ and $t$.

Under these assumptions, the upscaling approach aims to link the evolution of the nitrogen content $N(t)$ of a single patch, given by a differential equation in the form

$$\frac{dN}{dt} = f(N, t), \tag{S1}$$

to an age and time dependent function $n(a, t)$ which describes how the nitrogen is distributed between the patches of different age.

We start by defining the density function $w(a, t)$ such that the surface area occupied by patches aged between $a$ and $a + \Delta a$ is given by

$$W(t) = \int_a^{a+\Delta a} w(a', t) da'. \tag{S2}$$

Since the quantities simulated by FANv2 are expressed as area densities, $W(t)$ is a fraction (area of patches / total area) and $w(a, t)$ has the unit $s^{-1}$.

Any feature of the patches is now expressed as two-dimensional functions $\phi(a, t)$, and for every fixed age $a_0$, the single-variable functions

$$\Phi(t) = \phi(a_0 + t, t) \tag{S3}$$

define the evolution (Eq. S1) of an individual patch aged $a_0$ at $t = 0$. The total derivative of $\phi(a, t)$ can therefore be identified with the tendency of $\Phi$,

$$\frac{\partial \phi}{\partial a} + \frac{\partial \phi}{\partial t} = \frac{d\Phi}{dt}, \tag{S4}$$

where we substituted $\partial a / \partial t = 1$. Since FANv2 assumes that the area of a patch does not change with $a$, application of (S4) to the density function $w(a, t)$ yields a "continuity equation" for $w$:

$$\frac{\partial w}{\partial a} + \frac{\partial w}{\partial t} = 0. \tag{S5}$$

The equation (S1) governing the nitrogen density of a single patch can then be understood in the sense of Eq. (S4):

$$\frac{\partial N}{\partial t} + \frac{\partial N}{\partial a} = f(N, a, t). \tag{S6}$$

Combining Eqs. (S5) and (S6) yields and equation for the area-weighted N density $n(a, t) = w(a, t)N(a, t)$:

$$\frac{d}{dt}(wN) = \frac{\partial n}{\partial a} + \frac{\partial n}{\partial t} = w(a, t)f(N(a, t), a, t). \tag{S7}$$

If $f$ is linear with respect to $N$, Eq. (S7) simplifies to

$$\frac{\partial n}{\partial a} + \frac{\partial n}{\partial t} = f(n(a, t), a, t). \tag{S8}$$

Eq. (S7) is a hyperbolic first-order partial differential equation.  Applying a first-order, forward-in-time discretization yields a two-step numerical scheme implemented in FANv2:

1. For each $i$, update $n_i(t)$ according to Eq. (S1) as

$$n_i(t') = n_i(t) + w_i(t)\overline{f(N_i(t), t)}\Delta t \tag{S9}$$

where $\Delta t$ denotes the time step and the tendency $\overline{f(N_i(t), t)}$ is evaluated as a mean over the $i$th age class.

2. Transfer nitrogen from the younger to older age classes according to

$$n_i(t + \Delta t) = \begin{cases} n_i(t') - \Delta t \frac{n_i(t')}{\Delta a_i}, i = 1 \\ n_i(t') - \Delta t \left( \frac{n_i(t')}{\Delta a_i} - \frac{n_{i-1}(t')}{\Delta a_{i-1}} \right), i > 1 \end{cases} \tag{S10}$$

where the ages $a_{i-1}$ and $a_i$ define the $i$th age class and $\Delta a_i = a_i - a_{i-1}$.

In FANv2, the tendency $f$ in Eq. (S1) is linear with respect to $N$. Substituting $N_i = n_i/w_i$ simplifies Eq. (S9) to

$$n_i(t') = n_i(t) + \overline{f(n_i(t), t)}\Delta t, \tag{S11}$$

and the area fractions $w_i$ are therefore not needed.

The fertilizer or manure N is initially introduced to the youngest age class, and subsequently transferred through the sequence of age classes as described by Eq. (S10), until reaching the final class $i*$. By Eq. (S10), nitrogen is removed from the final age class at a rate equal to $1/\Delta a_{i*}$, which can be made arbitrarily small by the choice of $\Delta a_{i*}$. In FANv2, the final bins have $\Delta a_{i*} = 360$ days, which sets the maximum age of the N patches considered. Although not implemented in the current version, the nitrogen aged beyond $\Delta a_{i*}$ could be transferred into the soil N pools in the CLM.

If the tendency $f(N, a, t)$ is nonlinear for $N$, Eq. (S7) requires evaluating the density function $w(a, t)$. This can be obtained from the exact solution to Eq. (S5),

$$w(a, t) = w(0, t - a), \tag{S12}$$

which is the area fraction occupied by fresh patches at time $t - a$. Alternatively, $w(a, t)$ could be evaluated numerically with a similar procedure as for $n(a, t)$, which has the advantage of guaranteeing consistency between the evolution of $w(a, t)$ and $n(a, t)$.

**S1.1  Application to moisture content**

The framework described above can be used to aggregate the water budget for a class of patches. This is based on the assumption that the volumetric soil moisture of the patches can be expressed as a sum of a $t$-dependent background and an $a$-dependent perturbation:

$$\theta(a, t) = \theta_b(t) + \delta\theta(a). \tag{S13}$$

Applying Eq. (S4) to the single-patch water budget yields

$$\frac{d\theta}{dt} = \frac{\partial\theta}{\partial t} + \frac{\partial\theta}{\partial a} = \frac{q_{top} - q_{bot}}{\Delta z}, \tag{S14}$$

where $q_{top}$ and $q_{bot}$ are the water fluxes at the top and bottom of the layer with thickness $\Delta z$. Substituting Eq. (S13) and integrating Eq. (S14) over the age range $(a_i, a_{i+1})$ yields the budget averaged over the age class, as in Eq. (??) in the main text:

$$\frac{\overline{q_{top} - q_{bot}}}{\Delta z} = \overline{\frac{d\theta_b}{dt}} - \frac{\delta\theta(a_{i+1}) + \delta\theta(a_i)}{a_{i+1} - a_i}, \tag{S15}$$

where the lines denote $a$-averages, such as

$$\overline{q_{top}} = \frac{\int_{a_i}^{a_{i+1}} q_{top} \, da}{a_{i+1} - a_i}. \tag{S16}$$

**S2 Sub-models for pastures, slurry and synthetic fertilizers**

**S2.1 Grazed pastures**

[revised manuscript text omitted]

which justifies the implicit assumption $d_{sl} \le W/2$ in Fig. S1 and the Eqs. (S21).

[Figure]

Figure S1: Schematic description and the corresponding resistance chart for modeling a partly infiltrated slurry layer. The resistance within the slurry remaining on surface is $R_{sl}$, the resistances within saturated soil are denoted by $R_{ss\uparrow}$ and $R_{ss\downarrow}$; other resistances are as in Fig. ??1. Labels a) to d) refer to TAN concentrations: a) [NH$_3$ (g)] at atmospheric reference height ; b) [NH$_3$ (g)] and [TAN(aq)] at the slurry surface; c) [TAN (aq)] in the slurry and saturated soil; d) [TAN (aq)] and [NH$_3$ (g)] at the bottom of the saturated soil layer. Thicknesses of the slurry and soil layers are denoted by $d_{sl}$, $d_{sat}$ and $d_{1/2}$ as described in the text.

The evaporation rate $q_e$ (m s$^{-1}$) for slurry is evaluated as

$$q_e = \frac{\rho_{air}}{\rho_w} \frac{Q_{sat} - Q_{atm}}{R_a + R_b},$$ (S24)

where $\rho_{air}$ and $\rho_w$ are the densities of air and water, $Q_{atm}$ is specific humidity at the atmospheric reference height, $Q_{sat}$ is the specific humidity at saturation, and $R_a$ and $R_b$ are as in Eq. (??6). The initial slurry depth $d_0$ is given by the slurry application rate (m$^3$ m$^{-2}$), and in the global simulations we assume $d_0 = 5$ mm, equal to 50 m$^3$ ha$^{-1}$.

The moisture flux $q_p$, required to evaluate the leaching flux (Eq. ??9), is evaluated from the fraction of water in excess of $\Delta z \theta_s$ when the infiltration is complete,

$$q_p = \max\left(\frac{d_0 - \Delta a q_e - \Delta z \theta_s}{\Delta a}, 0\right).$$ (S25)

where the cumulative evaporation is subtracted from the initial water volume.

The pH of slurry tends to increase after application due to volatilization of $CO_2$; a constant value 8.0 is used for pools S1 and S2 based on the data published by Sommer and Olesen (1991), Bussink et al. (1994) and Sherlock et al. (2002). Similar to G3, the pH for S3 is taken from the HWSD database.

**S2.3 Synthetic fertilizers**

Three TAN age classes (F1, F2 and F3) and two urea age classes (U1 and U2) are used to evaluate the volatilization losses for urea fertilizers (Fig. 3). The peak pH following urea application is often between 8 and 9 (Black et al., 1985; Whitehead and Raistrick, 1990; Sommer, 2013), and pHs of 7.0, 8.5 and 8.0 were chosen for F1, F2 and F3.

**S3 Nitrogen inputs**

**S3.1 Manure N excretion**

The estimated nitrogen excretion of livestock is based on the coefficients and animal weights in IPCC (2006). The yearly N excretion rates are listed in Table S1. For cattle, the coefficients are given for dairy and other cattle; the average is evaluated assuming 26 % fraction of dairy cattle for Europe, North America and Oceania , 35 % for India, and 14 % for other regions. The dairy cattle fraction for India is based on statistics published by the Indian Ministry of Agriculture and Farmers Welfare (2015), for other regions, the fraction is adapted from Bouwman et al. (1997). Geographical distributions of manure N in pastoral and landless production systems are shown in Fig. S2.

**S3.2 Fertilizers**

As noted in the main text (Section 2.5.2), the N fertilizer application in the simulations is prescribed in the CLM input dataset (Lawrence et al., 2016). The disaggregation to urea, ammonium, and nitrate N was based on the consumption statistics of the International Fertilizer Association (IFA) for the year 2013. The country-level values were used to derive gridded maps of fraction of urea and nitrate N. For countries with missing data for urea, the fraction was extrapolated from the neighboring areas. To ensure that the sum of urea and nitrate fraction remains below 1, the nitrate fraction was not extrapolated but assumed zero when missing. Maps of urea and other fertilizer use are shown in Fig. S3.

**S4 Grazing and housing periods in mixed systems**

Ruminants in mixed/landless systems are assumed to graze when the 10-day mean daily minimum temperature exceeds $+10°$ C. At other times the livestock is assumed to remain in animal housings. Fig. S4 shows the number of yearly housing days estimated using the temperature threshold and

Table S1: Nitrogen excretion coefficients for livestock, kgN yr$^{-1}$ head $^{-1}$.

| | North America | Western Europe | Eastern Europe | Oceania | Latin America | Africa | Middle East | Asia | India |
|---|---|---|---|---|---|---|---|---|---|
| Cattle, average | 58.0 | 65.0 | 55.3 | 65.5 | 44.2 | 42.6 | 52.7 | 42.4 | 25.4 |
| Dairy cattle | 97.0 | 105.1 | 70.3 | 80.3 | 70.1 | 60.2 | 70.3 | 60.0 | 47.2 |
| Other cattle | 44.0 | 50.6 | 50.0 | 60.2 | 40.1 | 39.8 | 49.9 | 39.6 | 13.7 |
| Pigs, average | 11.2 | 16.1 | 17.0 | 15.6 | 16.8 | 16.8 | 16.8 | 5.1 | 5.1 |
| Market swine | 7.1 | 9.3 | 10.0 | 8.7 | 16.0 | 16.0 | 16.0 | 4.3 | 4.3 |
| Breeding swine | 17.3 | 30.4 | 30.2 | 30.2 | 5.6 | 5.6 | 5.6 | 2.5 | 2.5 |
| Poultry | 0.5 | 0.5 | 0.5 | 0.5 | 0.5 | 0.5 | 0.5 | 0.5 | 0.5 |
| Sheep | 7.4 | 15.0 | 15.9 | 20.0 | 12.0 | 12.0 | 12.0 | 12.0 | 12.0 |
| Goats | 6.3 | 18.0 | 18.0 | 20.0 | 15.0 | 15.0 | 15.0 | 15.0 | 15.0 |
| Buffalo | 44.4 | 44.4 | 44.4 | 44.4 | 44.4 | 44.4 | 44.4 | 44.4 | 34.5 |

[Figure]

Figure S2: Manure N production (gN m$^{-2}$yr$^{-1}$) in pastoral (left) and mixed/landless systems (right).

[Figure]

Figure S3: Yearly application (gN m$^{-2}$yr$^{-1}$) of urea (left) and other N fertilizers (right).

global temperature data from the NCEP reanalyses for 1980-2010. Fig. S5 compares the yearly housing days for Europe with the survey results (for cattle) reported by Klimont and Brink (2004). The values shown for European countries are weighted averages; the distribution of housing days was weighted by the population distribution of cattle within each country.

**S5    Sensitivity experiments**

The sensitivity to model parameters was examined with experiments consisting of 2-year CLM simulations forced by the GSWP3 dataset. The experiments, the modified parameters, and changes in $NH_3$ emission are listed in Table S2. Switching the meteorological forcing from the CAM simulation to GSWP3 changed to total emission by $\sim 2$ %; all other changes in emissions are reported with regard to the GSWP3-driven control simulation.

In the experiments evaluating sensitivity to the layer thickness $\Delta z$, only the thickness used in FAN computations was changed; the soil layers used in elsewhere in CLM were not changed. The water content used in FAN computations was taken for the topmost CLM layer in all experiments.

The sensitivity to manure TAN fraction by was evaluated by computing the emissions at 0 and 100 % TAN fractions. Since the model is linear with respect to N input, this allows calculating the $NH_3$ volatilization for any TAN fraction $f_{\mathrm{TAN}}$ as

$$F_M(f_{\mathrm{TAN}}) = f_{\mathrm{TAN}} \times F_M(f_{\mathrm{TAN}} = 1) + (1 - f_{\mathrm{TAN}}) \times F_M(f_{\mathrm{TAN}} = 0), \tag{S26}$$

where $F_M$ denotes the total $NH_3$ emission from manure. In the experiments, $F_M(0) = 6.8$ Tg N and $F_M(1) = 56$ Tg N. Varying the TAN fraction between 40 and 80 % would therefore result in about $\pm 10$ TgN variation in the global $NH_3$ emission.

[Figure]

Figure S4: Average number of housing days per year estimated using the 1980-2010 NCEP temperature reanalyses.

[Figure]

Figure S5: Yearly housing days reported by Klimont and Brink (2004) (left) and estimated using a temperature threshold (right).

Table S2: Relative changes in $NH_3$ emissions in the sensitivity experiments. The change for the control run is relative to the main run; for other experiments the change with respect to the control run is shown. Percent change in emission per percent change in parameter is shown in parentheses when applicable.

| Parameter | Value | Africa | Asia except China and India | China | Europe | India | Latin America | North America | Oceania | World Total | World Fertilizer | World Manure |
|---|---|---|---|---|---|---|---|---|---|---|---|---|
| | | | | | | | | | | Percent difference in $NH_3$ emission | | |
| Control[1] | | +8 | +2 | −0 | −1 | +1 | +4 | +4 | +1 | +2 | +6 | +1 |
| $\tau_{\text{infl}}$[2] | ×0.5 | −2 | −2 | −3 | −5 | −2 | −3 | −3 | −3 | −3 (+0.1) | +0 | −3 (+0.1) |
| | ×2.0 | +2 | +3 | +5 | +6 | +2 | +3 | +4 | +4 | +3 (+0.0) | +0 | +4 (+0.0) |
| $d_0$ for urine [3] | −2 mm | +7 | +4 | +2 | +1 | +4 | +5 | +2 | +4 | +4 (−0.1) | +0 | +5 (−0.1) |
| | +2 mm | −6 | −3 | −2 | −1 | −3 | −5 | −2 | −5 | −3 (−0.1) | +0 | −4 (−0.1) |
| pH for manure[4] | +0.5 | +9 | +8 | +8 | +12 | +6 | +9 | +9 | +19 | +9 | +0 | +11 |
| | −0.5 | −8 | −6 | −6 | −10 | −4 | −7 | −7 | −11 | −7 | +0 | −9 |
| $f_{\text{TAN}}$[5] | 0.0 0 | −70 | −62 | −57 | −77 | −46 | −72 | −54 | −64 | −63 (+0.6) | +0 | −81 (+0.8) |
| | 1.0 1 | +46 | +42 | +38 | +52 | +31 | +48 | +36 | +43 | +42 (+0.6) | +0 | +54 (+0.8) |
| pH for other fert.[6] | 7.0 7 | +1 | +2 | +1 | +8 | +0 | +3 | +11 | +0 | +3 | +12 | +0 |
| pH for urea[7] | | −0 | −0 | +2 | −1 | +1 | −2 | −0 | +2 | +0 | +1 | +0 |
| Urea decomp.[8] | ×0.5 | +0 | +2 | +4 | +1 | +3 | +1 | +3 | +2 | +2 (−0.0) | +9 (−0.2) | +0 |
| | ×2.0 | −0 | −3 | −5 | −2 | −4 | −2 | −3 | −2 | −3 (−0.0) | −11 (−0.1) | +0 |
| Fert. timing[9] | 1 day | +0 | −2 | −0 | +2 | −4 | +0 | +1 | −1 | −1 | −3 | +0 |
| | 90 days | −1 | +4 | −6 | −2 | −1 | −1 | +2 | +5 | −0 | −2 | +0 |
| $\Delta z$[10] | ×0.5 | +5 | +12 | +22 | +19 | +13 | +13 | +29 | +24 | +15 (−0.3) | +41 (−0.8) | +7 (−0.1) |
| | ×2.0 | −28 | −26 | −27 | −20 | −30 | −24 | −31 | −33 | −27 (−0.3) | −52 (−0.5) | −19 (−0.2) |
| $K_d$[11] | 0 | +9 | +12 | +20 | +17 | +13 | +16 | +21 | +24 | +15 (−0.2) | +30 (−0.3) | +11 (−0.1) |
| | ×10.0 | −35 | −31 | −38 | −32 | −36 | −34 | −40 | −43 | −35 (−0.0) | −55 (−0.1) | −29 (−0.0) |
| Fraction of grazing[12] | +0.30 | −17 | −12 | −8 | −6 | −19 | −20 | −5 | −15 | −14 (−0.3) | +0 | −18 (−0.4) |
| | −0.30 | +17 | +12 | +8 | +7 | +19 | +21 | +5 | +15 | +14 (−0.3) | +0 | +18 (+0.4) |

[1]Control run with GSWP3 forcing. [2]Slurry infiltration time. [3]See Section 2.4.2 S2.1. [4]Add or subtract from pH of N classes G1–G3 and S0–S3 . [5]Fraction of TAN in manure. [6]Fixed pH for age class F4. [7]pH for N classes F1, F2, F3 set to soil pH + 0.5, 2, 1.5 units. [8]Time constant $1/k_U$. [9]Fertilizer application window, days from leaf emergence. [10]Soil layer thickness. [11]Adsorption constant. [12]Maximum fraction of ruminants grazing in mixed production systems.

**S6 Sensitivity to mean temperature and precipitation**

The sensitivity of NH$_3$ emissions to mean temperature was investigated with a linear regression on the geographical distribution of the simulated emissions. The model grid cells were first categorized by yearly rainfall, and then, the normalized volatilization loss (NH$_3$ emitted / N applied) within each category was fit with a linear function of the yearly mean temperature, assuming that for each grid cell,

$$\text{NH}_3/\text{N}_{\text{appl}} = a + bT + r \tag{S27}$$

where $a$ and $b$ are the regression parameters, $T$ is the temperature (°C) and the residual $r$ represents the temperature-independent effects. The temperature sensitivity is obtained from Eq. (S27) as

$$\Delta(\text{NH}_3) = b\text{N}_{\text{appl}}\Delta T. \tag{S28}$$

The temperature sensitivity of the total emission is then obtained by summing Eq. (S28) over the gridcells and precipitation categories.

The regression was evaluated separately for manure, urea, and other synthetic fertilizers. Exponential fits were tested in addition to the linear fits, however, the exponential fits invariably had lower R$^2$ than the linear fits and thus were not analyzed further. Results of the regression and the temperature response are shown in Table S3.

Table S3: Parameters of linear fits of the normalized volatilization loss (N emitted / N applied) as function of local mean temperature in °C. Present day manure N production or fertilizer N application and the corresponding NH$_3$ emissions are shown for each category.

| Source | Precipitation, mm | Intercept | Slope, K$^{-1}$ | $R^2$ | N applied, Tg | NH$_3$ emitted, Tg N | Temperature sensitivity, % K$^{-1}$ |
|---|---|---|---|---|---|---|---|
| Manure | $< 200$ | 0.13 | 0.009 | 0.87 | 8.5 | 2.9 | 2.5 |
| | $200 - 500$ | 0.15 | 0.008 | 0.65 | 20.8 | 6.4 | 2.5 |
| | $500 - 1000$ | 0.18 | 0.006 | 0.40 | 41.1 | 12.5 | 1.9 |
| | $1000 - 2000$ | 0.19 | 0.005 | 0.18 | 36.4 | 11.2 | 1.7 |
| | $> 2000$ | 0.15 | 0.006 | 0.07 | 12.8 | 3.6 | 2.0 |
| | Total | | | | 119.6 | 36.5 | 2.0 |
| Urea | $< 200$ | 0.20 | 0.005 | 0.05 ($p = 0.01$) | 3.1 | 0.6 | 2.4 |
| | $200 - 500$ | 0.00 | 0.013 | 0.48 | 8.5 | 1.6 | 7.1 |
| | $500 - 1000$ | -0.03 | 0.014 | 0.70 | 13.1 | 2.4 | 7.5 |
| | $1000 - 2000$ | 0.00 | 0.010 | 0.45 | 12.8 | 2.8 | 4.7 |
| | $> 2000$ | 0.04 | 0.005 | 0.09 | 4.7 | 0.7 | 3.2 |
| | Total | | | | 42.2 | 8.1 | 5.7 |
| Other fert. | $< 200$ | -0.01 | 0.010 | 0.29 | 1.0 | 0.2 | 6.1 |
| | $200 - 500$ | -0.05 | 0.010 | 0.33 | 7.7 | 0.8 | 9.3 |
| | $500 - 1000$ | -0.05 | 0.007 | 0.34 | 20.9 | 1.4 | 10.8 |
| | $1000 - 2000$ | -0.04 | 0.005 | 0.22 | 8.2 | 0.5 | 9.0 |
| | $> 2000$ | 0.00 | 0.001 | 0.03 ($p = 0.02$) | 2.6 | 0.1 | 4.1 |
| | Total | | | | 40.4 | 2.9 | 9.7 |
| All sources | Total | | | | 202.1 | 47.5 | 3.1 |